# DISENTANGLING IMPROVES VAEs' ROBUSTNESS TO ADVERSARIAL ATTACKS

## ABSTRACT

This paper is concerned with the robustness of VAEs to adversarial attacks. We highlight that conventional VAEs are brittle under attack but that methods recently introduced for disentanglement such as $\beta$-TCVAE (Chen et al., 2018) improve robustness, as demonstrated through a variety of previously proposed adversarial attacks (Tabacof et al. (2016); Gondim-Ribeiro et al. (2018); Kos et al.(2018)). This motivated us to develop *Seatbelt-VAE*, a new hierarchical disentangled VAE that is designed to be significantly more robust to adversarial attacks than existing approaches, while retaining high quality reconstructions.

## 1 INTRODUCTION

Unsupervised learning of disentangled latent variables in generative models remains an open research problem, as is an exact mathematical definition of disentangling (Higgins et al., 2018). Intuitively, a disentangled generative model has a one-to-one correspondence between each input dimension of the generator and some interpretable aspect of the data generated.

For VAE-derived models (Kingma & Welling, 2013; Rezende et al., 2014) this is often based around rewarding independence between latent variables. Factor VAE (Kim & Mnih, 2018), $\beta$-TCVAE (Chen et al., 2018) and HFVAE (Esmaeili et al., 2019) have shown that the evidence lower bound can be decomposed to obtain a term capturing the degree of independence between latent variables of the model, the total correlation. By up-weighting this term, we can obtain better disentangled representations under various metrics compared to $\beta$-VAEs (Higgins et al., 2017a).

Disentangled representations, much like PCA or factor analysis, are not only human-interpretable but also offer more informative and robust latent space representations. In addition, information theoretic interpretations of deep learning show that having a disentangled hidden layer within a discriminative deep learning model increases robustness to adversarial attack (Alemi et al., 2017).

Adversarial attacks on deep generative models, more difficult than those on discriminative models (Tabacof et al., 2016; Gondim-Ribeiro et al., 2018; Kos et al., 2018), attempt to fool a model into reconstructing a chosen target image by adding distortions to the original input image. Generally, the most effective attack mode involves making the latent-space representation of the distorted input match that of the target image (Gondim-Ribeiro et al., 2018; Kos et al., 2018). This kind of attack is particularly relevant to applications where the encoder's output is used downstream.

Projections of data from VAEs, disentangled or not, are used for tasks such as: text classification (Xu et al., 2017); discrete optimisation (Kusner et al., 2017); image compression (Theis et al., 2017; Townsend et al., 2019); and as the perceptual part of a reinforcement learning algorithm (Ha & Schmidhuber, 2018; Higgins et al., 2017b), the latter of which uses a disentangled VAE's encoder to improve the robustness of the agent to domain shift.

Here we demonstrate that $\beta$-TCVAEs are significantly more robust to 'latent-space' attack than standard VAEs, and are generally more robust to attacks that act to maximise the evidence lower bound for the adversarial input. The robustness of these disentangled models is highly relevant because of the use-cases for VAEs highlighted above.

However, imposing additional disentangling constraints on a VAE training objective degrades the quality of resulting drawn or reconstructed images (Higgins et al., 2017a; Chen et al., 2018). We sought whether more powerful, expressive models, can help ameliorate this and in doing so built

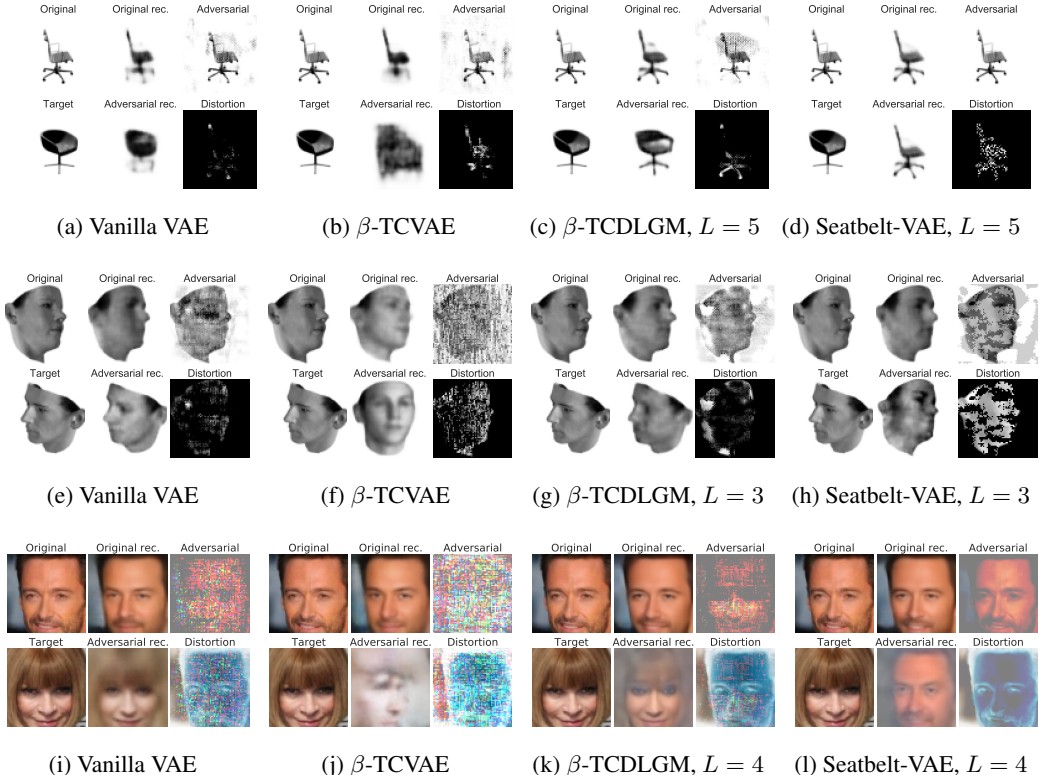

Figure 1: Latent-space adversarial attacks on Chairs, 3D Faces and CelebA for different models, including our proposed *Seatbelt-VAE*. $\beta = 10$ for $\beta$-TCVAE, $\beta$-TCDLGM and Seatbelt-VAE. $L$ is the number of stochastic layers. Clockwise within each plot we show the initial input, its reconstruction, the adversarial input, the adversarial distortion added to make it (shown normalised), the adversarial input's reconstruction, and the target image. Following Tabacof et al. (2016); Gondim-Ribeiro et al. (2018) we attack with different degrees of penalisation on the magnitude of the adversarial distortion; in choosing the distortion to show, we pick the one with the penalisation that resulted in the value of the attack objective just better than the mean. See Section 5 for more details.

a hierarchical disentangled VAE, *Seatbelt-VAE*, drawing on works like Ladder VAEs (Sønderby et al., 2016) and BIVA (Maaløe et al., 2019). We demonstrate that Seatbelt-VAEs are more robust to adversarial attacks than $\beta$-TCVAEs and $\beta$-TCDLGMs (the latter a simple generalisation we make of $\beta$-TC penalisation to hierarchical VAEs). See Figure 1 for a demonstration.

Rather than being concerned with human-interpretable controlled generation by our models, which has been the focus of much research into disentangling, instead we are interested in the robustness afforded by disentangled representations.

Thus our key contributions are:

- A demonstration that $\beta$-TCVAEs are significantly more robust to adversarial attacks via their latents than vanilla VAEs.
- The introduction of Seatbelt-VAE, a hierarchical version of the $\beta$-TCVAE, designed to further increase robustness to various types of adversarial attack, while also giving better perceptual quality of reconstructions even when regularised.

## 2 VARIATIONAL AUTOENCODERS

Variational autoencoders (VAEs) are a deep extension of factor analysis suitable for high-dimensional data like images (Kingma & Welling, 2013; Rezende et al., 2014). They have a joint distribution

over data $x$ and latent variables $z$: $p_\theta(x, z) = p_\theta(x|z)p(z)$ where $p(z) = \mathcal{N}(0, \mathcal{I})$ and $p_\theta(x|z)$ is an appropriate distribution given the form of the data, the parameters of which are represented by deep nets with parameters $\theta$. As exact inference is intractable for this model, in a VAE we perform amortised stochastic variational inference. By introducing an approximate posterior distribution $q_\phi(z|x) = \mathcal{N}(\mu_\phi(x), \Sigma_\phi(x))$, we can perform gradient ascent on the evidence lower bound (ELBO) $\mathcal{L}(x) = -D_{\mathrm{KL}}(q_\phi(z|x)||p_\theta(x, z)) = \mathbb{E}_{q_\phi(z|x)} \log p_\theta(x|z) - D_{\mathrm{KL}}(q_\phi(z|x)||p(z)) \geq \log p(x)$ w.r.t.both $\theta$ and $\phi$ jointly, using the reparameterisation trick to take gradients through Monte Carlo samples from $q_\phi(z|x)$.

## 2.1 Disentangling VAEs

In a $\beta$-VAE (Higgins et al., 2017a), a free parameter $\beta$ multiplies the $D_{\mathrm{KL}}$ term in $\mathcal{L}(x)$ above. This objective $\mathcal{L}_\beta(x)$ remains a lower bound on the evidence.

Decompositions of $\mathcal{L}(x)$ shed light on its meaning. As shown in Hoffman & Johnson (2016); Makhzani et al. (2016); Kim & Mnih (2018); Chen et al. (2018); Esmaeili et al. (2019), one can define the evidence lower bound not per data-point, but instead write it over a dataset $D$ of size $N$, $D = \{x^n\}$, so we have $\mathcal{L}(\theta, \phi, D)$.

Esmaeili et al. (2019) gives a decomposition of this dataset-level evidence lower bound:

$$\mathcal{L}(\theta, \phi, D) = - D_{\mathrm{KL}}(q_\phi(z, x)||p_\theta(x, z)) \tag{1}$$

$$= \mathbb{E}_{q_\phi(z,x)} \Big[ \underbrace{\log \frac{p_\theta(x|z)}{p_\theta(x)}}_{\textcircled{1}} - \underbrace{\log \frac{q_\phi(z|x)}{q_\phi(z)}}_{\textcircled{2}} \Big] - \underbrace{D_{\mathrm{KL}}(q(x)||p_\theta(x))}_{\textcircled{3}} - \underbrace{D_{\mathrm{KL}}(q_\phi(z)||p(z))}_{\textcircled{4}}$$

$$\tag{2}$$

where under the assumption that $p(z)$ factorises we can further decompose $\textcircled{4}$:

$$D_{\mathrm{KL}}(q_\phi(z)||p(z)) = \mathbb{E}_{q_\phi(z)} \underbrace{\Big[ \log \frac{q_\phi(z)}{\prod_j q_\phi(z_j)} \Big]}_{\textcircled{A}} + \sum_j \underbrace{D_{\mathrm{KL}}(q_\phi(z_j)||p(z_j))}_{\textcircled{B}} \tag{3}$$

where $j$ indexes over coordinates in $z$. $q_\phi(z, x) = q_\phi(z|x)q(x)$ and $q(x) := \frac{1}{N} \sum_{n=1}^{N} \delta(x - x^n)$ is the empirical data distribution. $q_\phi(z) := \frac{1}{N} \sum_{n=1}^{N} q_\phi(z|x^n)$ is called the *average encoding distribution* following Hoffman & Johnson (2016).

$\textcircled{A}$ is the total correlation (TC) for $q_\phi(z)$, a generalisation of mutual information to multiple variables (Watanabe, 1960). With this mean-field $p(z)$, Factor and $\beta$-TCVAEs upweight this term, so we have an objective:

$$\mathcal{L}^{\beta\mathrm{TC}}(\theta, \phi, D) = \textcircled{1} + \textcircled{2} + \textcircled{3} + \textcircled{B} + \beta\textcircled{A} \tag{4}$$

Chen et al. (2018) gives a differentiable, stochastic approximation to $\mathbb{E}_{q_\phi(z)} \log q_\phi(z)$, rendering this decomposition simple to use as a training objective using stochastic gradient descent. We also note that $\textcircled{A}$, the total correlation, is also the objective in Independent Component Analysis (Bell & Sejnowski, 1995; Roberts & Everson, 2001).

## 2.2 Hierarchical VAEs

We now have a set of $L$ layers of $z$ variables: $\mathbf{z} = [z^1, z^2, ..., z^L]$. The evidence lower bound for models of this form is:

$$\mathcal{L}^{\mathrm{DLGM}}(\theta, \phi, D) = \mathbb{E}_{q_\phi(\mathbf{z},x)} \log \frac{p_\theta(x, \mathbf{z})}{q_\phi(\mathbf{z}, x)} = \mathbb{E}_{q_\phi(\mathbf{z},x)}[\log p_\theta(x|\mathbf{z})] - \mathbb{E}_{q(x)}[D_{\mathrm{KL}}(q_\phi(\mathbf{z}, x)||p_\theta(\mathbf{z}))]$$

$$\tag{5}$$

The simplest VAE with a hierarchy of conditional stochastic variables in the generative model is the Deep Latent Gaussian Model (DLGM) of Rezende et al. (2014). The forward model factorises as a chain:

$$p_\theta(x, \mathbf{z}) = p_\theta(x|z^1) \prod_{i=1}^{L-1} p_\theta(z^i|z^{i+1})p(z^L) \tag{6}$$

Each $p_\theta(z^i|z^{i+1})$ is a Gaussian distribution with mean and variance parameterised by deep nets. $p(z^L)$ is a unit isotropic Gaussian.

We can understand this additional expressive power as coming from having a richer family of distributions for the likelihood over data $x$ marginalising out all intermediate layers: $p_\theta(x|z^L) = \int \prod_{i=1}^{L-1} \mathrm{d}z^i \, p_\theta(x, \mathbf{z})$ is a non-Gaussian, highly flexible, distribution.

To perform amortised variational inference one introduces a recognition network, which can be any directed acyclic graph where each node, each distribution over each $z^i$, is Gaussian conditioned on its parents. This could be a chain, as in Rezende et al. (2014):

$$q_\phi(\mathbf{z}|x) = \prod_{i=1}^{L-1} q_\phi(z^{i+1}|z^i) q_\phi(z^1|x) \tag{7}$$

Again, marginalising out intermediate $z^i$ layers, we see $q_\phi(z^L|x) = \int \prod_{i=1}^{L-1} \mathrm{d}z^i \, q_\phi(\mathbf{z}|x)$ is a non-Gaussian, highly flexible, distribution.

However, training DLGMs is challenging: the latent variables furthest from the data can fail to learn anything informative (Sønderby et al., 2016; Zhao et al., 2017). Due to the factorisation of $q_\phi(\mathbf{z}|x)$ and $p_\theta(x, \mathbf{z})$ in a DLGM, it is possible for a single-layer VAE to train in isolation within a hierarchical model: each $p_\theta(z^i|z^{i+1})$ distribution can become a fixed distribution not depending on $z^{i+1}$ such that each $D_{\mathrm{KL}}$ divergence present in the objective between corresponding $z^i$ layers can still be driven to a local minima. Zhao et al. (2017) gives a proof of this separation for the case where the model is perfectly trained, i.e. $D_{\mathrm{KL}}(q_\phi(z,x)||p_\theta(x,z)) = 0$.

This is the hierarchical version of the collapse of $z$ units in a single-layer VAE (Burda et al., 2016), but now the collapse is over entire layers $z^i$. It is part of the motivation for the Ladder VAE (Sønderby et al., 2016) and BIVA (Maaløe et al., 2019).

## 3  SEATBELT-VAE: HIERARCHICAL $\beta$-TCVAE WITH SKIP CONNECTIONS

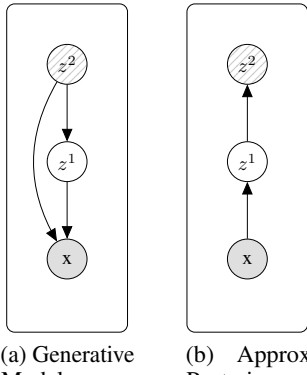

(a) Generative Model

(b) Approx. Posterior

Figure 2: $L = 2$ Seatbelt-VAE. Shaded lines indicate $\beta$-TC factorisation in a given node.

We propose novel hierarchical disentangled VAEs where we aim to disentangle only in the top-most latent variables $z^L$. Following the Factor and $\beta$-TCVAEs we upweight the term of the form of Ⓐ for $z^L$. Empirically we find models of this type are unable to converge when disentangling at the bottom most layer, or when disentangling at each layer. Intuitively, we want to capture high-level disentangled information at the top, but leave lower layers free to learn rich entangled representations. If $p_\theta(x|\mathbf{z}) = p_\theta(x|z^1)$, we obtain the generalisation of $\beta$-TC penalisation to a DLGM and call it $\beta$-TCDLGM. It suffers from the problems of collapse described above.

Inspired by BIVA (Maaløe et al., 2019), we choose instead to condition our likelihood on all $z^i$ layers:

$$p_\theta(x, \mathbf{z}) = p_\theta(x|\mathbf{z}) \prod_{i=1}^{L-1} p_\theta(z^i|z^{i+1}) p(z^L) \tag{8}$$

Combining Eqs (7, 5, 8) and applying $\beta$-TC penalisation to the $D_{\mathrm{KL}}$ term over $z^L$:

$$\mathcal{L}^{\mathrm{SB}}(\theta, \phi, D, \beta) = \mathbb{E}_{q_\phi(\mathbf{z}, x)} \log p_\theta(x|\mathbf{z}) - \mathbb{E}_{q(x)} \log q(x) - \mathbb{E}_{q(x, z^2)}[D_{\mathrm{KL}}(q_\phi(z^1|x)||p_\theta(z^1|z^2))]$$

$$- \sum_{m=2}^{L-1} \mathbb{E}_{q_\phi(z^{m-1}, z^{m+1})}[D_{\mathrm{KL}}(q_\phi(z^m|z^{m-1})||p_\theta(z^m|z^{m+1}))]$$

$$- D_{\mathrm{KL}}(q_\phi(z^L, z^{L-1})||q_\phi(z^L)q_\phi(z^{L-1})) - \beta D_{\mathrm{KL}}(q_\phi(z^L)|| \prod_{j=1} q_\phi(z_j^L))$$

$$- \sum_j D_{\mathrm{KL}}(q_\phi(z_j^L)||p(z_j^L)) \tag{9}$$

$$= \mathbb{E}_{q_\phi(\mathbf{z}, x)} \log p_\theta(x|\mathbf{z}) - \text{\textcircled{c}} \tag{10}$$

where $j$ is indexing over the coordinates in $z^L$. See Appendix for the derivation. We call this model *Seatbelt-VAE*, as with the extra conditional dependencies and nodes we increase the safety of our model to adversarial attacks, to noise, and to decreases in perceptual quality as $\beta$ increases. We find that using *free-bits* regularisation (Kingma et al., 2016) greatly ameliorates the optimisation challenges associated with DLGMs. For $L = 1$ this reduces to a $\beta$-TCVAE, and for $L > 1, \beta = 1$ it produces a DLGM with our augmented likelihood function.

For completeness, note that for $\beta$-TCDLGM:

$$\mathcal{L}^{\beta\mathrm{TCDLGM}}(\theta, \phi, D, \beta) = \mathbb{E}_{q_\phi(\mathbf{z}, x)} \log p_\theta(x|z^1) - \text{\textcircled{c}} \tag{11}$$

### 3.1 MINIBATCH TRAINING

VAEs and derived models are commonly trained using stochastic gradient ascent on the ELBO, on minibatches of the training data. With the ELBO in Eq (9), this would be challenging because of the presence of average encoding distributions, which depend on the entire dataset.

To avoid having to handle large mixture distributions in our objective functions, we derive minibatch estimators that are a simple generalisation to disentangled hierarchical VAEs of the Minibatch Weighted Sampling estimator proposed in Chen et al. (2018) in the context of $\beta$-TCVAEs. See Appendix for further details.

## 4 ROBUSTNESS OF VAEs TO ADVERSARIAL ATTACKS

Most adversarial attack research has focused on discriminative models (Akhtar & Mian, 2018; Gilmer et al., 2018) and recently VAEs have found use in protecting discriminative models against attack (Schott et al., 2019; Ghosh et al., 2019). Currently, two adversarial modes have been proposed for attacking VAEs (Tabacof et al., 2016; Gondim-Ribeiro et al., 2018; Kos et al., 2018). In both attack modes the adversary wants draws from the model to be close to a target image $x^t$, when given a distorted image $x^* = x + d$ as input. When attacking a discriminative model the aim is to manipulate the comparatively low-dimensional output layer of the network, commonly aiming with the attack to diminish or increase only a handful of the output units. However, for a generative model, the attacker is aiming to change a large number of pixel values in the output, changing the content of the reconstruction. Intuitively this is a harder task, and the attacks proposed in the above papers do not always result in adversarial examples that are very close to the initial image in appearance.

The first mode of attack, which we call the *output attack*, aims to reward draws from the decoder conditioned on $z \sim q_\phi(z|x^*)$ that are close to $x^t$ via the ELBO.

For a vanilla VAE, this attack's adversarial objective is:

$$\Delta_{\mathrm{output}}(x, d, x^t; \lambda) = \mathbb{E}_{q_\phi(z|x+d)}[\log p(x^t|z)] - D_{\mathrm{KL}}(q_\phi(z|x+d)||p(z)) + \lambda||d|| \tag{12}$$

The second mode of attack, the *latent attack*, aims to find $x^* = x + d$ such that $q_\phi(z|x^*) \approx q_\phi(z|x^t)$ under some similarity measure $r(\cdot, \cdot)$, which implicitly means that the likelihood $p_\theta(x^t|z)$ is high when conditioned on draws from the posterior of the adversarial example. This attack is important if

one is concerned with using the encoder network of a VAE as part of downstream task. For a single stochastic layer VAE, the latent-space adversarial objective is:

$$\Delta_{\text{latent}}(x, d, x^t; \lambda) = r(q_\phi(z|x + d), q_\phi(z|x^t)) + \lambda||d|| \tag{13}$$

Note that both modes of attack penalise the $L_2$ norm of $d$, prioritising smaller distortions. We denote samples from $q_\phi(z|x + d)$ as $\tilde{z}$.

For Tabacof et al. (2016); Gondim-Ribeiro et al. (2018) $r(\cdot, \cdot)$ is $D_{\text{KL}}(q_\phi(z|x + d)||q_\phi(z|x))$ and for Kos et al. (2018) it is the $L_2$ distance $||\tilde{z} - z^*||_2, \tilde{z} \sim q_\phi(z|x + d), z^* \sim q_\phi(z|x)$ between draws from the corresponding posteriors or $||\mu_\phi(x) - \mu_\phi(x + d)||_2$ between their means. We follow the former papers and use the $D_{\text{KL}}$ formulation. All three papers find that the latent attack mode is as or more effective than the output attack for single layer VAEs both under perceptual evaluation and various proposed metrics (Tabacof et al., 2016; Gondim-Ribeiro et al., 2018; Kos et al., 2018).

For latent attacks, the choice of which layers to attack depends on model architecture. For DLGMs and $\beta$-TCDLGMs the attacker only needs to match at the bottom latent layer as $p_\theta(x|\mathbf{z}) = p_\theta(x|z^1)$, see Eq (7). See Appendix for plots showing how effective this attack is regardless of $\beta$ and $L$.

Even though the decoder is conditioned on all latent layers, one could choose to attack individual layers for Seatbelt-VAE. For example, one could attack just the first layer $z^1$. If one were able to find a perfect latent-space attack in $z^1$, $D_{\text{KL}}(q_\phi(z^1|x + d)||q_\phi(z^1|x^t)) = 0$, then the variational posteriors in higher layers would also be well matched. Attacks that do not perfectly match the target $z^1$ may have their mismatch with the target posterior amplified in higher layers. In Seatbelt-VAE the likelihood over data is conditioned on all $z$ layers, being off-target in these higher layers matters. In the Appendix we show that targeting the top or base layers individually is not as effective as attacking all layers. Hence:

$$\Delta_{\text{latent}}^{\text{DLGM}}(x, d, x^t; \lambda) = D_{\text{KL}}(q_\phi(z^1|x + d), q_\phi(z^1|x^t)) + \lambda||d|| \tag{14}$$

$$\Delta_{\text{latent}}^{\text{SB}}(x, d, x^t; \lambda) = \sum_{i=1}^{L} D_{\text{KL}}(q_\phi(z^i|x + d), q_\phi(z^i|x^t)) + \lambda||d|| \tag{15}$$

## 5 EXPERIMENTS

Here we perform four tranches of experiments. Firstly, we demonstrate that the reconstructions given by Seatbelt-VAEs (and $\beta$-TCDLGMs) degrade much less strongly as $\beta$ is increased than in $\beta$-TCVAEs. Secondly, we perform a variety of adversarial attacks on all models. We demonstrate that increasing $\beta$ makes $\beta$-TCVAEs more robust to adversarial attacks than vanilla VAEs, and that Seatbelt-VAEs are more robust still. Thirdly, we show that these disentangled models are most robust than vanilla VAEs to unstructured noise distorting their inputs, with Seatbelt-VAEs again the most robust. Finally, we study the effect of disentangling on the sparsity of model weights.

We perform these experiments on Chairs (Aubry et al., 2014), 3D faces (Paysan et al., 2009), and CelebA (Liu et al., 2015). Additional results for dSprites (Higgins et al., 2017a) can be found in the Appendix. We used the same encoder and decoder architectures as Chen et al. (2018) for each dataset. For the details of neural network architectures and training, see Appendix and accompanying code. To show the degree to which our models are disentangling, the Appendix also contains the Mutual Information Gap (MIG) (Chen et al., 2018) at the top layer of each model. Though our models obtain high MIG at $z^L$, this does not imply that decoding from latent traversals in $z^L$ will result in the generation of images with human-interpretable factors of variation. This is made abundantly clear in the latent space traversal plots, also shown in the Appendix. As such, we do not believe existing disentangling metrics directly apply to hierarchical models.

### 5.1 ELBO AND RECONSTRUCTION QUALITY: $\beta$-TCVAEs TO SEATBELT-VAEs

We trained $\beta$-TCVAEs, $\beta$-TCDLGMs, and Seatbelt-VAEs for a range of $\beta$ penalisations. In Figure 3 we plot the final ELBO of our trained models, but calculated without the additional $\beta$ penalisation that was applied during training. The ELBO for $\beta$-TCVAE [Eq (4)] declines with $\beta$ much more quickly than Seatbelt VAEs [Eq (10)] or $\beta$-TCDLGMs [Eq (11)]. In the Appendix we also show that increasing $\beta$ reduces $D_{\text{KL}}$ collapse. This is interesting, as it shows that we can increase the $\beta$

penalisation for Seatbelt-VAEs, without a large degradation in the quality of the model as measured by the ELBO.

In Figure 4 we see the effect of depth and disentangling on reconstructions of CelebA. The bottom row, showing the reconstructions from a Seatbelt-VAE with $L = 4$ and $\beta = 20$ clearly maintains facial identity better than those from a $\beta$-TCVAE in the middle row. The effect is clearest for the $3^{rd}$, $4^{th}$ and $7^{th}$ columns, where many of the individuals' finer facial features are lost by the $\beta$-TCVAE but maintained by the Seatbelt-VAE. This fits with the results in Figure 3, and shows that resistance of the quality of the reconstructions of Seatbelt to increasing $\beta$ is visually perceptible as well as measurable.

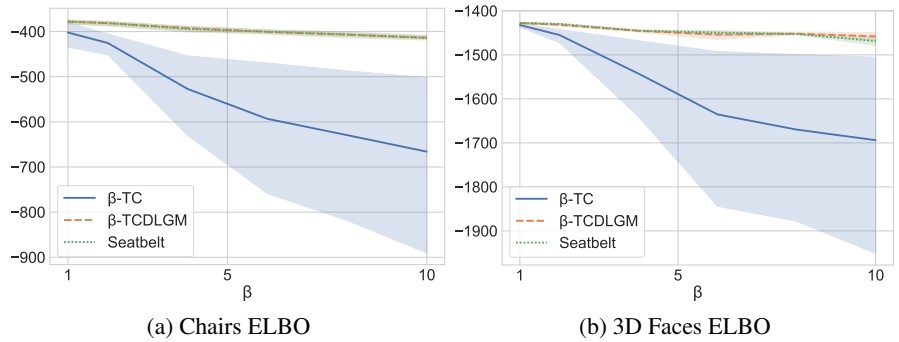

(a) Chairs ELBO  (b) 3D Faces ELBO

Figure 3: Plots showing the effect of varying $\beta$ under various datasets on the ELBO of $\beta$-TCVAEs, $\beta$-TCDLGMs and Seatbelt-VAEs [Eqs (4), (11) and (10) respectively]. Shading corresponds to the 95% CI over variation due to variation of $\|z\|$ and $L$.

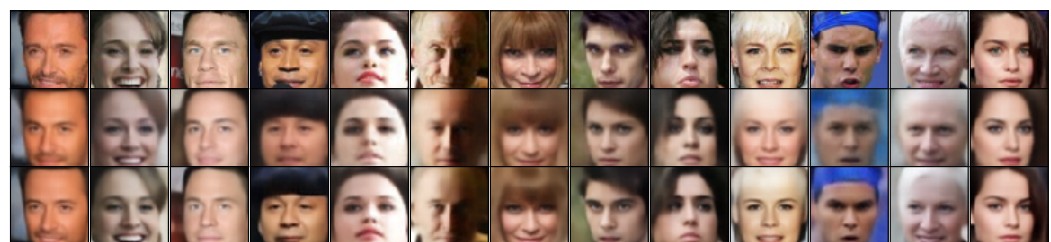

Figure 4: Top row shows CelebA input data. Below are reconstructions from $\beta$-TCVAE, $\beta = 20$ and then Seatbelt VAE, $L = 4$, $\beta = 20$.

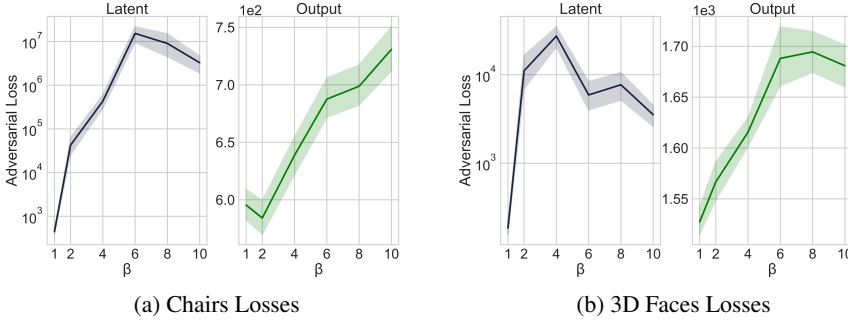

(a) Chairs Losses  (b) 3D Faces Losses

Figure 5: $\Delta_{\text{latent/output}}$ for (a) Chairs (b) 3D Faces, for $\beta$-TCVAE for different $\beta$ values. Shading corresponds to the 95% CI over variation due to our stable of images and our values of $\|z\|$ and $\lambda$.

## 5.2 ADVERSARIAL ATTACK

We apply attacks minimising each of $\Delta_{\text{output}}$ and $\Delta_{\text{latent}}$ on: vanilla VAEs, $\beta$-TCVAEs, $\beta$-TCDLGMs and Seatbelt-VAEs; trained on: Chairs (Aubry et al., 2014), 3D faces (Paysan et al., 2009), and CelebA (Liu et al., 2015); for a range of $\beta$, $L$ and $\lambda$ values.

We randomly sampled 10 input-target pairs for each dataset. As in Tabacof et al. (2016); Gondim-Ribeiro et al. (2018), for each pair of images used $\lambda$ takes 50 geometrically-distributed values from $2^{-20}$ to $2^{20}$. Thus each model undergoes 500 attacks for each attack mode. Like Tabacof et al. (2016); Gondim-Ribeiro et al. (2018), we used L-BFGS-B for gradient descent (Byrd et al., 1995),

We prefer to avoid classifier based metrics (Kos et al., 2018) as in general we think that such analysis can be hard to interpret given the many available choices of classifier. Instead, we evaluate the effectiveness of adversarial attacks from the values reached by $-\log p_\theta(x^t|\tilde{z})$, by the attack objectives $\{\Delta_{\text{output}}, \Delta_{\text{latent}}\}$ and by visually appraising the adversarial input $(x + d)$ and the adversarial reconstruction. Note that higher values of $-\log p_\theta(x^t|\tilde{z}), \Delta_{\text{output}}, \Delta_{\text{latent}}$ indicate less effective attacks.

Figure 1 shows latent space attacks and demonstrates that they are less effective on disentangled models. As in Gondim-Ribeiro et al. (2018), we are showing the attack for the $\lambda$ that gives us an attack objective just better than the average objective over all attacks tried. Note that for Seatbelt-VAEs, for high values of $\beta$ and $L$ latent attacks often result in the outputs from adversarial attack resembling the original inputs. See Appendix for more examples of the attacks for $\{\Delta_{\text{latent}}, \Delta_{\text{output}}\}$ for the models trained on dSprites (a toy dataset for disentangling), Chairs, 3D Faces and CelebA; each over a range of values for $\beta$, $L$, and $\lambda$. Note that we rarely observe perceptually effective output attacks regardless of model or settings, though vanilla VAEs are the most susceptible.

One might expect that adversarial attacks targeting a single factor of the data would be easier for the attacker. However, we find that disentangled models protect effectively against these attacks as well. See the Appendix for plots showing an attacker attempting to rotate a dSprites heart.

Figure 5 quantitatively shows that $\beta$-TCVAEs become harder to attack as $\beta$ increases. The values of $\Delta_{\text{latent}}$ for $\beta$-TCVAEs are $\approx 10^3$ times higher than for a standard VAE on Chairs, and still greater than a factor of 10 for 3D faces. $\Delta_{\text{output}}$ attack is also less effective, by a smaller factor $\approx 1.2$.

Figure 6 shows $-\log p_\theta(x^t|\tilde{z}_{\text{latent/output}})$ and Figure 7 shows $\Delta_{\text{latent/output}}$ over a range of datasets for $\beta$-TCDLGMs and Seatbelt-VAEs, varying $L$ and $\beta$. Larger values of these metrics correspond to less successful adversarial attacks. Generally, $\beta$-TCDLGMs are very sensitive to latent attack, as we expect. Like $\beta$-TCVAEs, Seatbelt-VAEs offer significant protection to latent attacks, and somewhat increased protection to output attacks compared to vanilla VAEs. For Seatbelt-VAEs, as we go to the largest values of $\beta$ and $L$ for both Chairs and 3D Faces, $\Delta_{\text{latent}}$ grows by a factor of $\approx 10^7$.

The bottom rows of Figures 6 & 7 (c) (d) have $L = 1$, and thus correspond to $\beta$-TCVAEs. They contain relatively low values of the adversarial objectives compared to $L > 1$. Similarly the first column, corresponding to $\beta$=1 models, contains relatively low values. These results tell us that depth and disentangling together offer the most effective protection from the adversarial attacks studied.

In the Appendix we also calculate the $L_2$ distance between target images and adversarial outputs and show that the loss of effectiveness of adversarial attacks is not due to the degradation of reconstruction quality from increasing $\beta$. By these metrics too Seatbelt-VAEs outperform other models.

## 5.3 ROBUSTNESS TO NOISE

In addition to studying the robustness of these models to highly structured distortion, we can also consider robustness to random noise. We add $\epsilon \sim \mathcal{N}(0, \mathcal{I})$ to the datasets, which are scaled to $-1 \leq x \leq 1$, and then evaluate $\mathbb{E}_{q_\phi(z|x+\epsilon)} p_\theta(x|z^*)$, where $z^*$ corresponds to the encoder embedding of $x + \epsilon$ and $x$ is the original (non-noisy) data. See Figure 8 for smoothed histogram plots of this for different models for different degrees of $\beta$. Both $\beta$-TC and Seatbelt-VAEs are effectively denoising autoencoders. They become more robust to noise with increasing $\beta$, while $\beta$-TCDLGMs get worse. See Appendix for plots showing the robustness of these models to smaller magnitude noise.

Some of the robustness of disentangled models to adversarial attacks may be conferred by their robustness to random perturbations of their inputs.

## 5.4 TOTAL CORRELATION PENALISATION AS REGULARISATION

In the auto-encoder view of these models, the $D_{\text{KL}}$ terms in $\mathcal{L}(\theta, \phi, D)$ are associated with a form of regularisation of the model (Doersch, 2016). Recent work shows that for linear autoencoders,

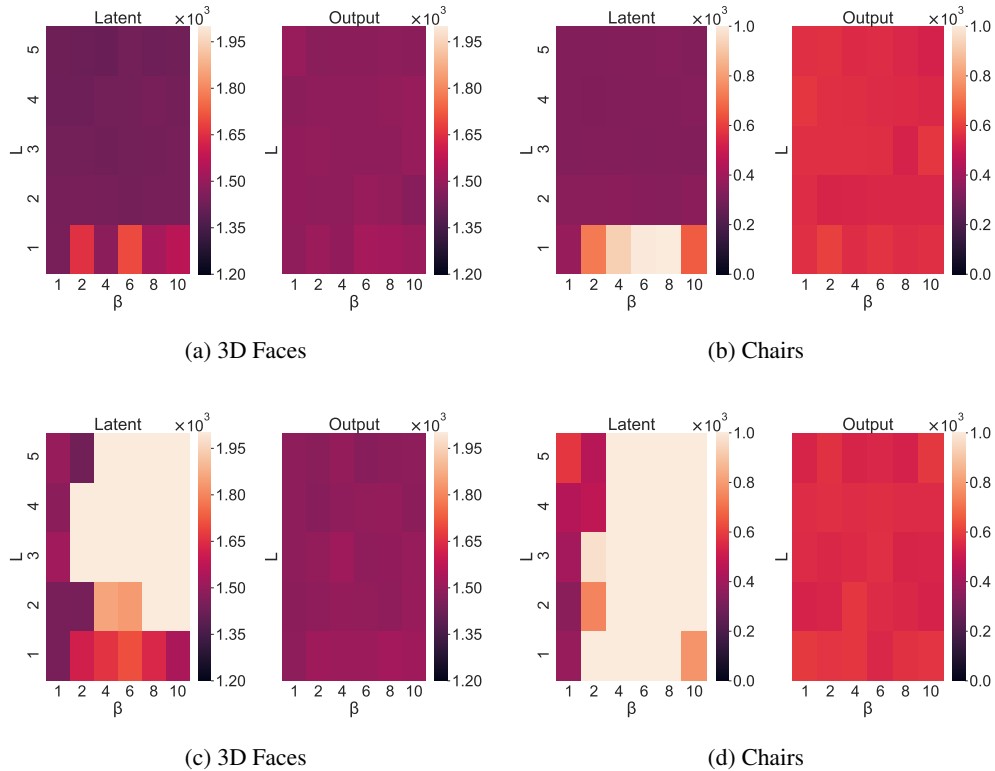

Figure 6: $-\log p_\theta(x^t|\tilde{z})$ for (a) (b) $\beta$-TCDLGMs and (c) (d) Seatbelt-VAEs for Chairs and 3D Faces; over $\beta$ and $L$ (total number of stochastic layers) values and for *latent* and *output* attacks. Larger values of $-\log p_\theta(x^t|\tilde{z})$ correspond to less successful adversarial attacks.

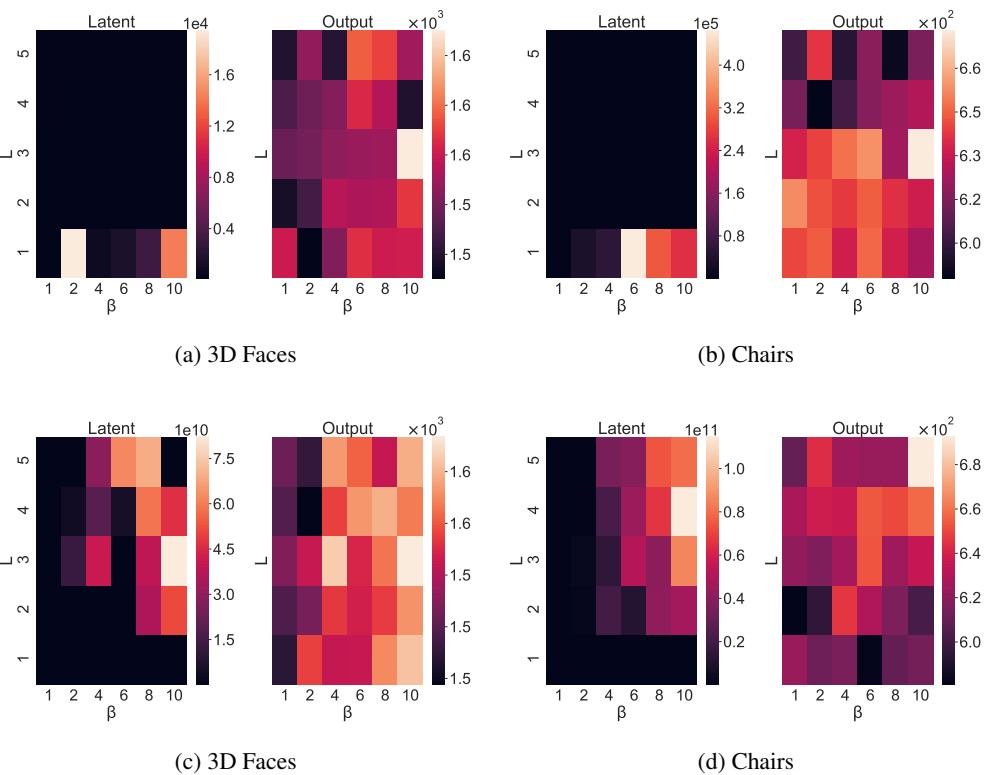

Figure 7: $\{\Delta_{\text{latent}}, \Delta_{\text{output}}\}$ for (a) (b) $\beta$-TCDLGMs and (c) (d) Seatbelt-VAEs for Chairs and 3D Faces; under varying $\beta$ and $L$ (total number of stochastic layers) values.

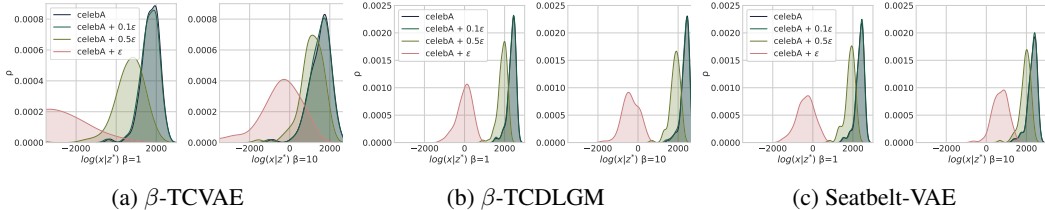

(a) $\beta$-TCVAE      (b) $\beta$-TCDLGM      (c) Seatbelt-VAE

Figure 8: Robustness of $\log p_\theta(x|z)$ to Gaussian noise $\epsilon \sim \mathcal{N}(0,1)$ scaled by different magnitudes and added to $x$ on CelebA; for $\beta$-TCVAE, $\beta$-TCDLGM, Seatbelt-VAE; $\beta = 0, 10$ Best viewed digitally.

Table 1: Relative change of the $L_2$ of Encoders and Decoders by dataset for $\beta$-TCVAE and Seatbelt-VAE ($L = 4$) when increasing $\beta$ from 1 to 10.

|  |  | Chairs | 3D Faces | CelebA |
|---|---|---|---|---|
|  |  | $\beta : 1 \to 10$ | $\beta : 1 \to 10$ | $\beta : 1 \to 10$ |
| Encoder | $\beta$-TCVAE | +5.0% | +19.5% | +73.7% |
|  | Seatbelt-VAE, $L = 4$ | +1.0% | +2.7% | +40.2% |
| Decoder | $\beta$-TCVAE | -19.4% | -15.0% | -6.8% |
|  | Seatbelt-VAE, $L = 4$ | -7.6% | -6.0% | -11.4% |

$L_2$ regularisation of the weights corresponds to orthogonality of the latent projections (Kunin et al., 2019). For deep models we expect that disentangling is associated with regularised decoders and more complex encoders. The decoder receives a simpler representation, but building this representation requires more calculation. Here we measure the $L_2$ norm of the weights of our networks as a function of $\beta$, shown in Table 1. See Appendix for results for $\beta$-TCDLGM.

As we increase $\beta$ for $\beta$-TCVAEs and Seatbelt-VAEs for Chairs, 3D Faces, and CelebA the $L_2$ norm increases for the encoder and decreases for the decoder. A more complex encoder is more difficult to match in the latent space and regularised decoders may be contributing to the denoising properties seen in Figure 8. That the changes are generally greater for $\beta$-TCVAE than Seatbelt-VAE makes sense, as the encoder and decoder of the former interact directly with the disentangled representation. For the latter the decoder receives inputs from all $z^i$, of varying degrees of disentanglement.

## 6 CONCLUSION

We have presented the increases in robustness to adversarial attack afforded by $\beta$-TCVAEs. This increase in robustness is strongest for attacks via the latent space. While disentangled models are often motivated by their ability to provide interpretable conditional generation, many use cases for VAEs centre on the learnt latent representation of data. Given the use of these representations as inputs for other tasks, the latent attack mode is the most important to protect against.

Recent work by Shamir et al. (2019) gives a constructive proof for the existence of adversarial inputs for deep neural network classifiers with small Hamming distances. The proof holds with deterministic defence procedures that work as additional deterministic layers of the networks, and in the presence of adversarial training (Szegedy et al., 2014; Ganin et al., 2016; Tramèr et al., 2018; Shaham et al., 2018). Shamir et al. (2019) thus give a theoretical grounding for using stochastic methods to defend against adversarial inputs. As VAEs are already used to defend deep net classifiers (Schott et al., 2019; Ghosh et al., 2019), more robust VAEs, like $\beta$-TCVAEs, could find use in this area.

We introduce Seatbelt-VAE, a particular hierarchical VAE disentangled on the top-most layer with skip connections down to the decoder. This model further increases robustness to adversarial attacks, while also increasing the quality of reconstructions. The performance of our model under adversarial attack to robustness is mirrored in robustness to uncorrelated noise: these models are effective denoising autoencoders as well. We hope this work stimulates further interest in defending and attacking VAEs.

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

# Supplementary Material for: Disentangling Improves VAEs' Robustness to Adversarial Attacks

**Anonymous authors**

## Contents

## A DERIVATION OF ELBO FOR SEATBELT-VAES

Start with Eq (5) cf. Eq (7) in the main paper. The likelihood is conditioned on all $z$ layers: $p_\theta(x|\mathbf{z})$.

$$\mathcal{L}(\theta, \phi, D) = \mathbb{E}_{q_\phi(\mathbf{z},x)} \log \frac{p_\theta(x, \mathbf{z})}{\log q_\phi(\mathbf{z}, x)} = \mathbb{E}_{q_\phi(\mathbf{z},x)}[\log p_\theta(x|\mathbf{z})] - \mathbb{E}_{q(x)}[D_{\mathrm{KL}}(q_\phi(\mathbf{z}, x)||p_\theta(\mathbf{z}))]$$

(A.1)

$$= \mathbb{E}_{q(\mathbf{z},x)} \log p_\theta(x|\mathbf{z}) - \mathbb{E}_{q(x)} \log q(x) + \mathbb{E}_{q(\mathbf{z},x)} \log \frac{p_\theta(\mathbf{z})}{q(\mathbf{z}|x)}$$ (A.2)

$$= \mathbb{E}_{q(\mathbf{z},x)} \log p_\theta(x|\mathbf{z}) + \mathcal{H}(q(x))$$ (A.3)

$$+ \underbrace{\int \mathrm{d}x\, \mathrm{d}z^1 \prod_{i=2}^{L}(\mathrm{d}z^i q_\phi(z^i|z^{i-1})) q_\phi(z^1|x) q(x) \log \frac{p(z^L)\prod_{k=1}^{L-1} p_\theta(z^k|z^{k+1})}{q_\phi(z^1|x)\prod_{m=1}^{L-1} q_\phi(z^{m+1}|z^m)}}_{\text{W}}$$

$$\text{W} = \underbrace{\int \mathrm{d}x \prod_{i=1}^{L}(\mathrm{d}z^i) q_\phi(\mathbf{z}|x) q(x) \log \frac{p(z^L)}{q_\phi(z^L|z^{L-1})}}_{\text{T}}$$

$$+ \underbrace{\int \mathrm{d}x \prod_{i=1}^{L}(\mathrm{d}z^i) q_\phi(\mathbf{z}|x) q(x) \log \frac{\prod_{k=1}^{L-1} p_\theta(z^k|z^{k+1})}{q_\phi(z^1|x)\prod_{m=1}^{L-2} q_\phi(z^{m+1}|z^m)}}_{\text{R}}$$ (A.4)

$$\text{T} = -\mathbb{E}_{q_\phi(z^{L-1})} D_{\mathrm{KL}}(q_\phi(z^L|z^{L-1})||p(z^L))$$ (A.5)

$$\text{R} = \underbrace{\int \mathrm{d}x \prod_{i=1}^{L}(\mathrm{d}z^i) q_\phi(\mathbf{z}|x) q(x) \log \frac{p_\theta(z^1|z^2)}{q_\phi(z^1|x)}}_{\text{R}_a}$$

$$+ \underbrace{\sum_{m=2}^{L-1} \int \mathrm{d}x \prod_{i=1}^{L}(\mathrm{d}z^i) q_\phi(\mathbf{z}|x) q(x) \log \frac{p_\theta(z^m|z^{m+1})}{q_\phi(z^m|z^{m-1})}}_{\text{R}_b}$$ (A.6)

$$\text{R}_a = -\mathbb{E}_{q_\phi(z^2,x)} D_{\mathrm{KL}}(q_\phi(z^1|x)||p_\theta(z^1|z^2))$$ (A.7)

$$\text{R}_b = \sum_{m=2}^{L-1} \int \mathrm{d}x \prod_{i=1}^{L}(\mathrm{d}z^i) q_\phi(z^1|x) q(x) \prod_{k=1, k \neq m}^{L-1}(q_\phi(z^{k+1}|z^k)) q_\phi(z^m|z^{m-1}) \log \frac{p_\theta(z^m|z^{m+1})}{q_\phi(z^m|z^{m-1})}$$

(A.8)

$$= -\sum_{m=2}^{L-1} \int \mathrm{d}x \prod_{i=1}^{L}(\mathrm{d}z^i) q_\phi(z^1|x) q(x) \prod_{k=1, k \neq m}^{L-1}(q_\phi(z^{k+1}|z^k)) D_{\mathrm{KL}}(q_\phi(z^m|z^{m-1})||p_\theta(z^m|z^{m+1}))$$

(A.9)

$$= -\sum_{m=2}^{L-1} \mathbb{E}_{q_\phi(z^{m+1}, z^{m-1})} D_{\mathrm{KL}}(q_\phi(z^m|z^{m-1})||p_\theta(z^m|z^{m+1}))$$ (A.10)

Now we have:

$$\mathcal{L}(\theta, \phi, D) = \mathbb{E}_{q(\mathbf{z},x)} \log p_\theta(x|\mathbf{z}) + \mathcal{H}(q(x)) + \textcircled{R_a} + \textcircled{R_b} + \textcircled{T} \tag{A.11}$$

$$\tag{A.12}$$

Apply $\beta$TC decomposition to $\textcircled{T}$ as in Chen et al. (2018). $j$ indexes over units in $z^L$.

$$\textcircled{T} = -\mathbb{E}_{q_\phi(z^{L-1})} \left[ \mathbb{E}_{q_\phi(z^L|z^{L-1})}[\log q_\phi(z^L|z^{L-1}) - \log p(z^L) + \log q_\phi(z^L)\right.$$
$$\left. - \log q_\phi(z^L) + \log \prod_j q_\phi(z_j^L) - \log \prod_j q_\phi(z_j^L)]] \tag{A.13}$$

$$= -\mathbb{E}_{q_\phi(z^L,z^{L-1})}[\log \frac{q_\phi(z^L|z^{L-1})}{q_\phi(z^L)}] - \mathbb{E}_{q_\phi(z^L)}[\log \frac{q_\phi(z^L)}{\prod_j q_\phi(z_j^L)}] - \mathbb{E}_{q_\phi(z^L)}[\log \frac{\prod_j q_\phi(z_j^L)}{p(z^L)}] \tag{A.14}$$

$$= -\mathbb{E}_{q_\phi(z^L,z^{L-1})}[\log \frac{q_\phi(z^L|z^{L-1})q_\phi(z^{L-1})}{q_\phi(z^L)q_\phi(z^{L-1})}] - \mathbb{E}_{q_\phi(z^L)}[\log \frac{q_\phi(z^L)}{\prod_j q_\phi(z_j^L)}] - \sum_j \mathbb{E}_{q_\phi(z^L)}[\log \frac{q_\phi(z_j^L)}{p(z_j^L)}] \tag{A.15}$$

$$= \underbrace{-D_{\mathrm{KL}}(q_\phi(z^L, z^{L-1})||q_\phi(z^L)q_\phi(z^{L-1}))}_{\textcircled{T_a}} \underbrace{-D_{\mathrm{KL}}(q_\phi(z^L)||\prod_j q_\phi(z_j^L))}_{\textcircled{T_b}} \underbrace{-\sum_j D_{\mathrm{KL}}(q_\phi(z_j^L)||p(z_j^L))}_{\textcircled{T_c}} \tag{A.16}$$

Where we have used $p(z^L) = \prod_j p(z_j^L)$ for our chosen generative model. As in Chen et al. (2018), we choose to weight $\textcircled{T_b}$, the total correlation for $q_\phi(z^L)$, by a prefactor $\beta$.

$$\mathcal{L}^{\mathrm{SB}}(\theta, \phi, D, \beta) = \mathbb{E}_{q(\mathbf{z},x)} \log p_\theta(x|\mathbf{z}) + \mathcal{H}(q(x)) + \textcircled{R_a} + \textcircled{R_b} + \textcircled{T_a} + \beta\textcircled{T_b} + \textcircled{T_c} \tag{A.17}$$

Giving us the ELBO for Seatbelt-VAEs, Eq (10).

# B   MINIBATCH WEIGHTED SAMPLING FOR $z^i$

As in Chen et al. (2018), applying $\beta$-TC decomposition requires us to calculate terms of the form:

$$\mathbb{E}_{q_\phi(z^i)} \log q_\phi(z^i) \tag{B.1}$$

The $i = 1$ case is covered in the appendix of Chen et al. (2018). First we will repeat the argument for $i = 1$ as made in Chen et al. (2018), but in our notation, and then we cover the case $i > 1$ for models with factorisation of $q_\phi(\mathbf{z}|x)$ as in Eq 7 in the main paper.

## B.1   MWS FOR $q_\phi(z^1)$: $\beta$-TCVAEs

Introduce $\mathcal{B}_M = \{x_1, x_2, ..., x_M\}$, a minibatch of datapoints drawn uniformly iid from $q(x) = 1/N \sum_{n=1}^N \delta(x - x_n)$. For for any minibatch we have $p(\mathcal{B}_M) = \frac{1}{N}^M$. Chen et al. (2018) introduce $r(\mathcal{B}_M|x)$, the probability of a sampled minibatch given that one member is $x$ and the remaining $M - 1$ points are sampled iid from $q(x)$, so $r(\mathcal{B}_M|x) = \frac{1}{N}^{M-1}$.

$$\mathbb{E}_{q_\phi(z^i)} \log q_\phi(z^i) = \mathbb{E}_{q_\phi(z^1,x)}[\log \mathbb{E}_{q(x)}[q_\phi(z^1|x)]] \tag{B.2}$$

$$= \mathbb{E}_{q_\phi(z^1,x)}[\log \mathbb{E}_{p(\mathcal{B}_M)}[\frac{1}{M} \sum_{m=1}^M q_\phi(z^1|x_m)]] \tag{B.3}$$

$$\geq \mathbb{E}_{q_\phi(z^1,x)}[\log \mathbb{E}_{r(\mathcal{B}_M|x)}[\frac{p(\mathcal{B}_M)}{r(\mathcal{B}_M|x)} \frac{1}{M} \sum_{m=1}^M q_\phi(z^1|x_m)]] \tag{B.4}$$

$$= \mathbb{E}_{q_\phi(z^1,x)}[\log \mathbb{E}_{r(\mathcal{B}_M|x)}[\frac{1}{NM} \sum_{m=1}^M q_\phi(z^1|x_m)]] \tag{B.5}$$

$$\tag{B.6}$$

So then during training, one samples a minibatch $\{x_1, x_2, ..., x_M\}$ and can estimate $\mathbb{E}_{q_\phi(z^1)} \log q_\phi(z^1)$ as:

$$\mathbb{E}_{q_\phi(z^1)} \log q_\phi(z^1) \approx \frac{1}{M} \sum_{i=1}^M [\log \sum_{j=1}^M q_\phi(z_i^1|x_j) - \log NM] \tag{B.7}$$

and $z_i^1$ is a sample from $q_\phi(z^1|x_i)$.

## B.2   MINIBATCH WEIGHTED SAMPLING FOR $q_\phi(z^i), i > 1$: $\beta$-TCGLGMs AND SEATBELT-VAEs

Here we have that $q(\mathbf{z}, x) = \prod_{l=2}^L [q_\phi(z^l|z^{l-1})]q_\phi(z^1|x)q(x)$. Now instead of having a minibatch of datapoints, we have a minibatch of draws of $z^{i-1}$: $\mathcal{B}_M^{i-1} = \{z_1^{i-1}, z_2^{i-1}, ..., z_M^{i-1}\}$. Each member of which is the result of sequentially sampling along a chain, starting with some particular datapoint $x_m \sim q(x)$.

For $i > 2$, members of $\mathcal{B}_M^{i-1}$ are drawn:

$$z_j^{i-1} \sim q_\phi(z^{i-1}|z_j^{i-2}) \tag{B.8}$$

and for $i = 2$:

$$z_j^1 \sim q_\phi(z^1|x_j) \tag{B.9}$$

Thus each member of this batch $\mathcal{B}_M^{i-1}$ is the descendant of a particular datapoint that was sampled in an iid minibatch $\mathcal{B}_M$ as defined above. We similarly define $r(\mathcal{B}_M^{i-1}|z^{i-1})$ as the probability of selecting a particular minibatch $\mathcal{B}_M^{i-1}$ of these values out from our set $\{z_n^{i-1}\}$ (of cardinality $N$) given that we have selected into our minibatch one particular $z^{i-1}$ from these $N$ values. Like above, $r(\mathcal{B}_M^{i-1}|z^{i-1}) = \frac{1}{N}^{M-1}$

Now we can consider $\mathbb{E}_{q_\phi(z^i)} \log q_\phi(z^i)$ for $i > 1$:

$$\mathbb{E}_{q_\phi(z^i)} \log q_\phi(z^i) = \mathbb{E}_{q_\phi(z^i, z^{i-1})}[\log \mathbb{E}_{q_\phi(z^{i-1})}[q_\phi(z^i|z^{i-1})]] \tag{B.10}$$

$$= \mathbb{E}_{q_\phi(z^i, z^{i-1})}[\log \mathbb{E}_{p(\mathcal{B}_M^{i-1})}[\frac{1}{M} \sum_{m=1}^M q_\phi(z^i|z_m^{i-1})]] \tag{B.11}$$

$$\geq \mathbb{E}_{q_\phi(z^i, z^{i-1})}[\log \mathbb{E}_{r(\mathcal{B}_M^{i-1}|z^{i-1})}[\frac{p(\mathcal{B}_M)}{r(\mathcal{B}_M|x)} \frac{1}{M} \sum_{m=1}^M q_\phi(z^i|z_m^{i-1})]] \tag{B.12}$$

$$= \mathbb{E}_{q_\phi(z^i, z^{i-1})}[\log \mathbb{E}_{r(\mathcal{B}_M^{i-1}|z^{i-1})}[\frac{1}{NM} \sum_{m=1}^M q_\phi(z^i|z_m^{i-1})]] \tag{B.13}$$

Where we have followed the same steps as in the previous subsection.

During training, one samples a minibatch $\{z_1^{i-1}, z_2^{i-1}, ..., z_M^{i-1}\}$, where each is constructed by sampling ancestrally. Then one can estimate $\mathbb{E}_{q_\phi(z^i)} \log q_\phi(z^i)$ as:

$$\mathbb{E}_{q_\phi(z^i)} \log q_\phi(z^i) \approx \frac{1}{M} \sum_{k=1}^M [\log \sum_{j=1}^M q_\phi(z_k^i|z_j^{i-1}) - \log NM] \tag{B.14}$$

and $z_k^i$ is a sample from $q_\phi(z^i|z_k^{i-1})$. In our model we only need terms of this form for $i = L$, so we have:

$$\mathbb{E}_{q_\phi(z^L)} \log q_\phi(z^L) \approx \frac{1}{M} \sum_{k=1}^M [\log \sum_{j=1}^M q_\phi(z_k^L|z_j^{L-1}) - \log NM] \tag{B.15}$$

and $z_k^L$ is a sample from $q_\phi(z^L|z_k^{L-1})$.

## C  IMPLEMENTATION DETAILS

All runs were done on the Azure cloud system on NC6 GPU machines.

### C.1  ENCODER AND DECODER ARCHITECTURES

We used the same architectures as Chen et al. 2018. See file `src/stochastic_layers/encoders.py` and `src/stochastic_layers/decoders.py` in the accompanying repository.

For $\beta$-TCVAE the range of $||z||$ values used was $\{4, 6, 8, 16, 32, 64, 128\}$. For $\beta$-TCDLGMs and Seatbelt-VAEs the number of units in each layer $z^i$ decreases sequentially. There is a list $z_s$izes for each dataset, and for a model of $L$ layers that the last $L$ entries to give $||z^i||, i \in \{1, ..., L\}$.

$$\{||z||\}^{\text{dSprites}} = \{96, 48, 24, 12, 6\} \tag{C.1}$$

$$\{||z||\}^{\text{Chairs}} = \{96, 48, 24, 12, 6\} \tag{C.2}$$

$$\{||z||\}^{\text{3DFaces}} = \{96, 48, 24, 12, 6\} \tag{C.3}$$

$$\{||z||\}^{\text{CelebA}} = \{256, 128, 64, 32\} \tag{C.4}$$

$$\tag{C.5}$$

For $\beta$-TCDLGMs and Seatbelt-VAEs we also have the mappings $q_\phi(z^{i+1}|z^i)$ and $p_\theta(z^i|z^{i+1})$. These are amortised as MLPs with 2 hidden layers with batchnorm and Leaky-ReLU activation. The dimensionality of the hidden layers also decreases as a function of layer index $i$:

$$||h||(q_\phi(z^{i+1}|z^i)) = h_{\text{sizes}}[i] \tag{C.6}$$

$$||h||(p_\theta(z^i|z^{i+1})) = h_{\text{sizes}}[i] \tag{C.7}$$

$$h_{\text{sizes}} = [1024, 512, 256, 128, 64] \tag{C.8}$$

To train the model we used ADAM (Kingma & Ba, 2015) with default parameters and a learning rate of 0.001. All data was preprocessed to fall on the interval -1 to 1. CelebA and Chairs were both downsampled and cropped as in (Chen et al., 2018) and (Kulkarni et al., 2015) respectively.

## D  $L_2$ NORM OF $\beta$-TCDLGM

Table D.1: $L_2$ of Encoders and Decoders by dataset for $\beta$-TCDLGM ($L = 4$) showing the proportional change from increasing $\beta$ from 1 to 10.

|  |  | Chairs | 3D Faces | CelebA |
|---|---|---|---|---|
|  |  | $\beta : 1 \to 10$ | $\beta : 1 \to 10$ | $\beta : 1 \to 10$ |
| Encoder | $\beta$-TCDLGM | +3.1% | +5.7% | +1.22% |
| Decoder | $\beta$-TCDLGM | -4.8% | -3.5% | +7.3% |

# E ROBUSTNESS TO NOISE

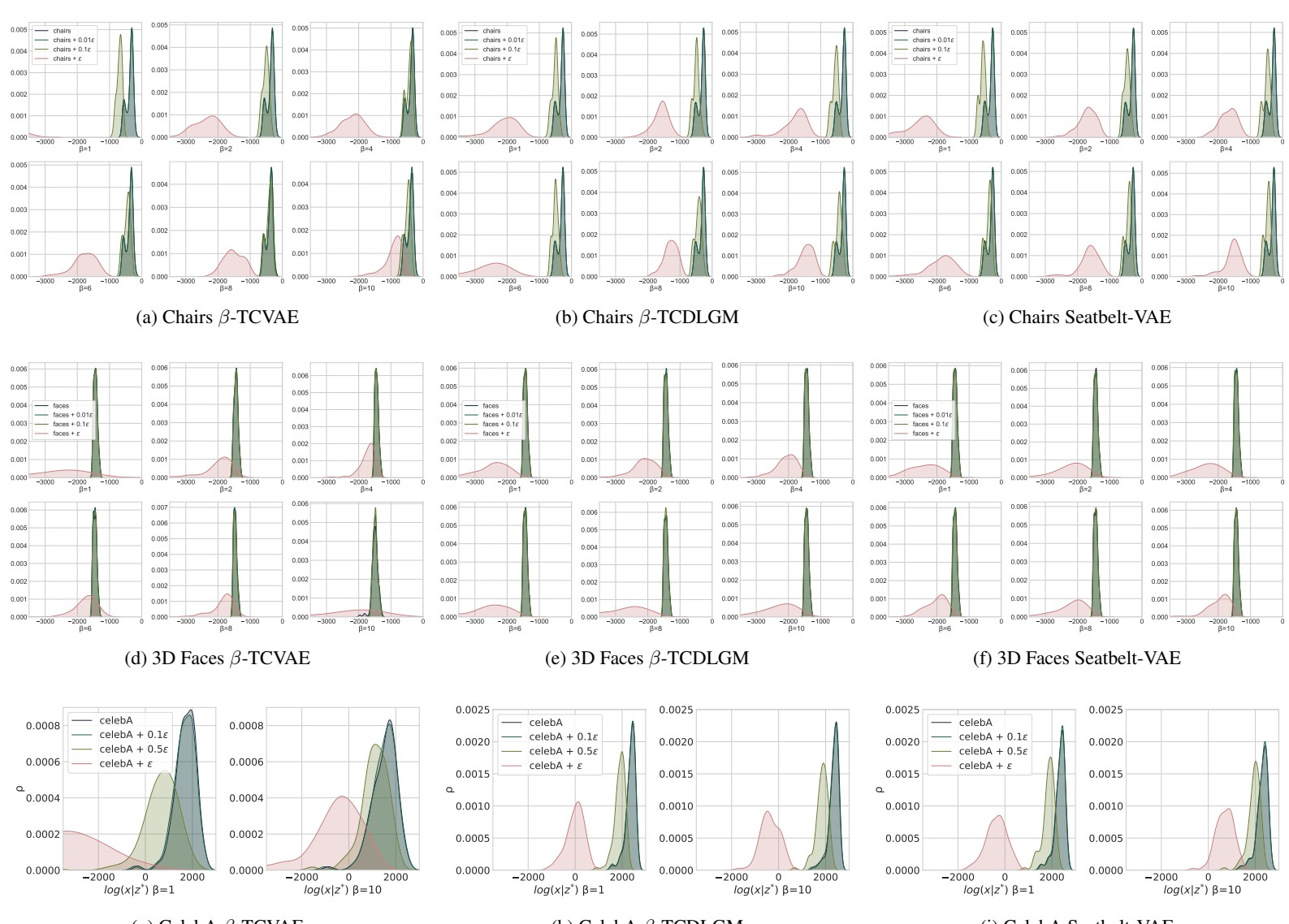

(a) Chairs $\beta$-TCVAE

(b) Chairs $\beta$-TCDLGM

(c) Chairs Seatbelt-VAE

(d) 3D Faces $\beta$-TCVAE

(e) 3D Faces $\beta$-TCDLGM

(f) 3D Faces Seatbelt-VAE

(g) CelebA $\beta$-TCVAE

(h) CelebA $\beta$-TCDLGM

(i) CelebA Seatbelt-VAE

Figure E.1: Robustness of $\log p_\theta(x|z)$ to Gaussian noise $\epsilon \sim \mathcal{N}(0,1)$ scaled by different magnitudes and added to $x$: on Chairs, 3D Faces and CelebA by row; for $\beta$-TCVAE, $\beta$-TCDLGM, Seatbelt-VAE by column. Within each plot a range of $\beta$ values are shown. Best viewed digitally

# F    ACTIVATION OF $z$

## F.1    $\beta$-TCVAEs

For this subsection. within each subplot we order the units of $z$ by the values of their $D_{\mathrm{KL}}$ divergence.

**dSprites**

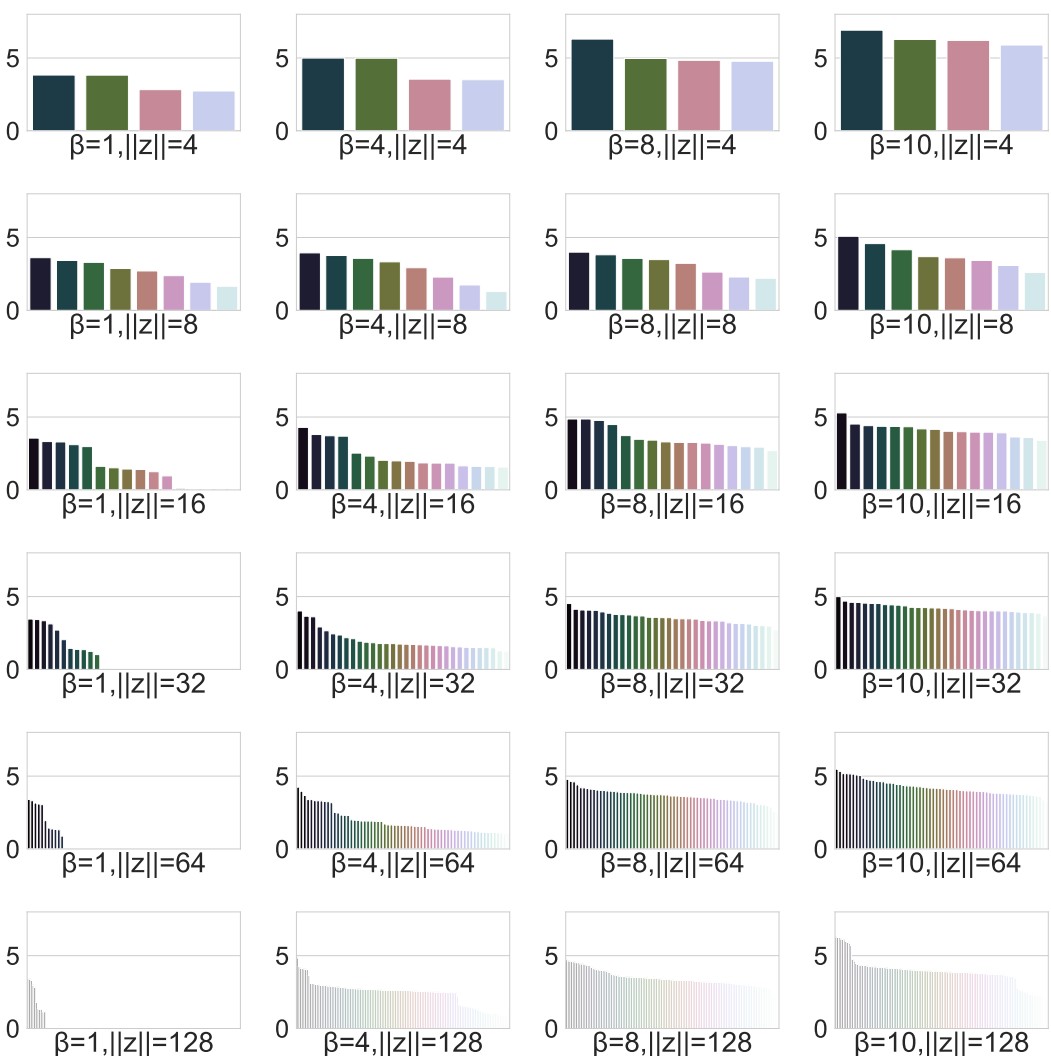

Figure F.2: $\mathbb{E}_{q(x)} D_{\mathrm{KL}}(q_\phi(z_j|x)||p(z_j))$ over dSprites for $\beta$-TCVAE over values of $||z||$ and $\beta$. Best viewed digitally.

**Chairs**

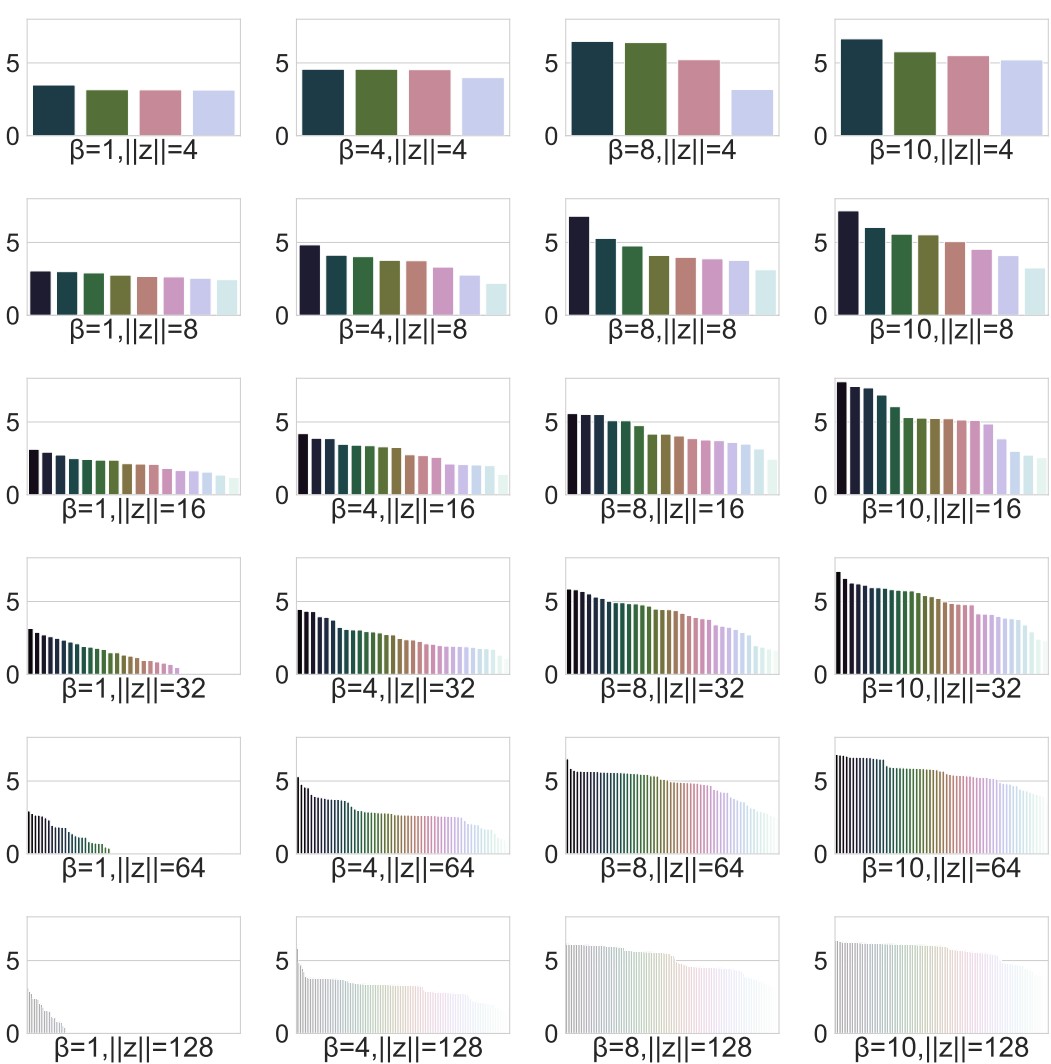

Figure F.3: $\mathbb{E}_{q(x)} D_{\mathrm{KL}}(q_\phi(z_j|x)||p(z_j))$ over Chairs for $\beta$-TCVAE over values of $||z||$ and $\beta$. Best viewed digitally.

**3D Faces**

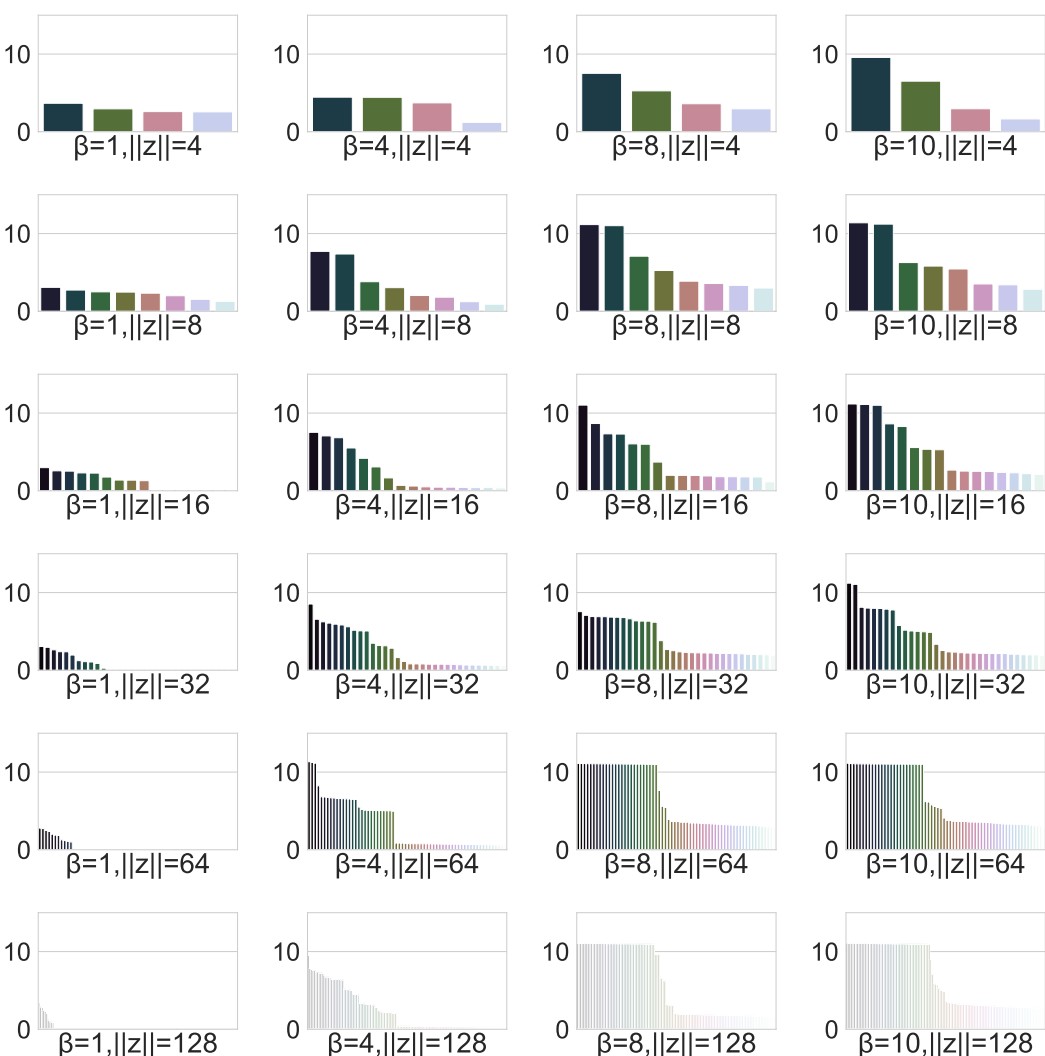

Figure F.4: $\mathbb{E}_{q(x)} D_{\mathrm{KL}}(q_\phi(z_j|x)||p(z_j))$ over 3D Faces for $\beta$-TCVAE over values of $||z||$ and $\beta$. Best viewed digitally.

## F.2 SEATBELT-VAES

Here the $D_{\mathrm{KL}}$ divergences are calculated per layer. **dSprites**

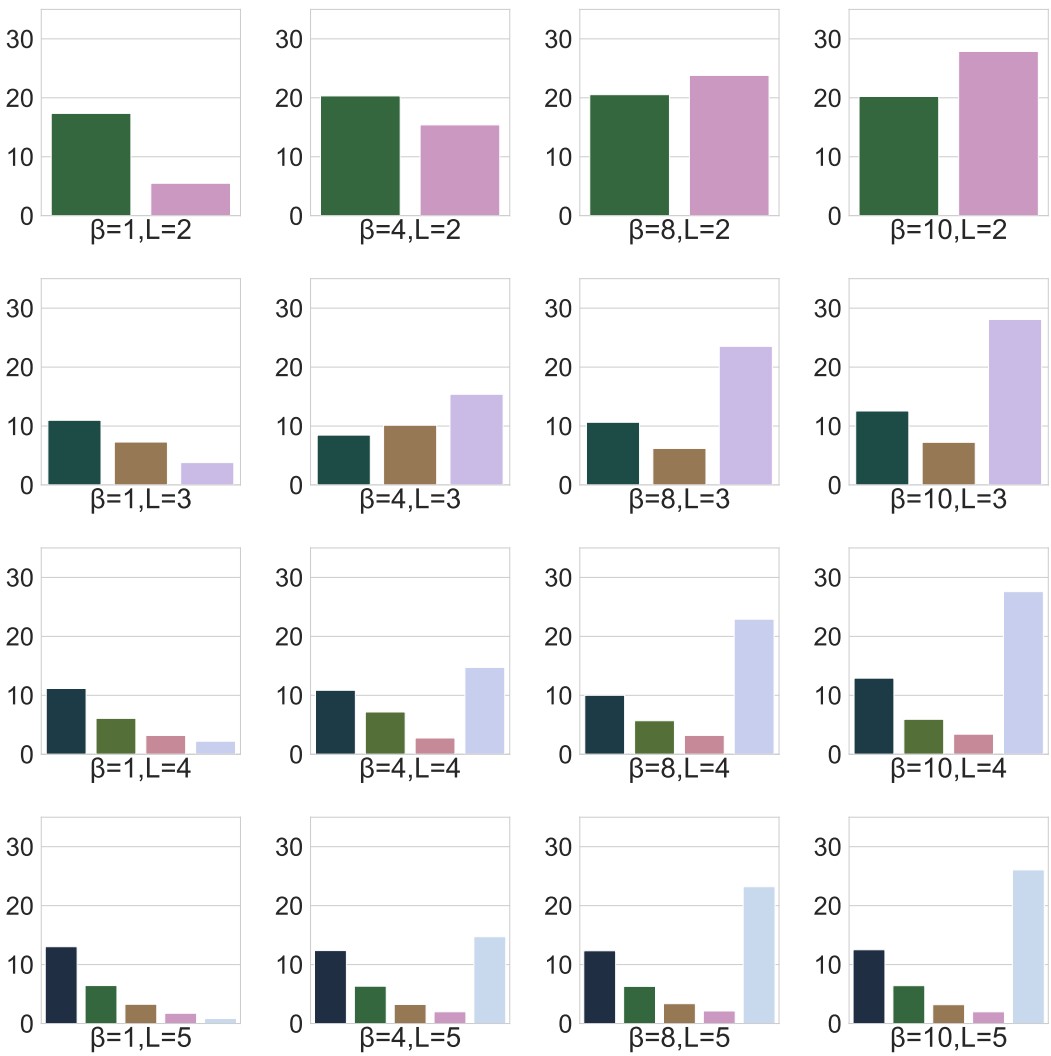

Figure F.5: $\mathbb{E}_{q_\phi(z^{i-1}, z^{i+1})} D_{\mathrm{KL}}(q_\phi(z^i|z^{i-1})||p(z^i|z^{i+1}))$ where $x = z^0$ and $p(z^L|z^{L+1}) = p(z^L)$, over dSprites for Seatbelt-VAEs over values of $L$ and $\beta$. Best viewed digitally.

**Chairs**

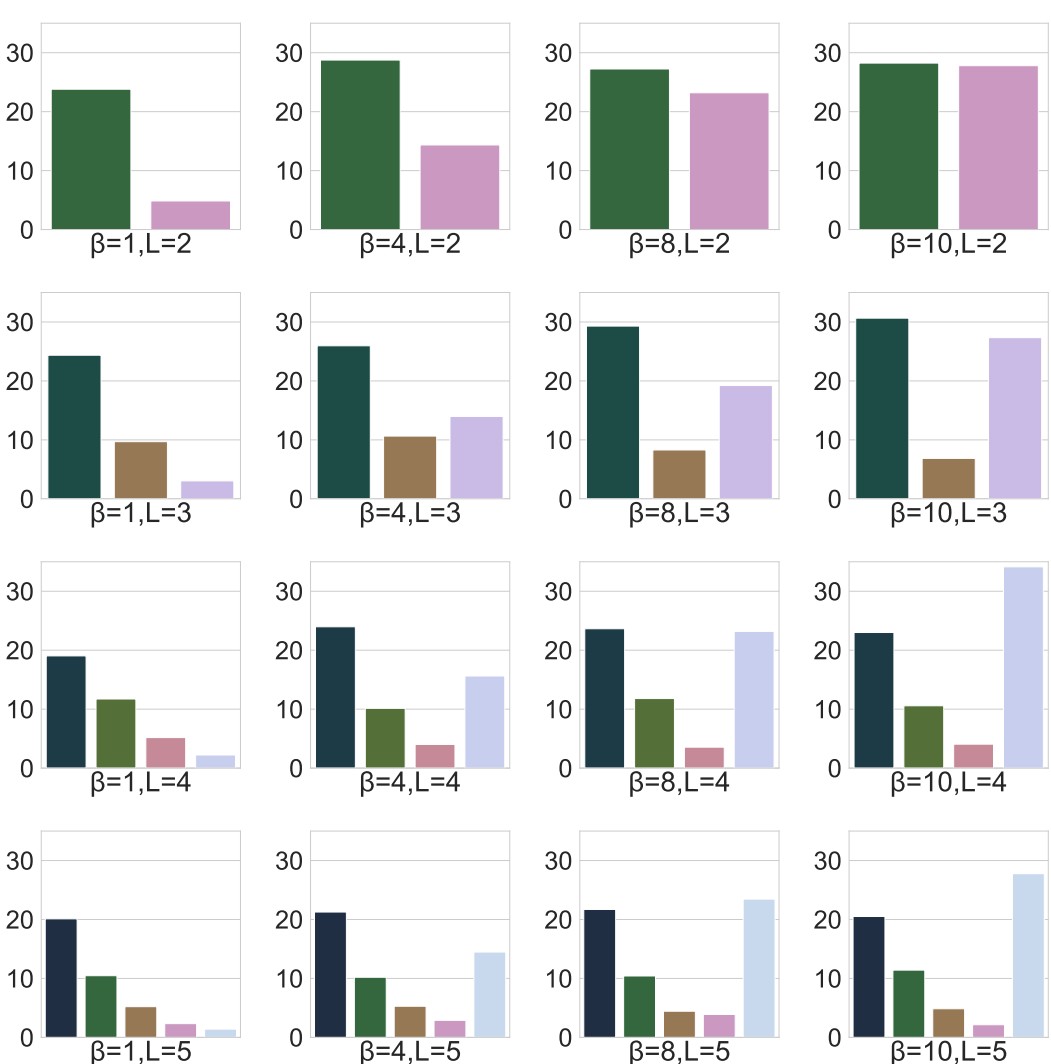

Figure F.6: $\mathbb{E}_{q_\phi(z^{i-1}, z^{i+1})} D_{\mathrm{KL}}(q_\phi(z^i|z^{i-1})||p(z^i|z^{i+1}))$ where $x = z^0$ and $p(z^L|z^{L+1}) = p(z^L)$, over Chairs for Seatbelt-VAEs over values of $L$ and $\beta$. Best viewed digitally.

**3D Faces**

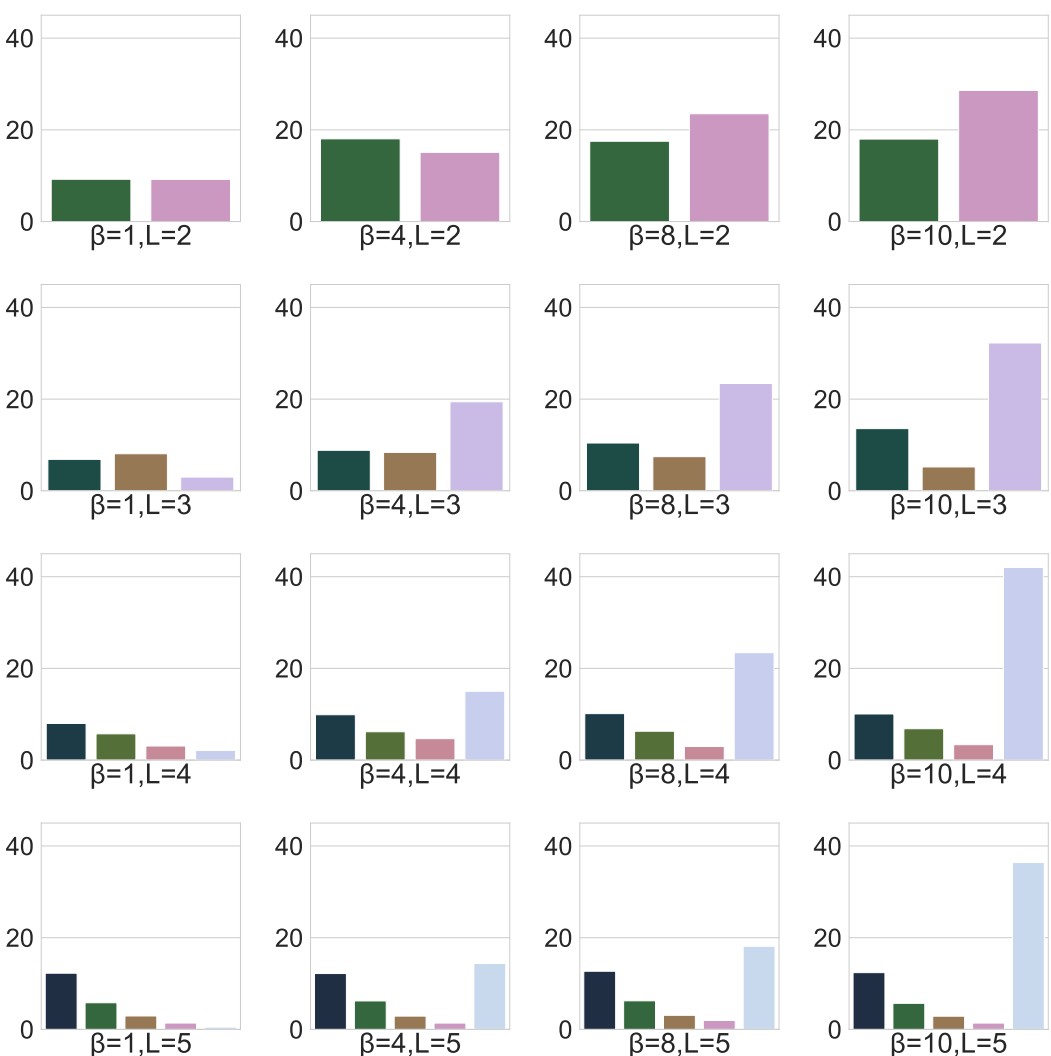

Figure F.7: $\mathbb{E}_{q_\phi(z^{i-1},z^{i+1})} D_{\mathrm{KL}}(q_\phi(z^i|z^{i-1})||p(z^i|z^{i+1}))$ where $x = z^0$ and $p(z^L|z^{L+1}) = p(z^L)$, over 3D Faces for Seatbelt-VAEs over values of $L$ and $\beta$. Best viewed digitally.

# G AGGREGATE ANALYSIS OF ADVERSARIAL ATTACK

## G.1 $\beta$-TCVAE

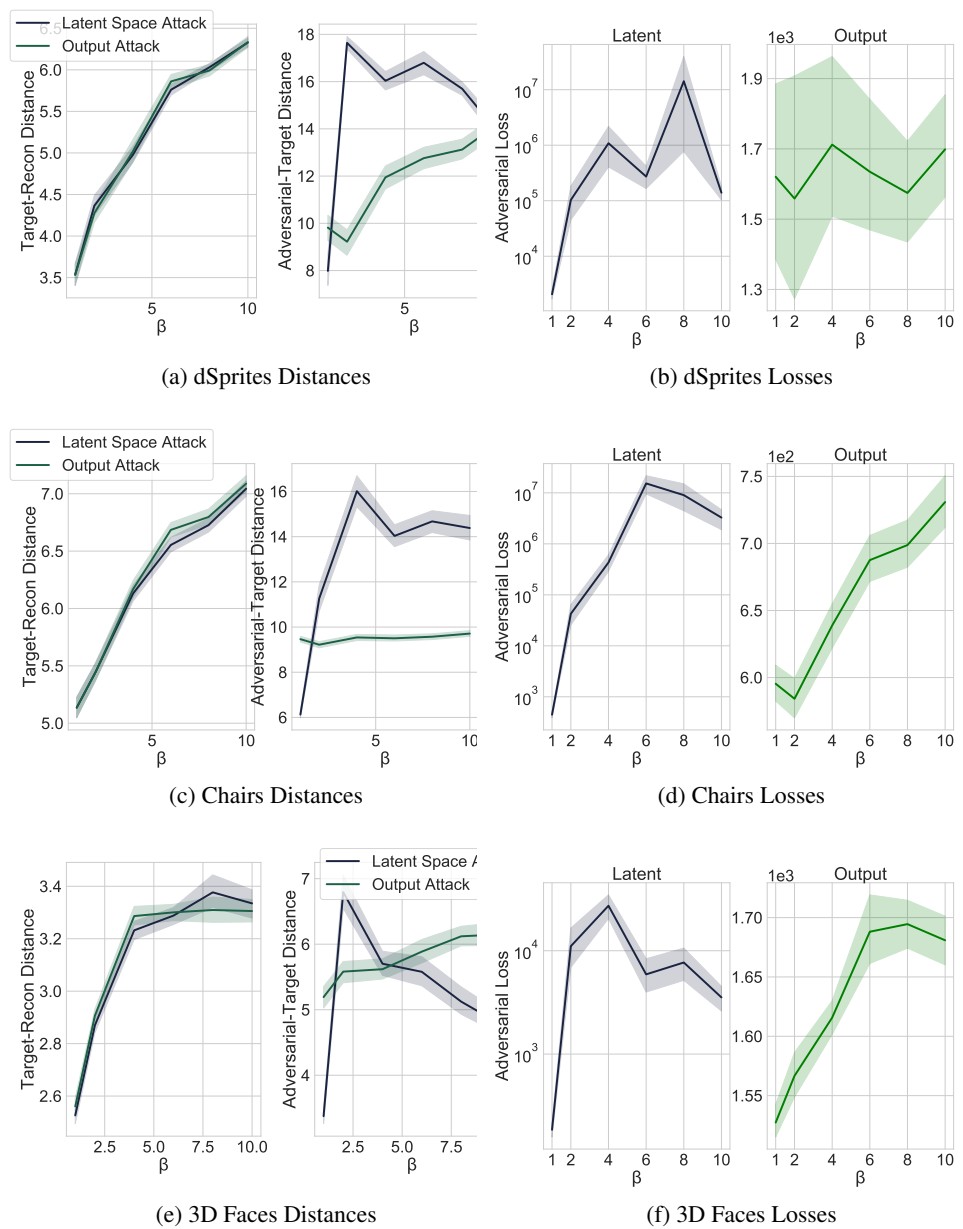

Figure G.8: Plots showing the effect of varying $\beta$ in a $\beta$-TCVAE for dSprites and 3D Faces on: (a),(d) the $L_2$ distance from $x^t$ to its reconstruction when given as input and the $L_2$ distance from the adversarial input $x^*$ and its reconstruction; (b),(e) the adversarial objectives $\Delta_{\text{latnet/output}}$

## G.2 $\beta$-TCDLGMs

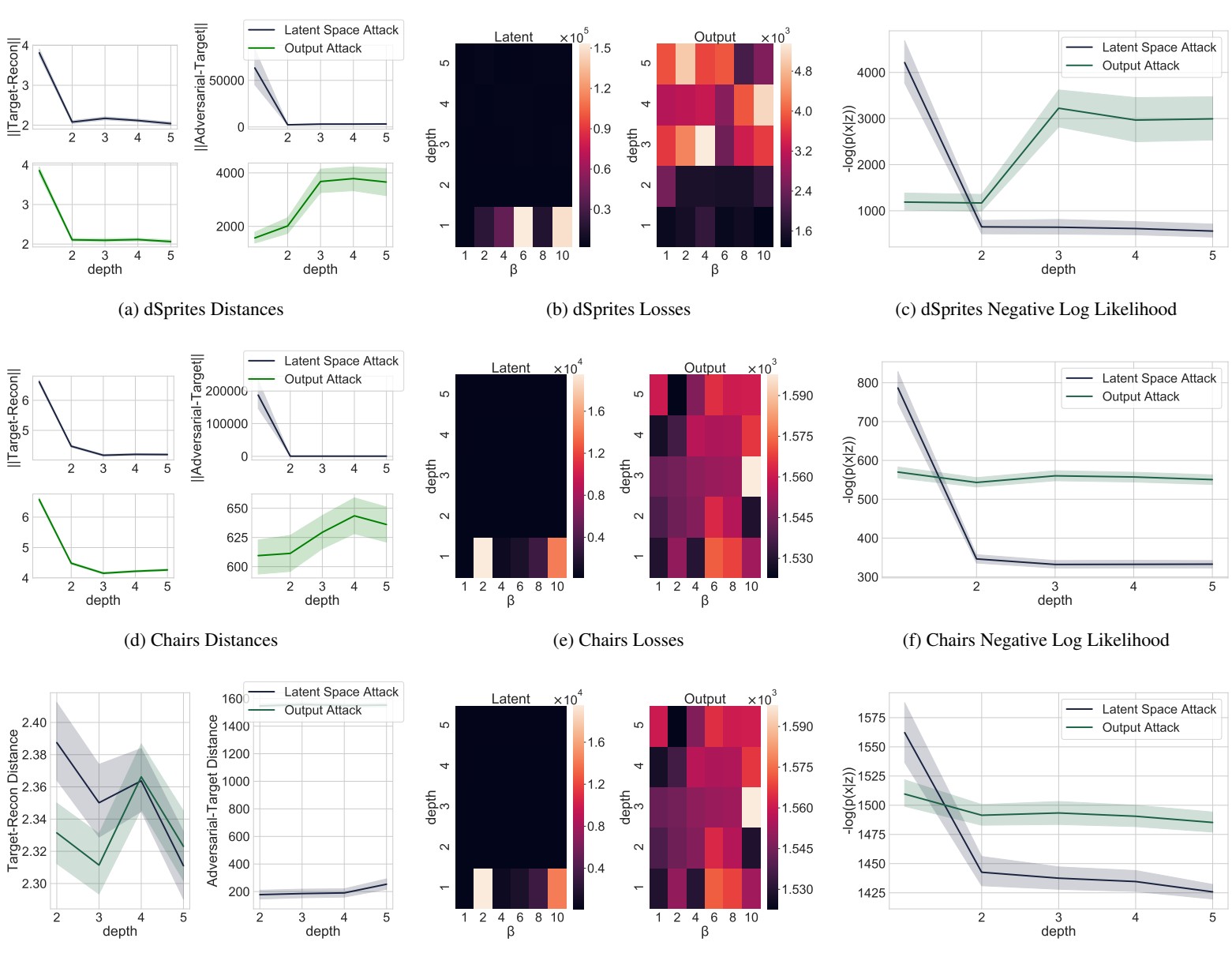

Figure G.9: Plots showing the effect of varying $L$ on $\beta$-TCDLGMs for dSprites 3D Faces and Chairs, on: (a),(d),(g) the $L_2$ distance between $x^t$ and its reconstruction when given as input and the same between the adversarial input $x^*$ and its reconstruction; (b),(e),(h) the adversarial objectives $\Delta_{\text{output/image}}$; (c),(f),(i) $-\log p_\theta(x^t|z), z \sim q_\phi(z|x^*)$ and the MIG.

For a DLGM (Rezende et al., 2014) with 2-5 $z$ layers, with $q_\phi(\mathbf{z}|x)$ factorised as in Eq (7), $p_\theta(x, \mathbf{z})$ factorised as in Eq (6), and $\beta$TC penalisation applied to the top layer, we find that latent attacks targeted at $z^1$ are highly effective and remain so as $L$ and $\beta$ each increase. These models are, however, slightly more robust to output attacks and this attack becomes less effective as $\beta$ increases, but more effective as $L$ increases.

The ease of attacking via $z^1$ is consistent with its separation out from the rest of the model.

## G.3 SEATBELT-VAEs

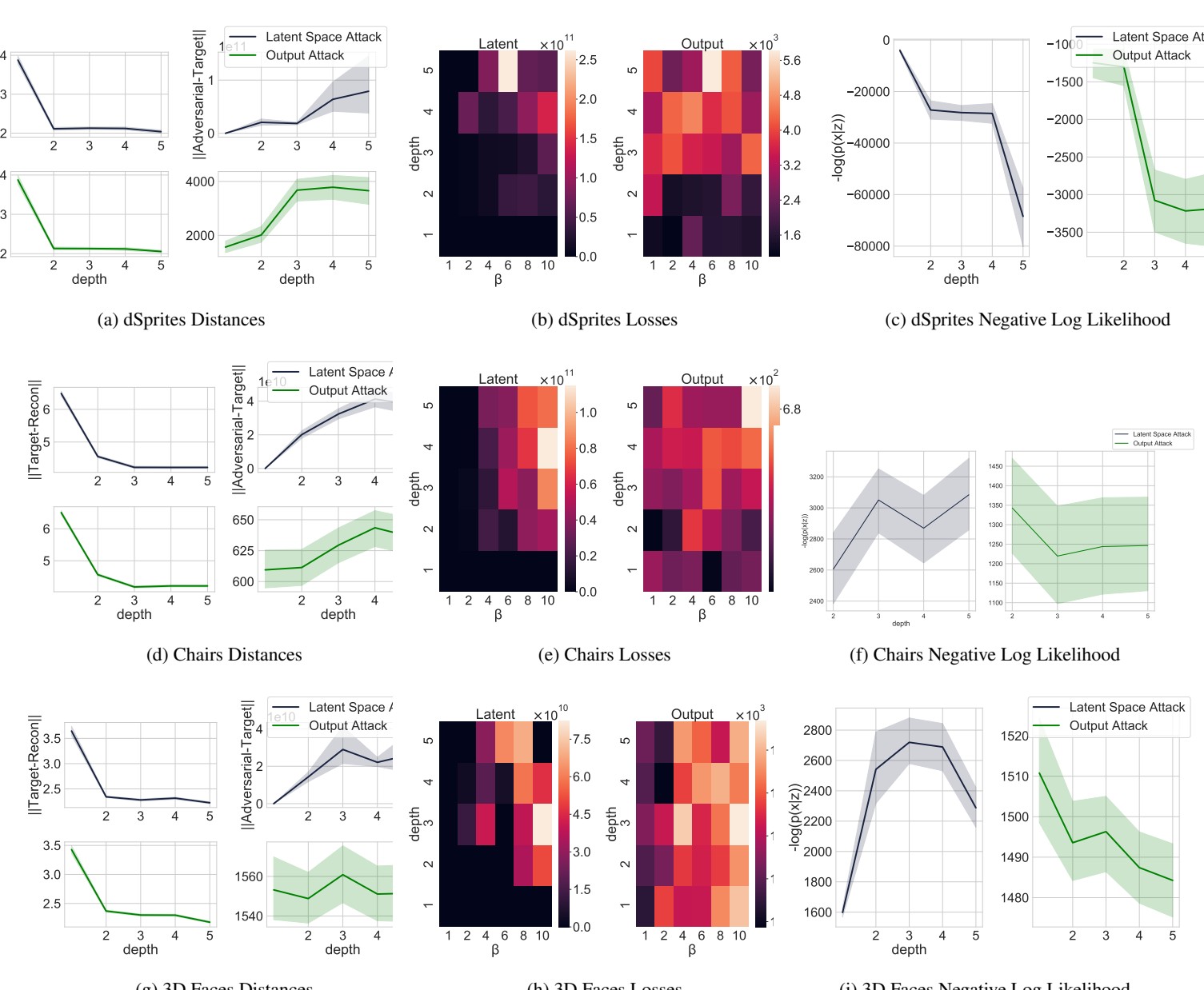

Figure G.10: Plots showing the effect of varying $L$, $\beta$ on Seatbelt-VAEs trained on dSprites, 3D Faces and Chairs on: (a),(d),(g) the $L_2$ distance between $x^t$ and its reconstruction when given as input and the same between the adversarial input $x^*$ and its reconstruction; (b),(e),(h) the adversarial objectives $\Delta_{\text{output/image}}$; (c),(f),(i) $-\log p_\theta(x^t|z), z \sim q_\phi(z|x^*)$.

### G.4 SEATBELT-VAE LAYERWISE ATTACKS

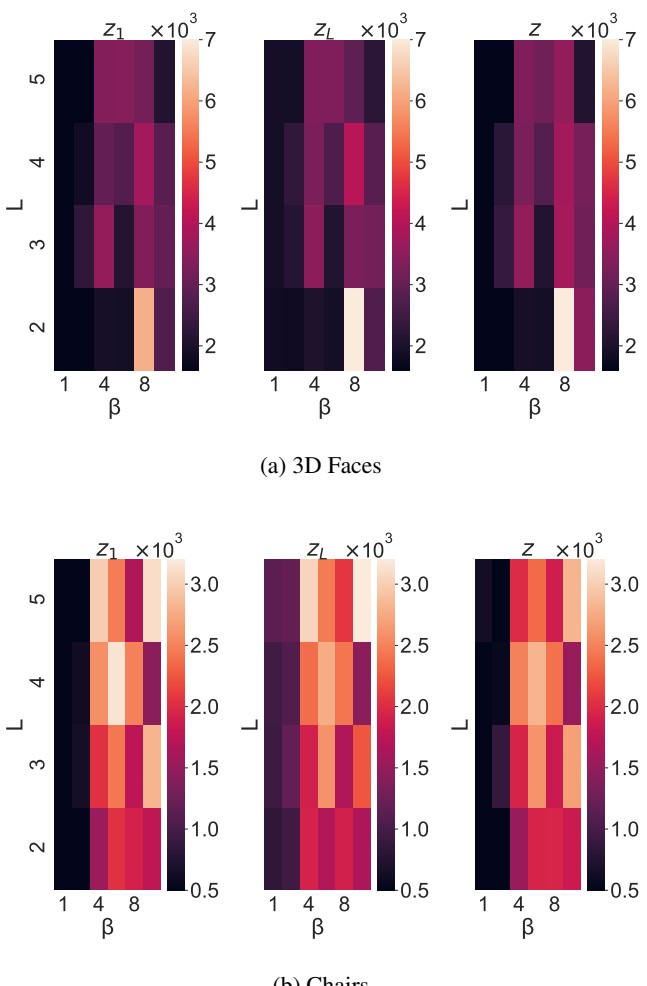

(a) 3D Faces

(b) Chairs

Figure G.11: $-\log p_\theta(x^t|\tilde{z})$ for Seatbelt-VAEs for (a) 3D Faces and (b) Chairs; over $\beta$ and $L$ values for *latent* attacks. We attack the bottom layer ($z^1$), the top layer ($z^L$), and finally show the effect when attacking all layers ($z$). Larger values of $-\log p_\theta(x^t|\tilde{z})$ correspond to less successful adversarial attacks.

# H    ADVERSARIAL ATTACK PLOTS

## H.1    DSPRITES ADVERSARIAL ATTACK ON A SINGLE FACTOR

**Latent Attack**

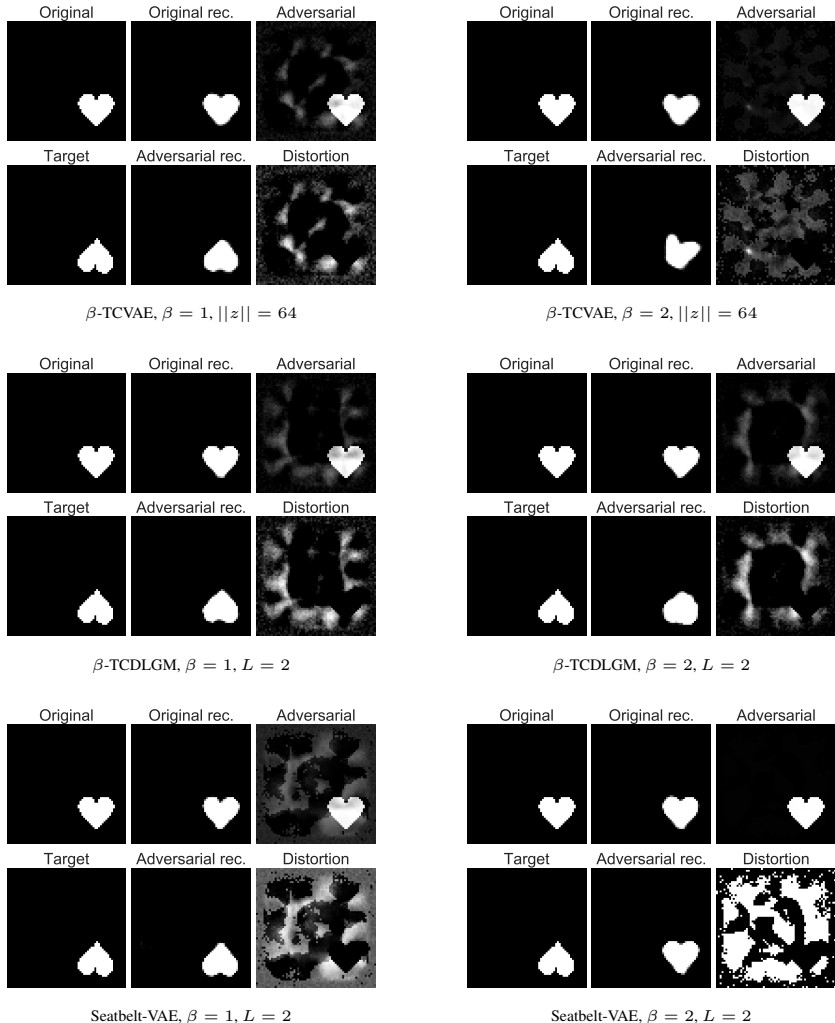

Figure H.12: Latent space attacks on rotation only of a heart-shaped dSprite for $\beta$-TCVAEs, $\beta$-TCDLGMs and Seatbelt-VAEs for $\beta = \{1, 2\}$.

## H.2 DSPRITES ADVERSARIAL ATTACK

### H.2.1 $\beta$-TCVAEs

**Output Attack**

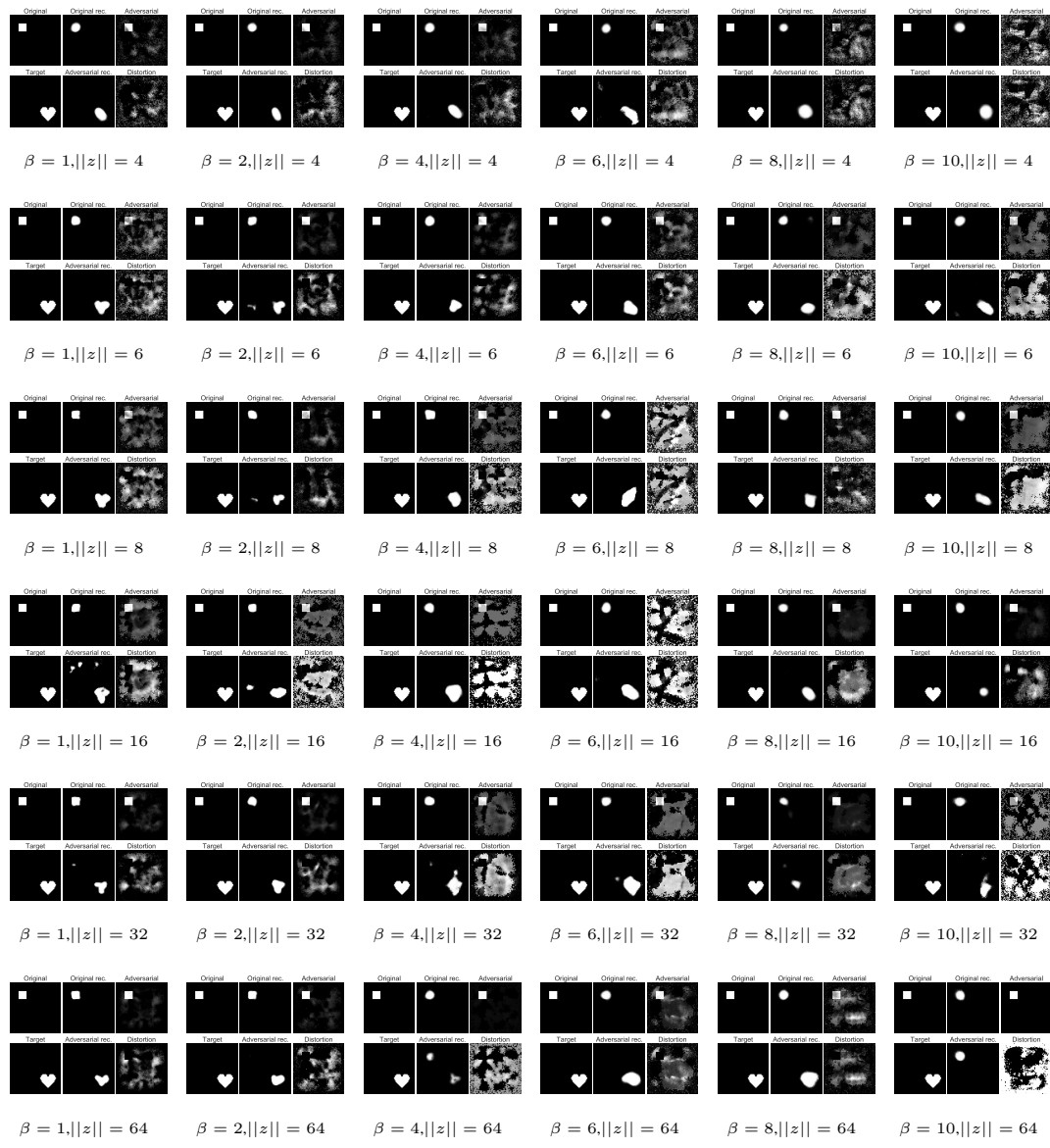

Figure H.13: Output attacks on dSprites on $\beta$-TCVAEs for $\beta = \{1, 2, 4, 6, 8, 10\}$ and $||z|| = \{4, 6, 8, 16, 32, 64\}$.

**Latent Attack**

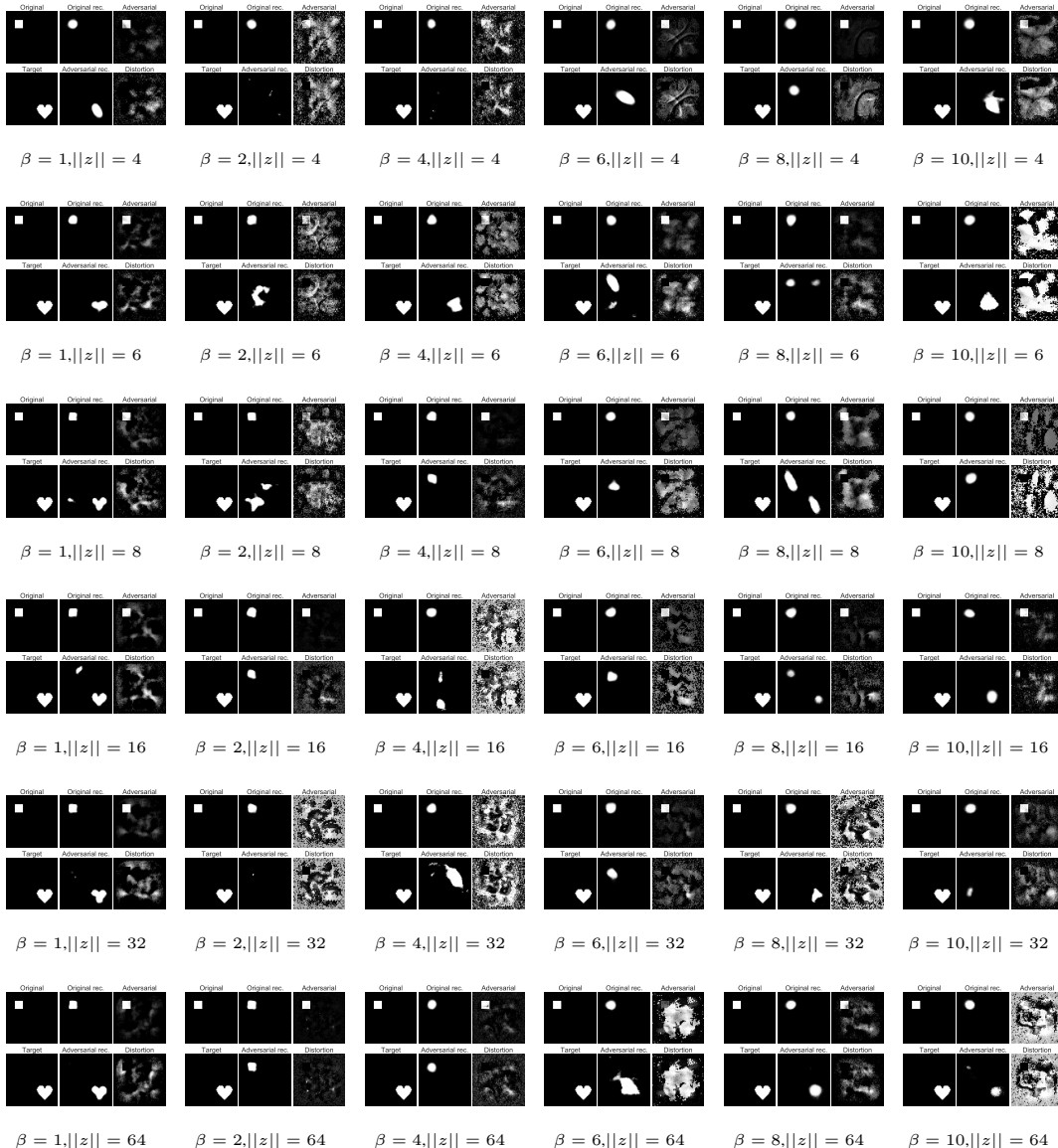

Figure H.14: Latent attacks on dSprites on $\beta$-TCVAEs for $\beta = \{1, 2, 4, 6, 8, 10\}$ and $||z|| = \{4, 6, 8, 16, 32, 64\}$.

### H.2.2 $\beta$-TCDLGMs

**Output Attack**

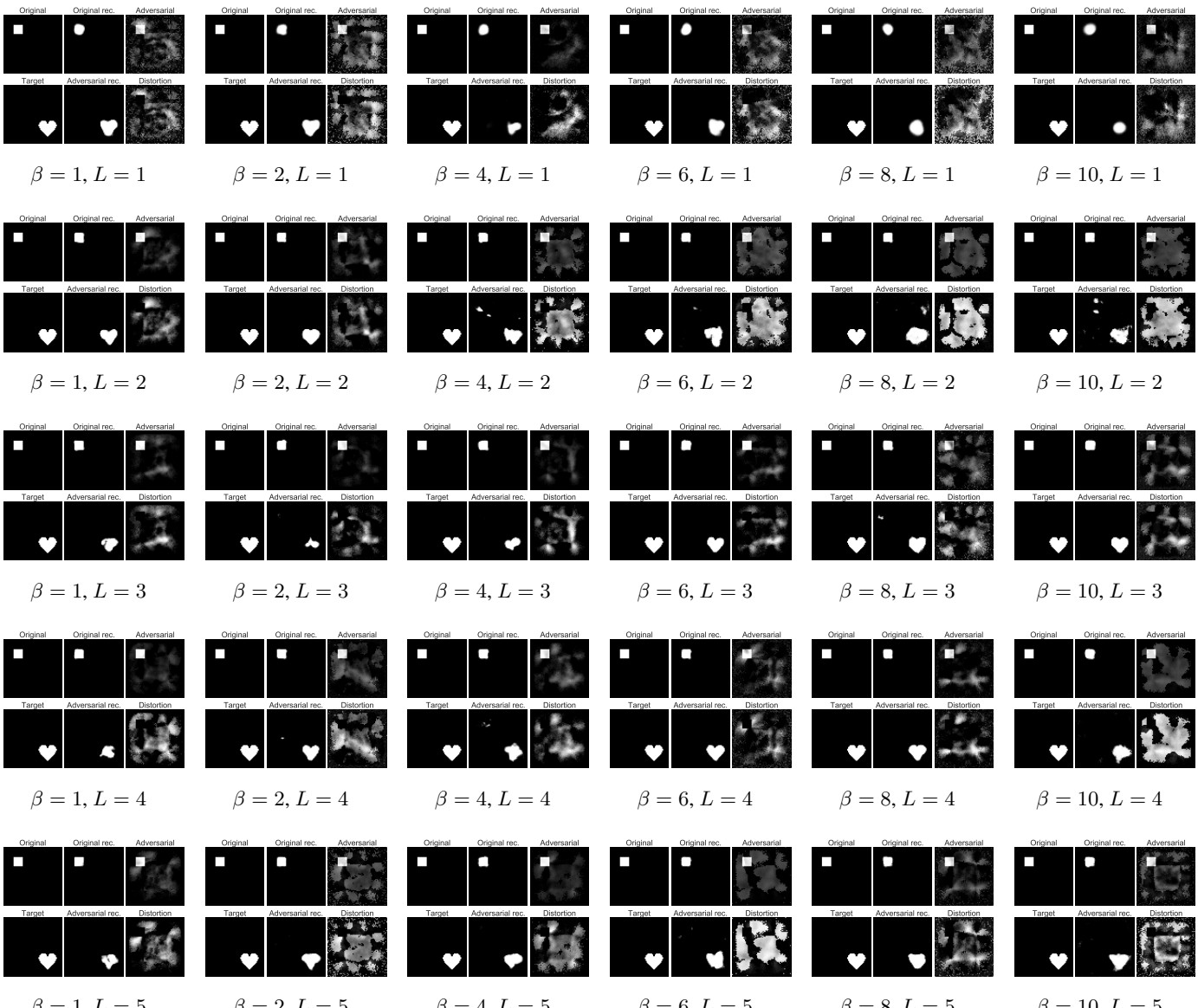

Figure H.15: Output attacks on dSprites on $\beta$-TCDLGMs for $\beta = \{1, 2, 4, 6, 8, 10\}$ and $L = \{1, 2, 3, 4, 5\}$.

**Latent Attack**

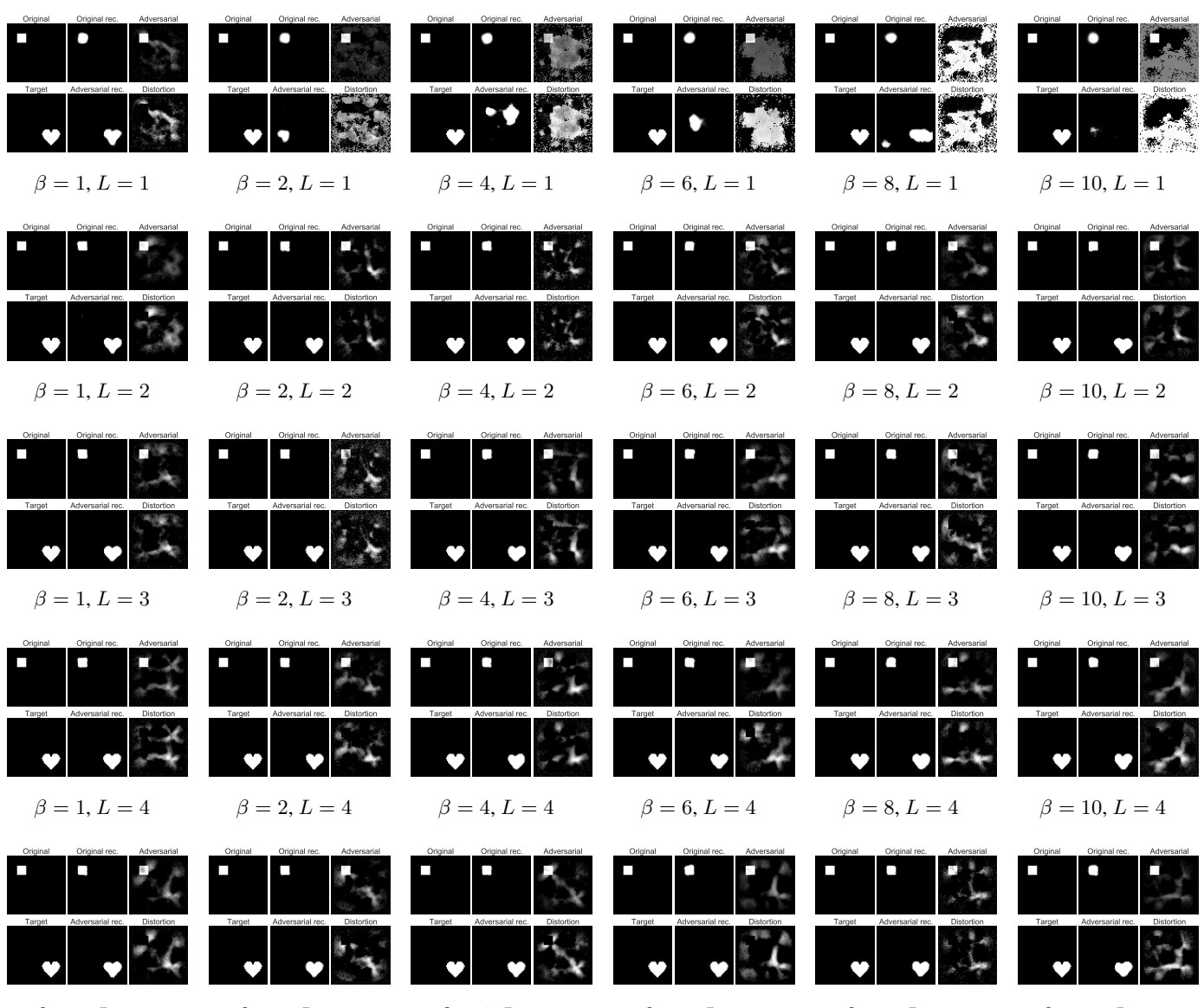

Figure H.16: Latent attacks on dSprites on $\beta$-TCDLGMs for $\beta = \{1, 2, 4, 6, 8, 10\}$ and $L = \{1, 2, 3, 4, 5\}$.

### H.2.3 SEATBELT-VAEs

**Output Attack**

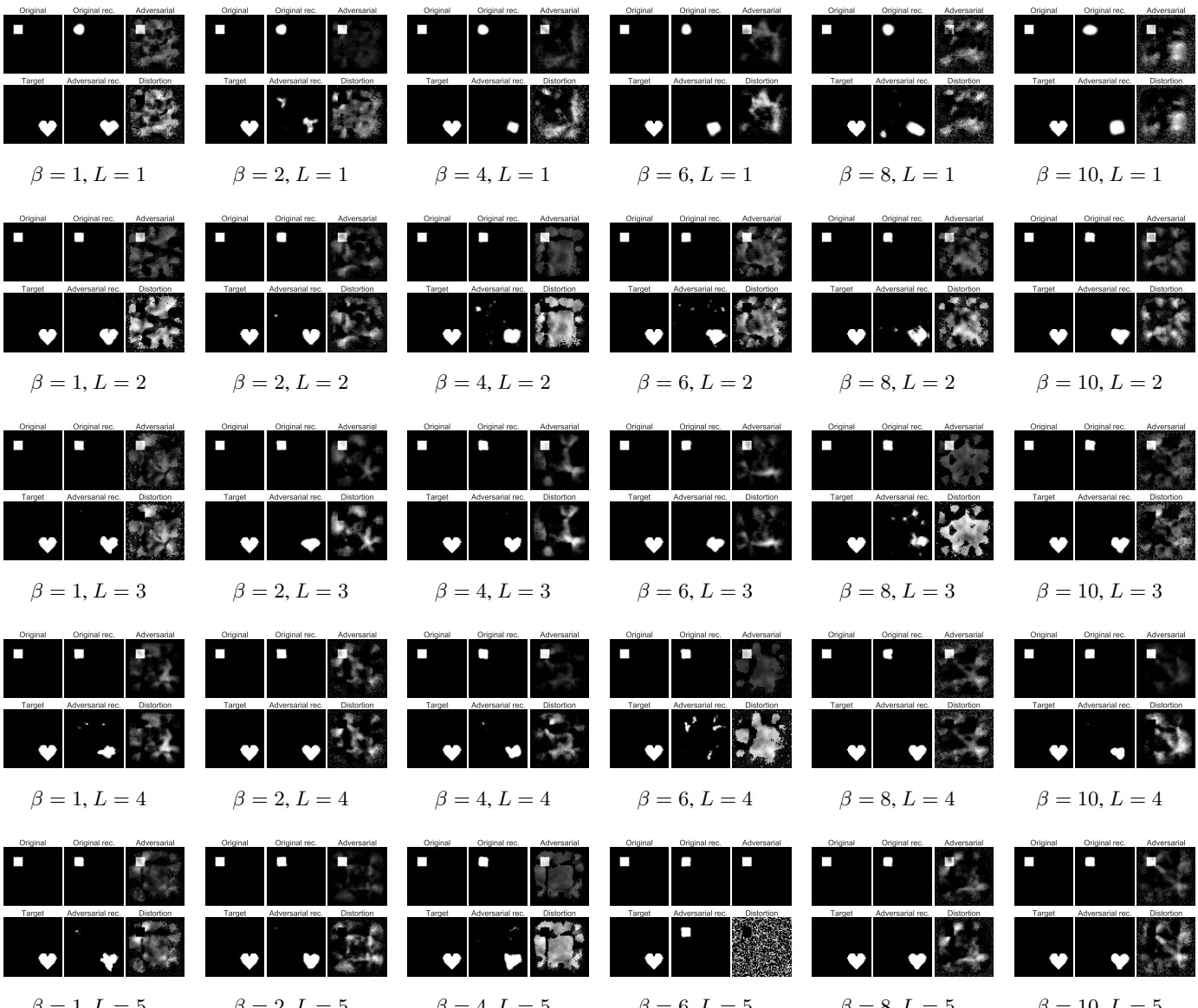

Figure H.17: Output attacks on dSprites on Seatbelt-VAEs for $\beta = \{1, 2, 4, 6, 8, 10\}$ and $L = \{1, 2, 3, 4, 5\}$.

**Latent Attack**

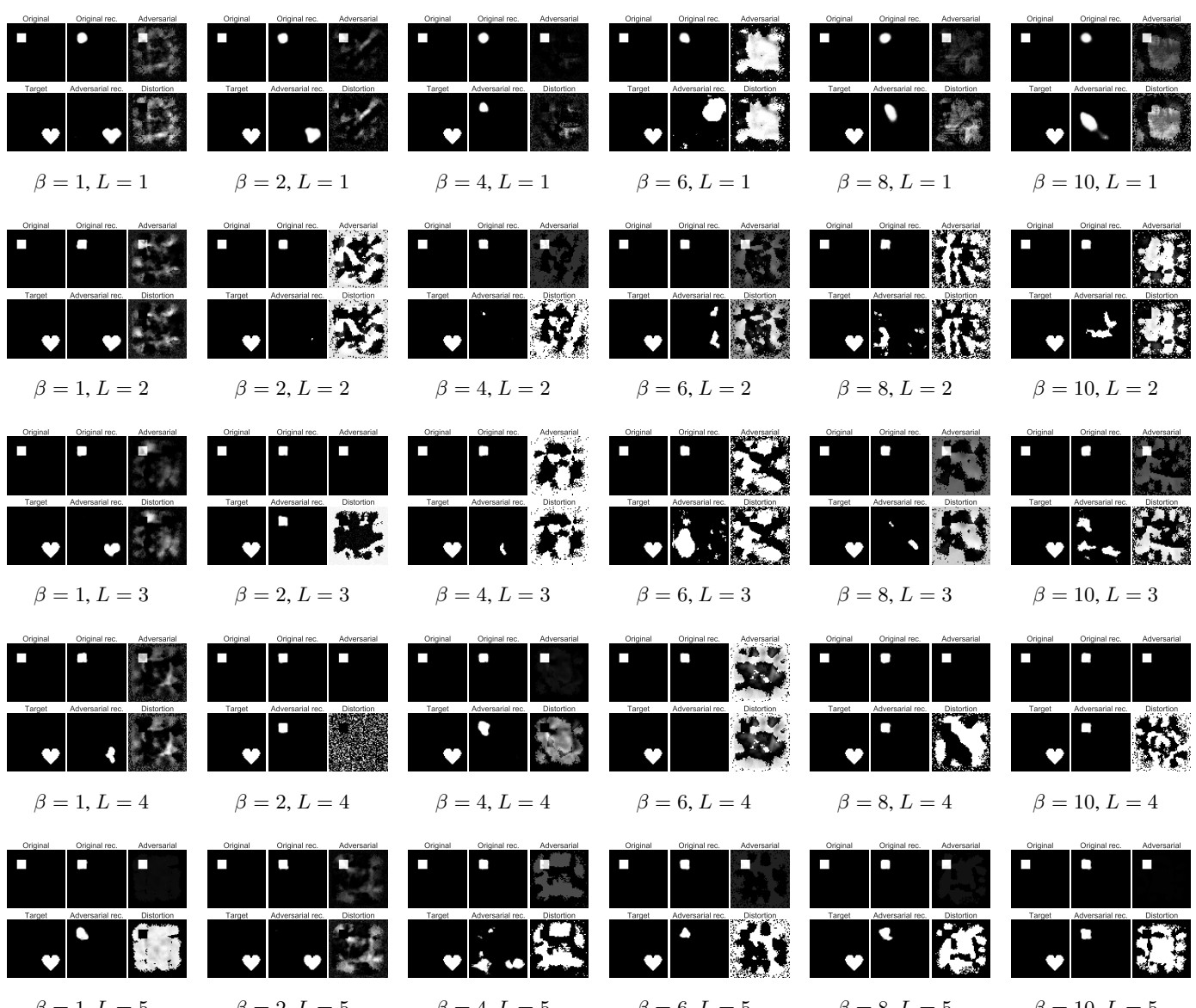

Figure H.18: Latent attacks on dSprites for Seatbelt-VAEs for $\beta = \{1, 2, 4, 6, 8, 10\}$ and $L = \{1, 2, 3, 4, 5\}$.

## H.3 CHAIRS ADVERSARIAL ATTACK

### H.3.1 $\beta$-TCVAEs

**Output**

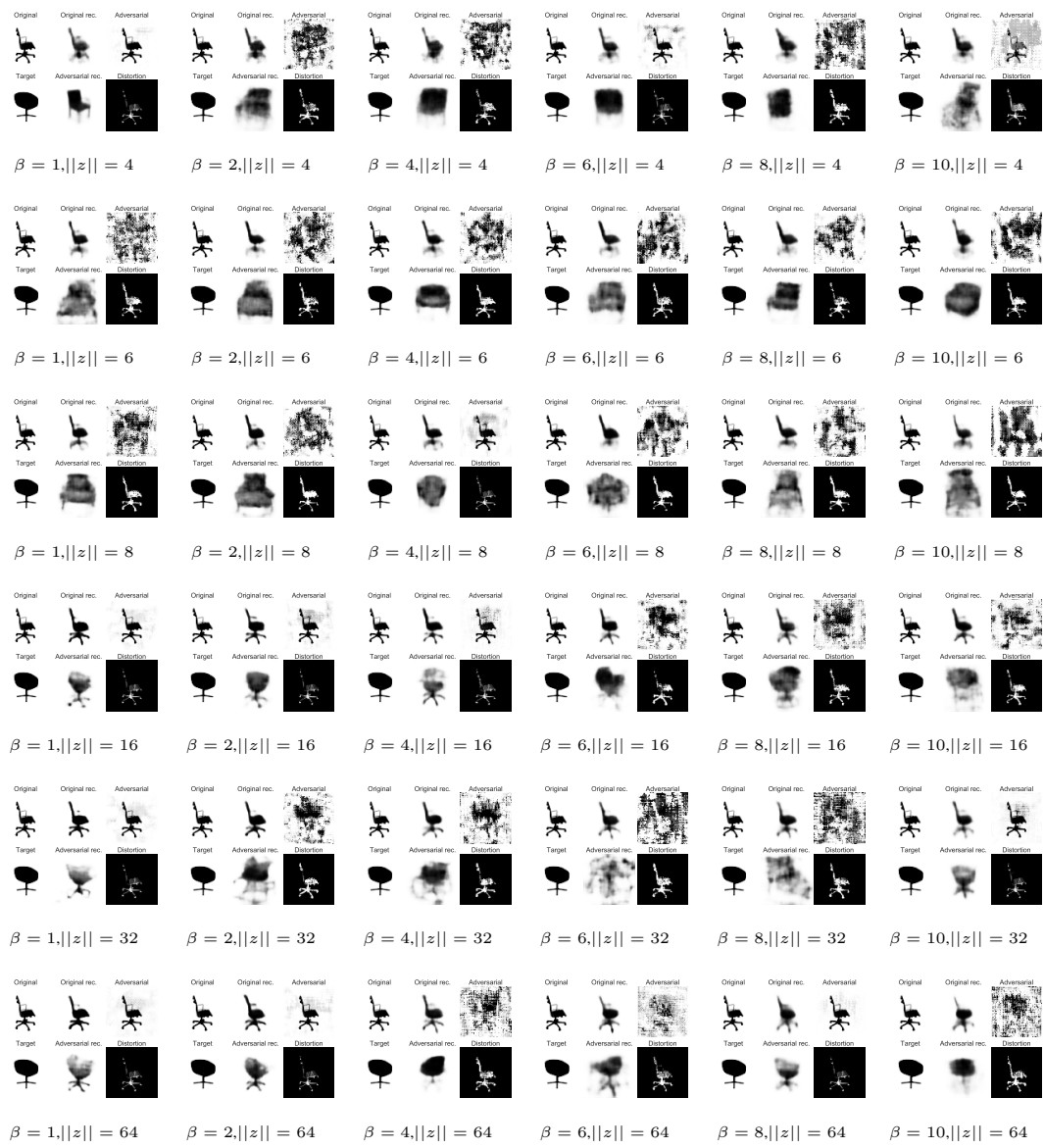

Figure H.19: Output attacks on Chairs for $\beta$-TCVAEs for $\beta = \{1, 2, 4, 6, 8, 10\}$ and $||z|| = \{4, 6, 8, 16, 32, 64\}$.

**Latent**

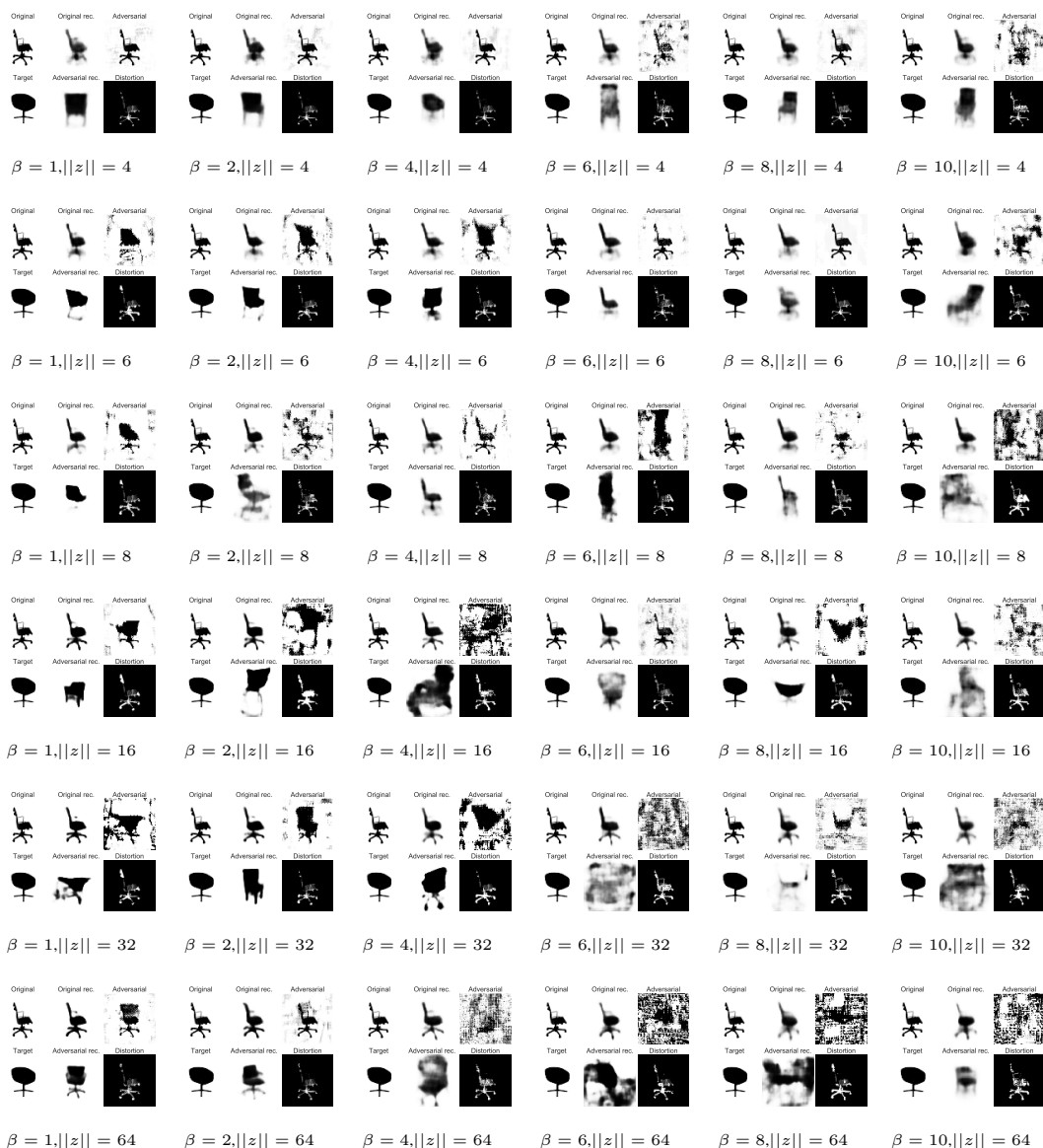

Figure H.20: Latent attacks on Chairs for $\beta$-TCVAEs for $\beta = \{1, 2, 4, 6, 8, 10\}$ and $||z|| = \{4, 6, 8, 16, 32, 64\}$.

### H.3.2 $\beta$-TCDLGMs

**Output Attack**

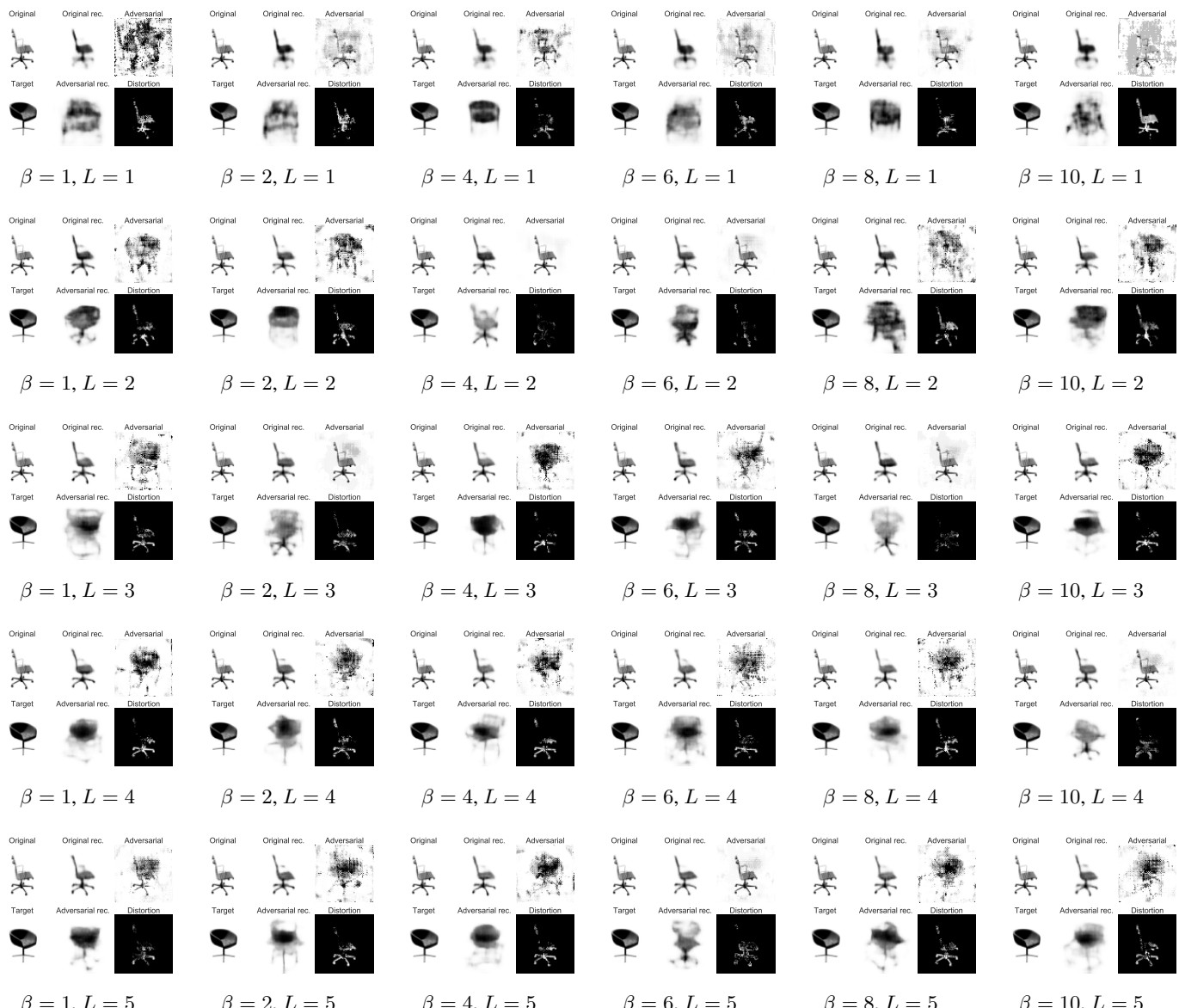

Output attacks on Chairs for $\beta$-TCDLGMs for $\beta = \{1, 2, 4, 6, 8, 10\}$ and $L = \{1, 2, 3, 4, 5\}$.

**Latent Attack**

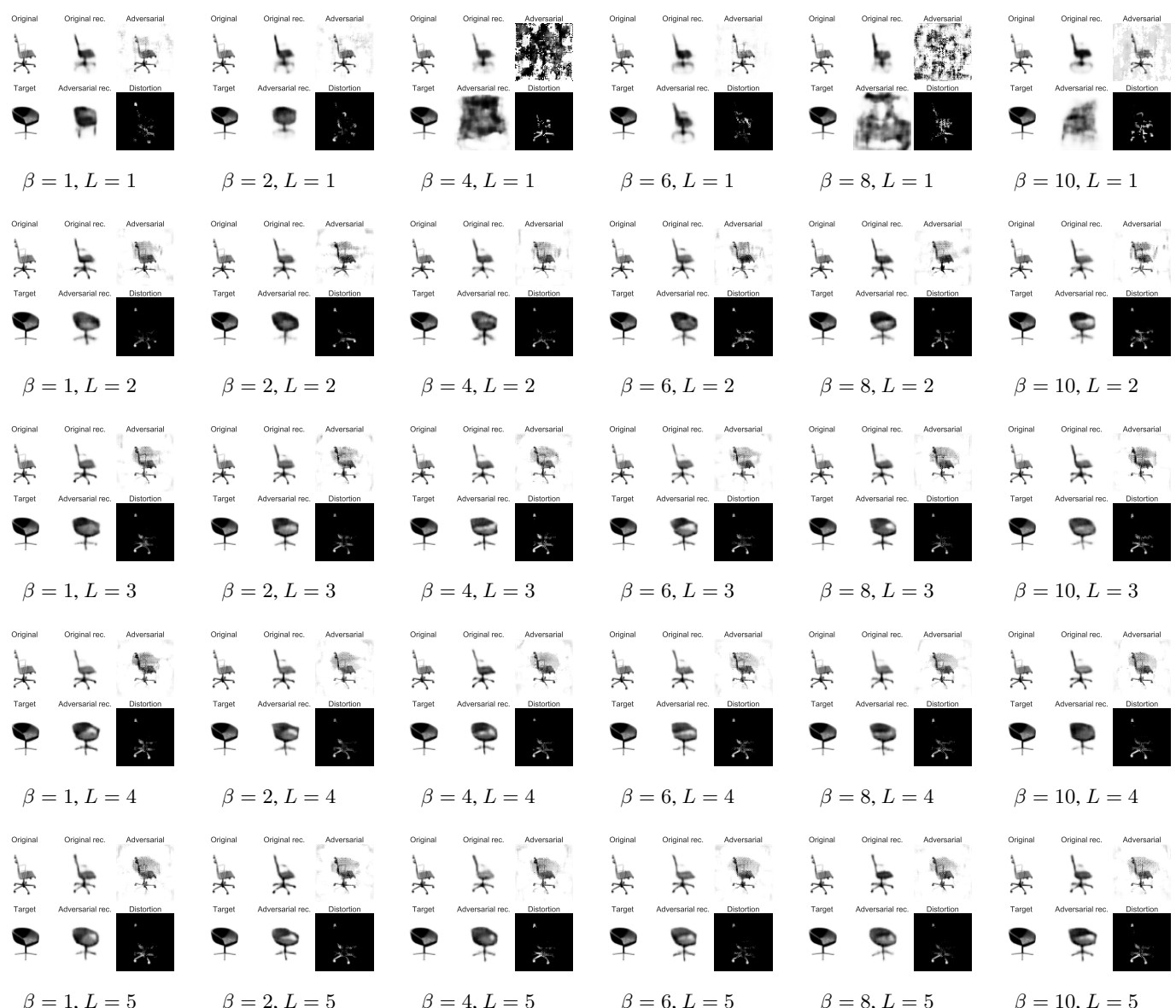

Figure H.21: Latent attacks on Chairs for $\beta$-TCDLGMs for $\beta = \{1, 2, 4, 6, 8, 10\}$ and $L = \{1, 2, 3, 4, 5\}$.

### H.3.3 SEATBELT-VAEs

**Output Attack**

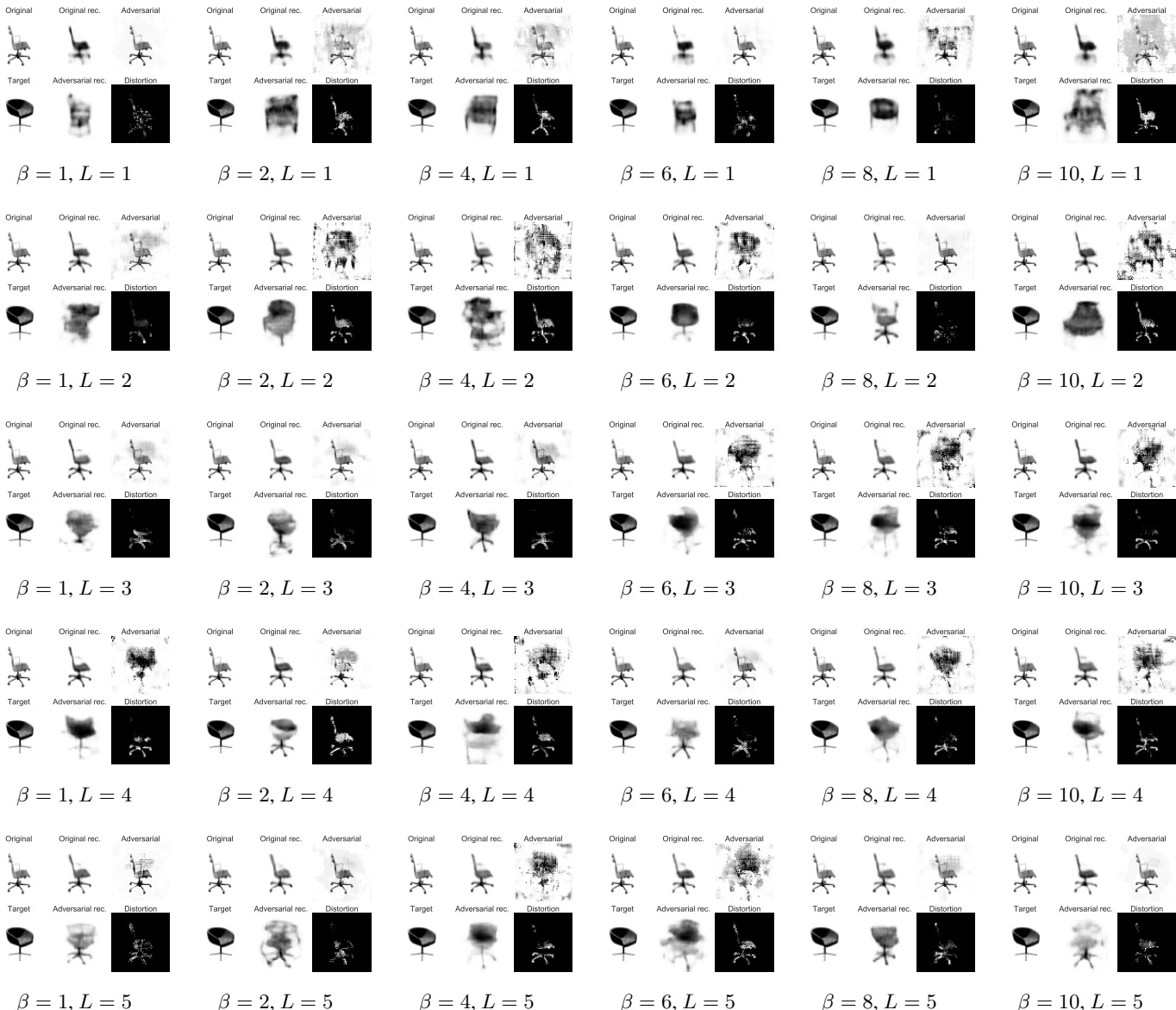

Figure H.22: Output attacks on Chairs for Seatbelt-VAEs for $\beta = \{1, 2, 4, 6, 8, 10\}$ and $L = \{1, 2, 3, 4, 5\}$.

**Latent Attack**

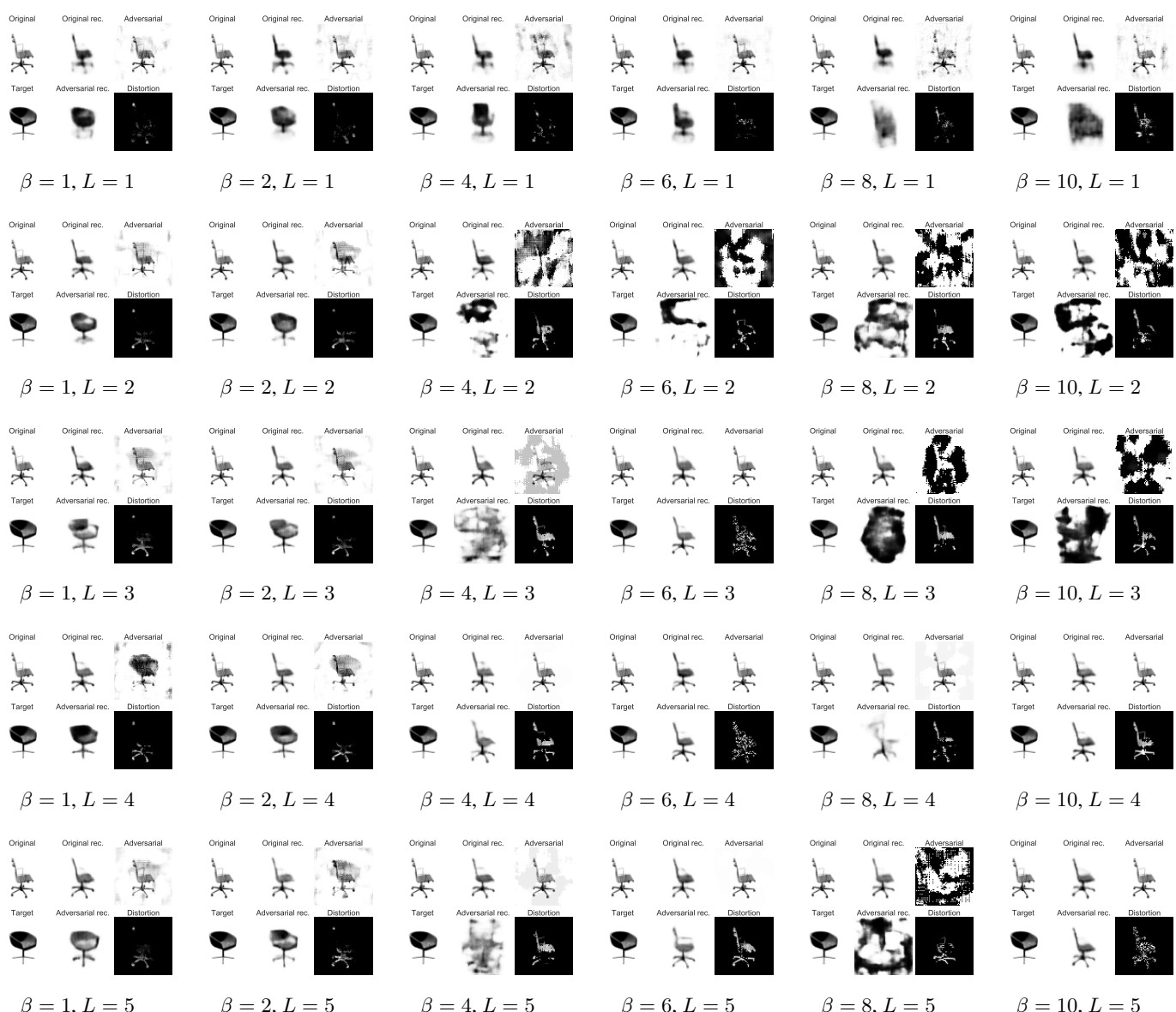

Figure H.23: Latent attacks on Chairs for Seatbelt-VAEs for $\beta = \{1, 2, 4, 6, 8, 10\}$ and $L = \{1, 2, 3, 4, 5\}$.

## H.4 3D FACES ADVERSARIAL ATTACK

### H.4.1 $\beta$-TCVAES

**Output Attack**

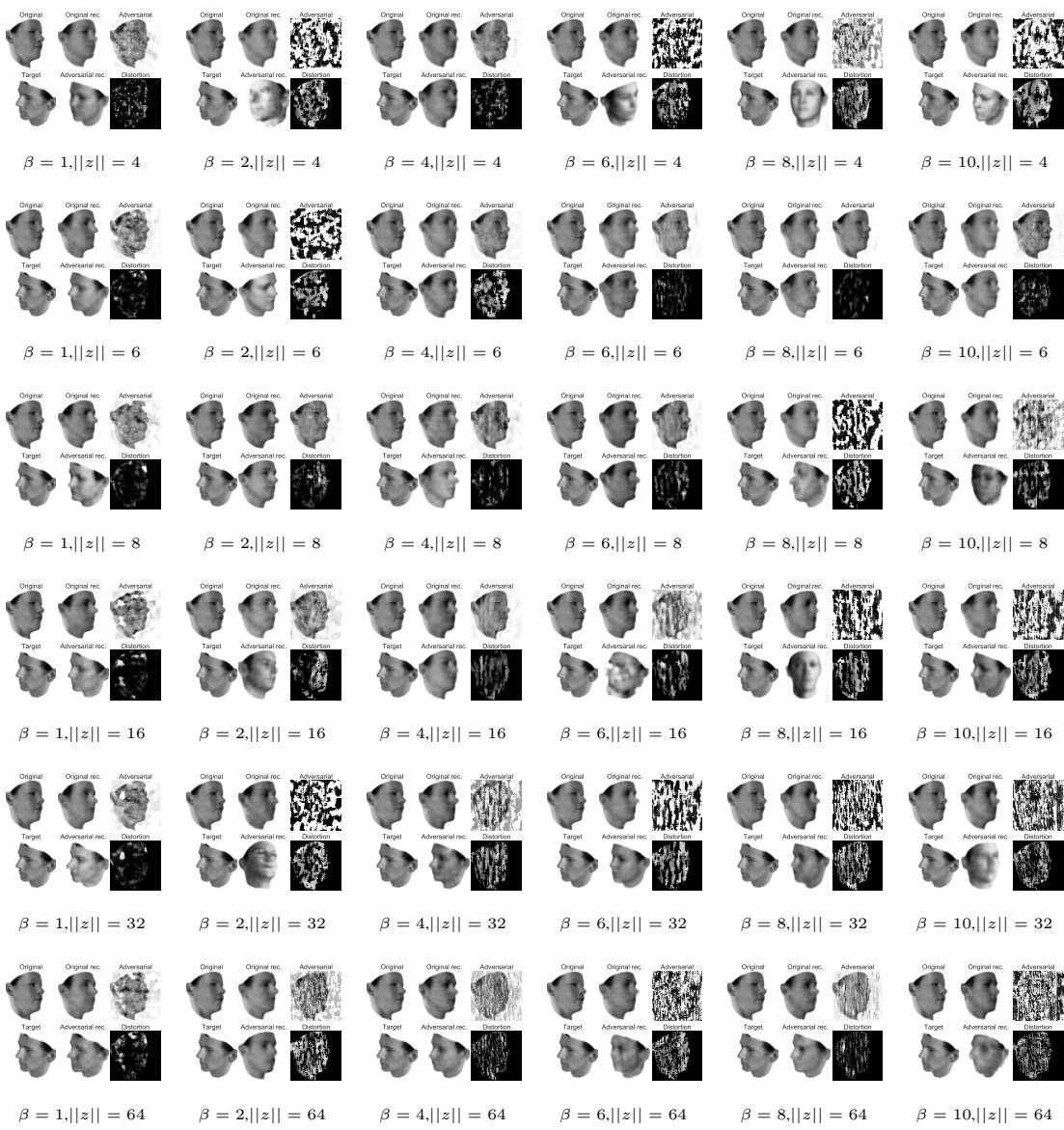

Figure H.24: Output attacks on 3D Faces for $\beta$-TCVAEs for $\beta = \{1, 2, 4, 6, 8, 10\}$ and $||z|| = \{4, 6, 8, 16, 32, 64\}$.

**Latent Attack**

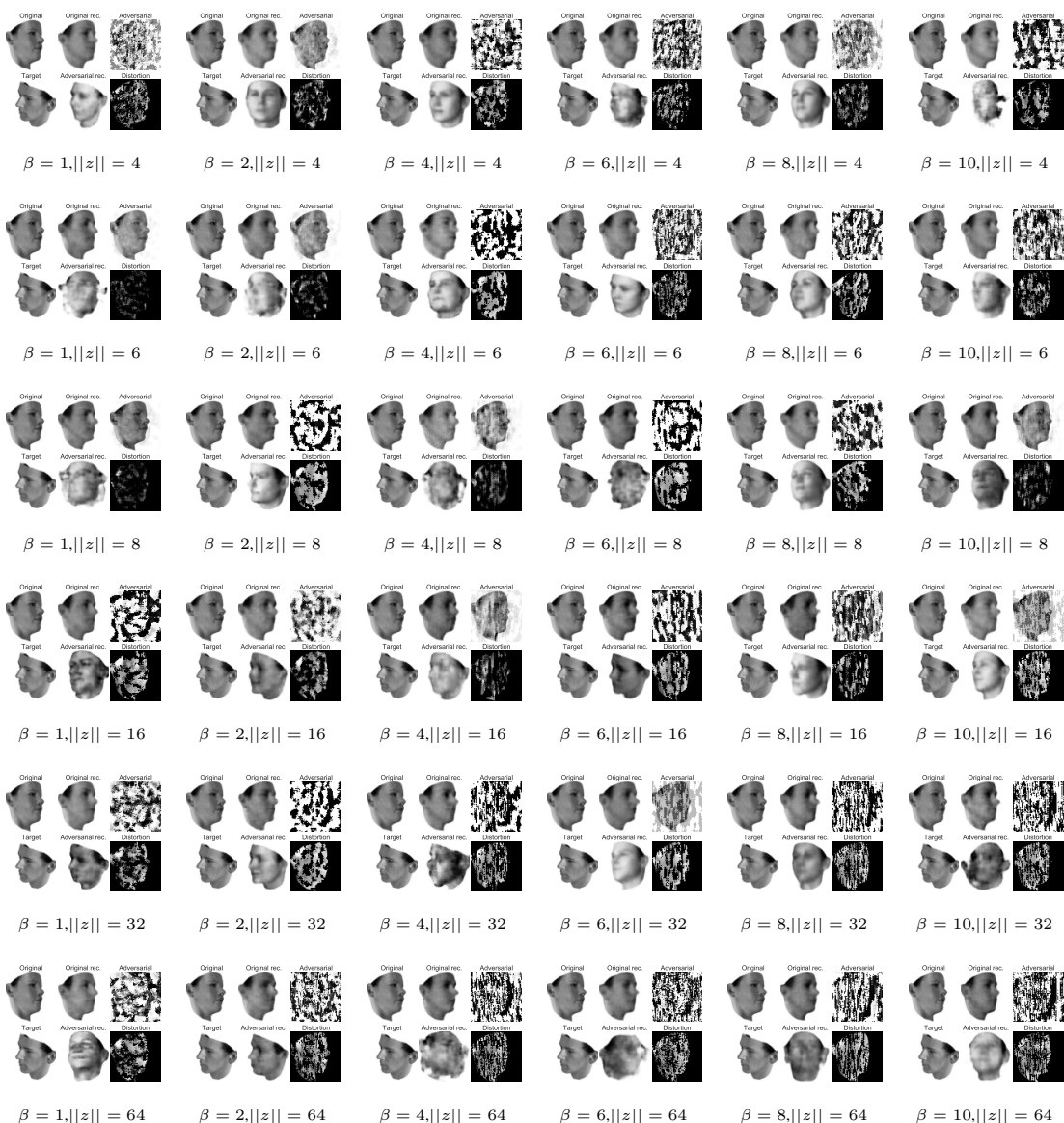

Figure H.25: Latent attacks on 3D Faces for $\beta$-TCVAEs for $\beta = \{1, 2, 4, 6, 810\}$ and $||z|| = \{4, 6, 8, 16, 32, 64\}$.

### H.4.2 $\beta$-TCDLGMs

**Output Attack**

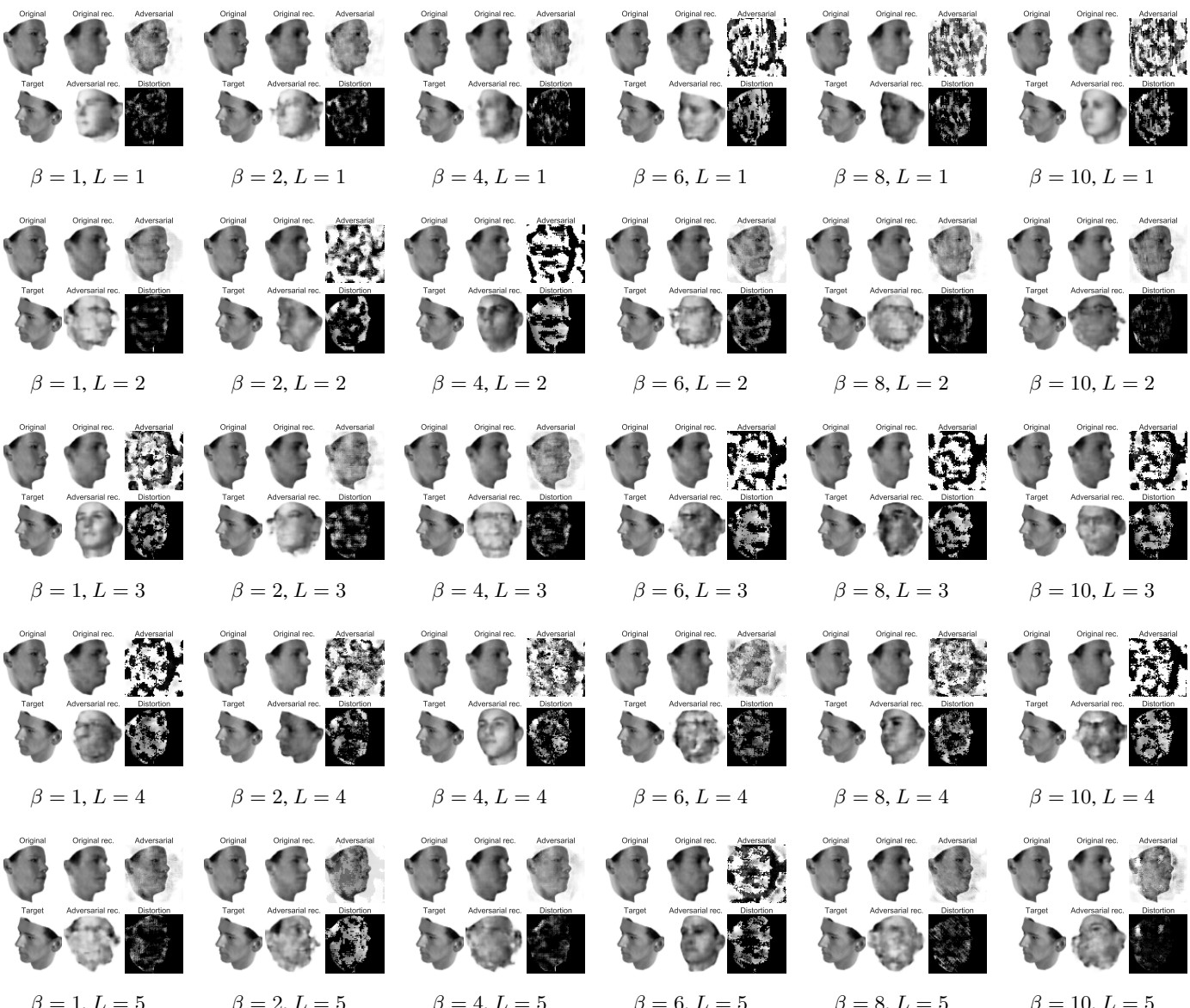

Figure H.26: Output attacks on 3D Faces for $\beta$-TCDLGMs for $\beta = \{1, 2, 4, 8, 10\}$ and $L = \{1, 2, 3, 4, 5\}$.

**Latent Attack**

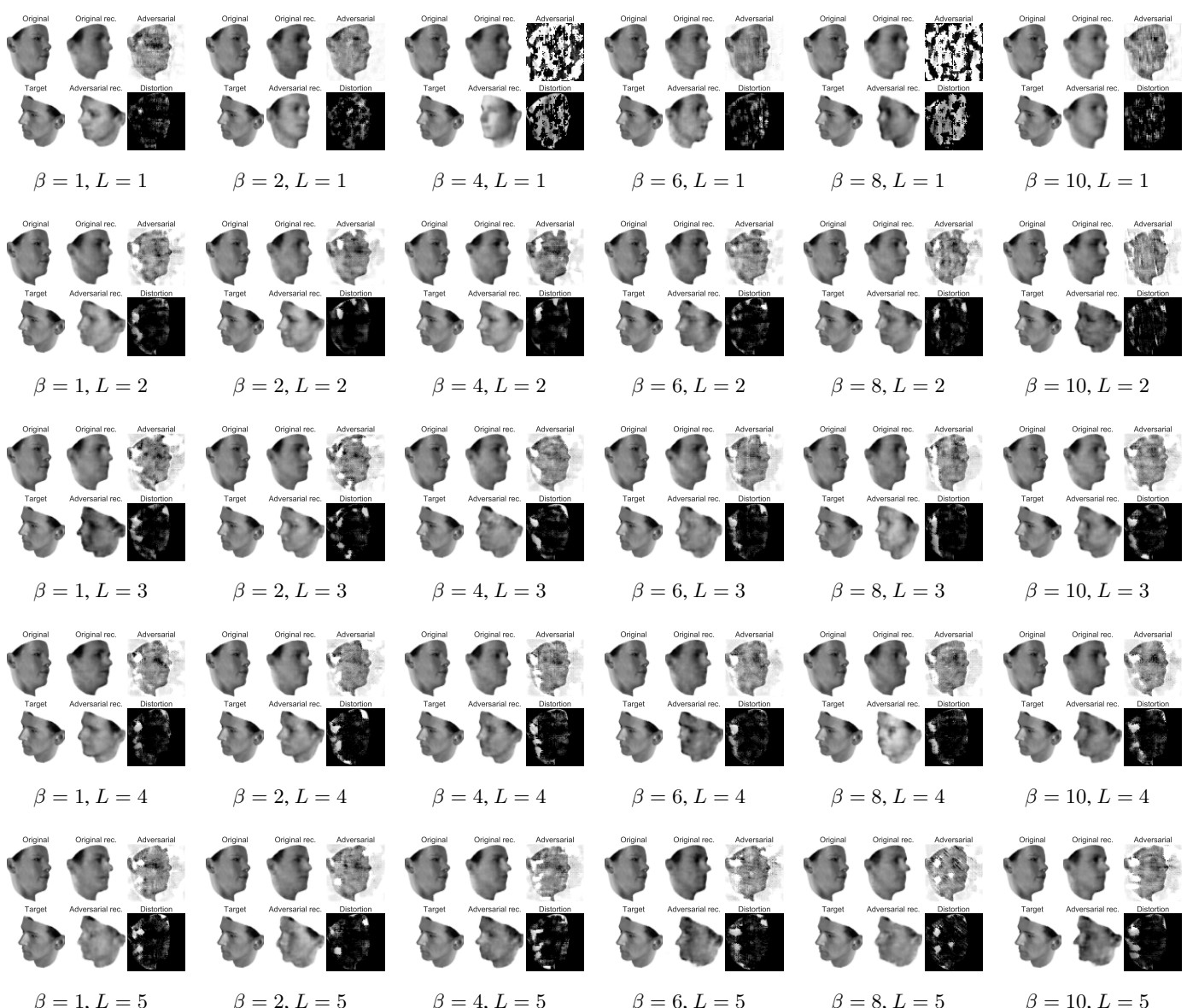

Figure H.27: Latent attacks on 3D Faces for $\beta$-TCDLGMs for $\beta = \{1, 2, 4, 8, 10\}$ and $L = \{1, 2, 3, 4, 5\}$.

### H.4.3 SEATBELT-VAES

**Output Attack**

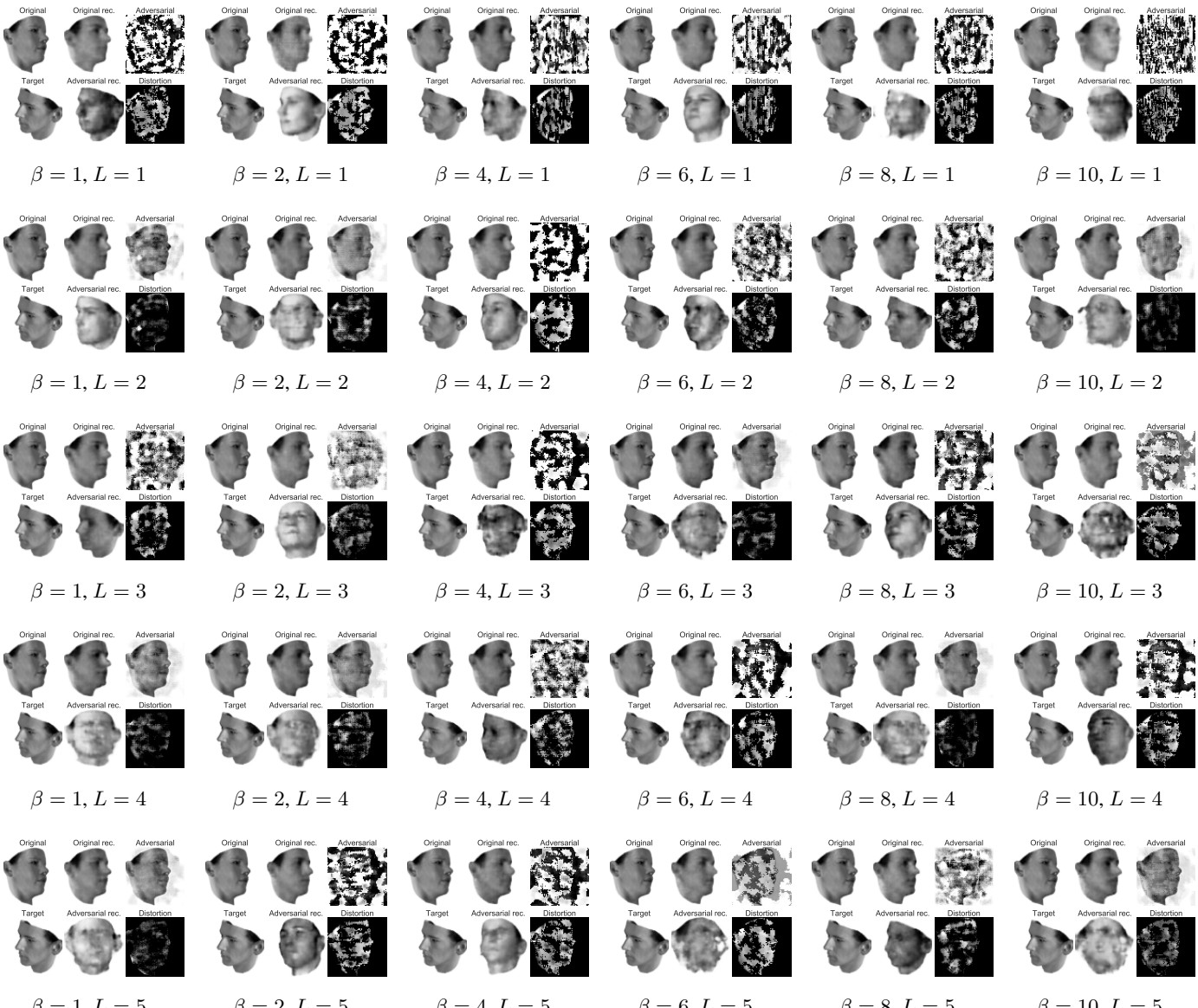

Figure H.28: Output attacks on 3D Faces for Seatbelt-VAEs for $\beta = \{1, 2, 4, 6, 8, 10\}$ and $L = \{1, 2, 3, 4, 5\}$.

**Latent Attack**

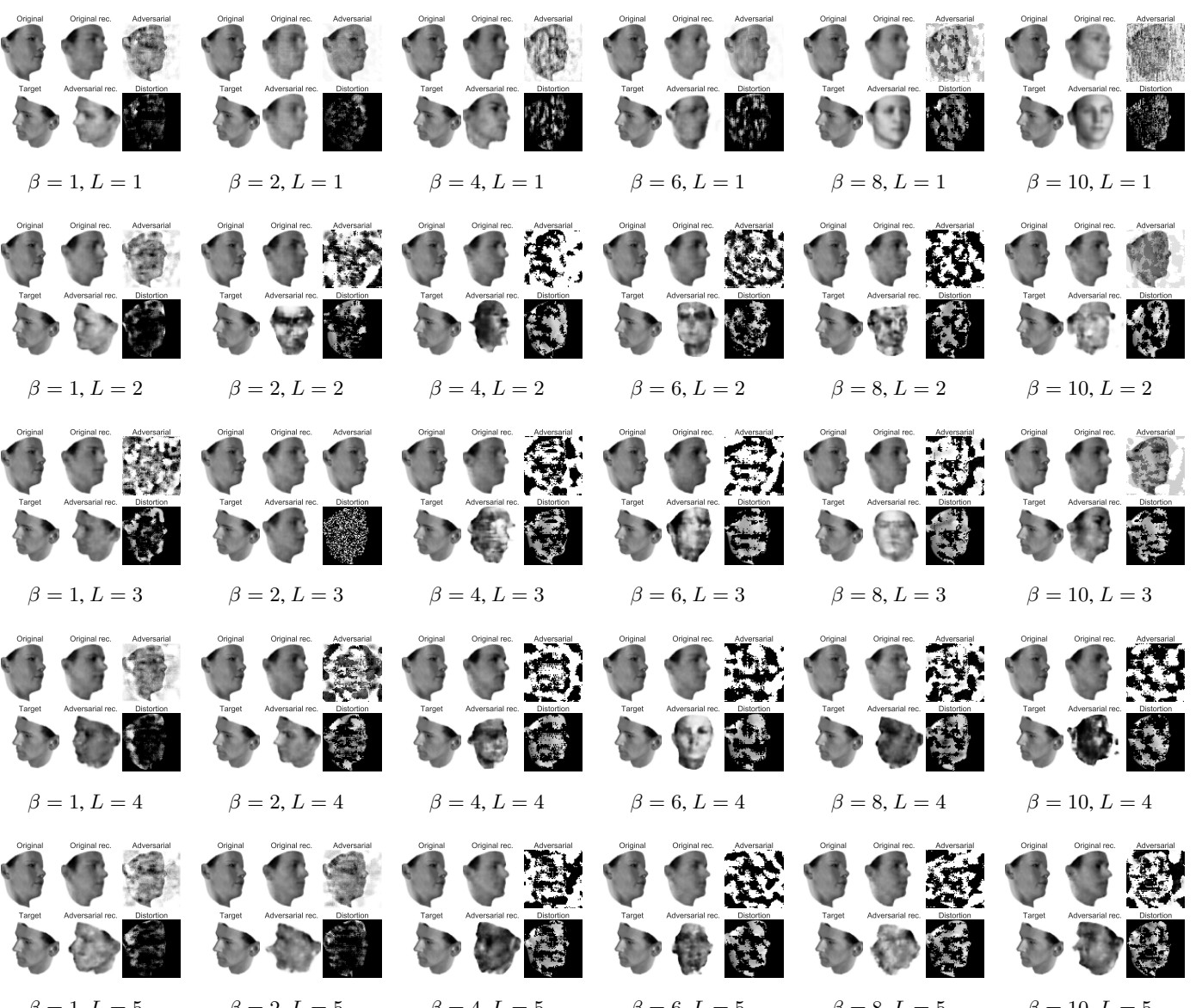

$\beta = 1, L = 1$   $\beta = 2, L = 1$   $\beta = 4, L = 1$   $\beta = 6, L = 1$   $\beta = 8, L = 1$   $\beta = 10, L = 1$

$\beta = 1, L = 2$   $\beta = 2, L = 2$   $\beta = 4, L = 2$   $\beta = 6, L = 2$   $\beta = 8, L = 2$   $\beta = 10, L = 2$

$\beta = 1, L = 3$   $\beta = 2, L = 3$   $\beta = 4, L = 3$   $\beta = 6, L = 3$   $\beta = 8, L = 3$   $\beta = 10, L = 3$

$\beta = 1, L = 4$   $\beta = 2, L = 4$   $\beta = 4, L = 4$   $\beta = 6, L = 4$   $\beta = 8, L = 4$   $\beta = 10, L = 4$

$\beta = 1, L = 5$   $\beta = 2, L = 5$   $\beta = 4, L = 5$   $\beta = 6, L = 5$   $\beta = 8, L = 5$   $\beta = 10, L = 5$

Figure H.29: Latent attacks on 3D Faces for Seatbelt-VAEs for $\beta = \{1, 2, 4, 6, 8, 10\}$ and $L = \{1, 2, 3, 4, 5\}$.

## H.5    CELEBA ADVERSARIAL ATTACK

### H.5.1    $\beta$-TCVAEs

**Output and Latent Attacks**

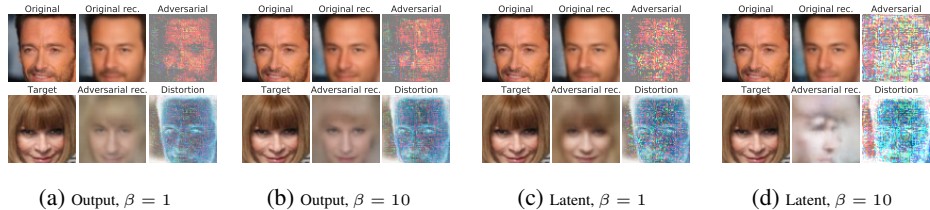

(a) Output, $\beta = 1$     (b) Output, $\beta = 10$     (c) Latent, $\beta = 1$     (d) Latent, $\beta = 10$

Figure H.30: Output (a) (b) and Latent (c) (d) attacks on CelebA on $\beta$-TCVAEs for $\beta = \{1, 10\}$ and $||z|| = 32$.

### H.5.2    $\beta$-TCDLGMs

**Output and Latent Attack**

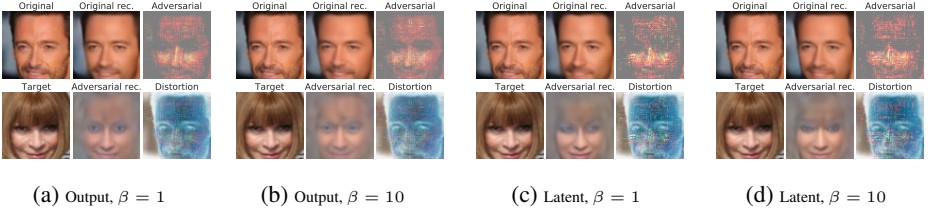

(a) Output, $\beta = 1$     (b) Output, $\beta = 10$     (c) Latent, $\beta = 1$     (d) Latent, $\beta = 10$

Figure H.31: Output (a) (b) and Latent (c) (d) attacks on CelebA on $L = 4$ $\beta$-TCDLGMs for $\beta = \{1, 10\}$.

### H.5.3    SEATBELT-VAEs

**Output and Latent Attack**

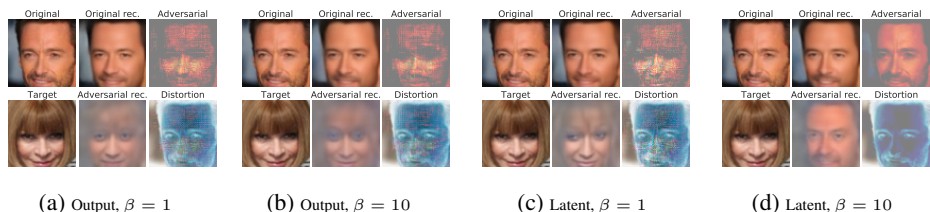

(a) Output, $\beta = 1$     (b) Output, $\beta = 10$     (c) Latent, $\beta = 1$     (d) Latent, $\beta = 10$

Figure H.32: Output (a) (b) and Latent (c) (d) attacks on CelebA on $L = 4$ Seatbelt-VAEs for $\beta = \{1, 10\}$.

# I  DATA GENERATION FROM MODELS

**Ancestral Sampling in CelebA**

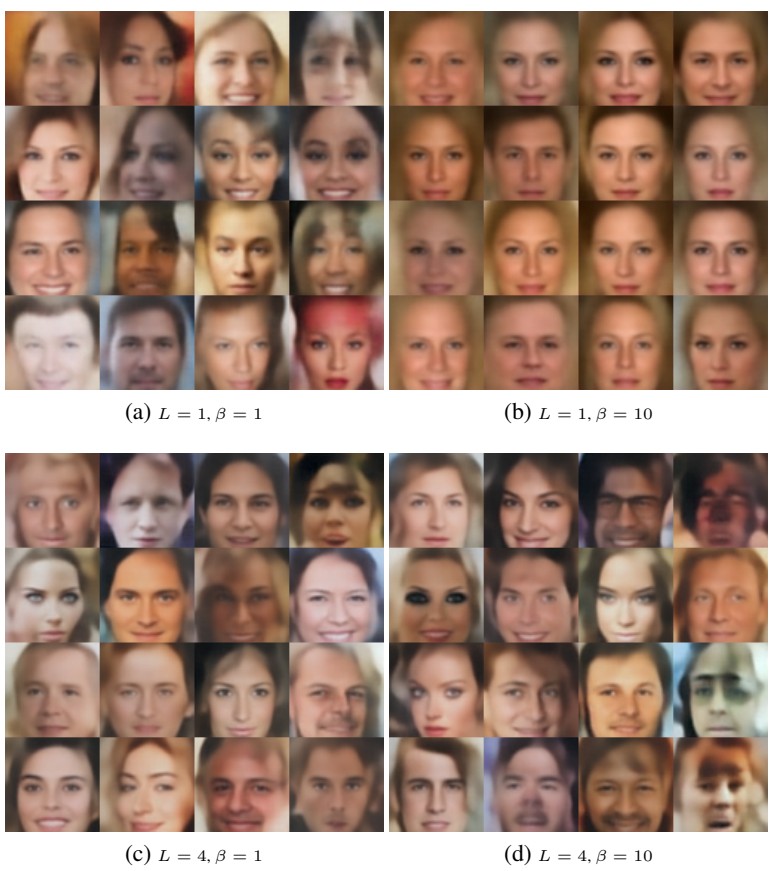

(a) $L = 1, \beta = 1$        (b) $L = 1, \beta = 10$

(c) $L = 4, \beta = 1$        (d) $L = 4, \beta = 10$

Figure I.33: Means of the decoder from ancestral sampling in $z$, for Seatbelt-VAEs with $L = \{1, 4\}$, $\beta = \{1, 10\}$. Note that there is a reduction in diversity of the samples for $L = 1$ (ie a $\beta$-TC VAE), $\beta = 10$, which is not the case for the samples from the $\beta = 10$ $L = 4$ Seatbelt-VAE.

**Latent Traversals for dSprites**

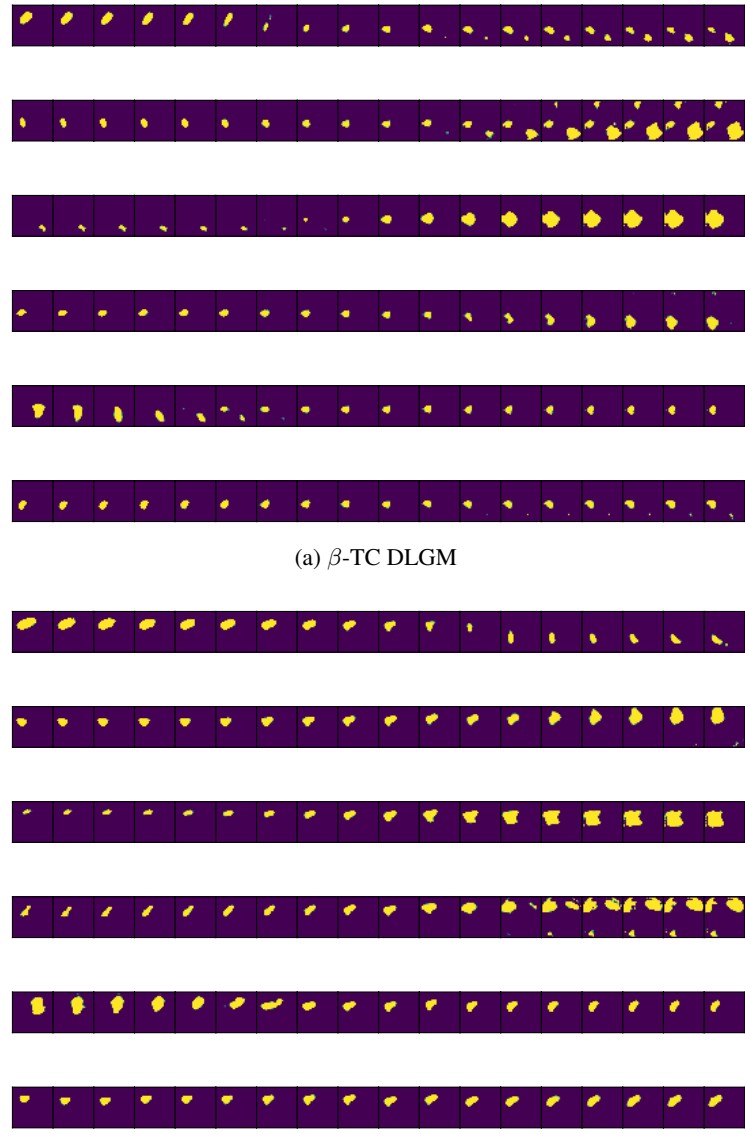

(a) $\beta$-TC DLGM

(b) Seatbelt-VAE

Figure I.34: Latent traversals in the $|z| = 6$ top layer of $L = 2$, $\beta = 2$ for a $\beta$-TC DLGM and a Seatbelt-VAE trained on dSprites. Note that the traversals do not capture the ground-truth factors of variation.

## J   MUTUAL INFORMATION GAP

The Mutual Information Gap (Chen et al., 2018) is average over ground truth factors of variation of the entropy-normalised difference between the greatest mutual information between the any of the units in $z$ and a given ground-truth factor of variation $\nu$ and the second-greatest such mutual information:

$$\text{MIG} = \sum_{k=1}^{K} \frac{1}{\mathcal{H}(\nu_k)} [\text{I}(z_{j^*}, \nu_k) - \max_m \text{I}(z_{j \neq j^*}, \nu_k)] \qquad \text{(J.1)}$$

where $z_j^* = \arg\max_j \text{I}(z_j, \nu_k)$.

Table J.2: MIG in $z^L$: for $L = 2$; for $\beta$-TC DLGMs and Seatbelt-VAEs; for a range of $\beta$ values; for dSprites, 3D Faces and Chairs.

| | dSprites | | 3D Faces | | Chairs | |
|---|---|---|---|---|---|---|
| $\beta$ | $\beta$-TC DLGM | Seatbelt-VAE | $\beta$-TC DLGM | Seatbelt-VAE | $\beta$-TC DLGM | Seatbelt-VAE |
| 1 | 0.0411 | 0.0475 | 0.0211 | 0.0300 | 0.0381 | 0.0134 |
| 2 | 0.3294 | **0.3589** | **0.4038** | 0.2200 | 0.2641 | 0.5366 |
| 4 | 0.2751 | 0.3235 | 0.2904 | 0.2349 | **0.7660** | 0.4963 |
| 6 | 0.3213 | 0.3258 | 0.1806 | 0.1890 | 0.1929 | 0.3111 |
| 8 | 0.3076 | 0.3182 | 0.2046 | 0.2301 | 0.2526 | 0.3329 |
| 10 | 0.3547 | 0.3415 | 0.1440 | 0.1053 | 0.1625 | 0.2802 |

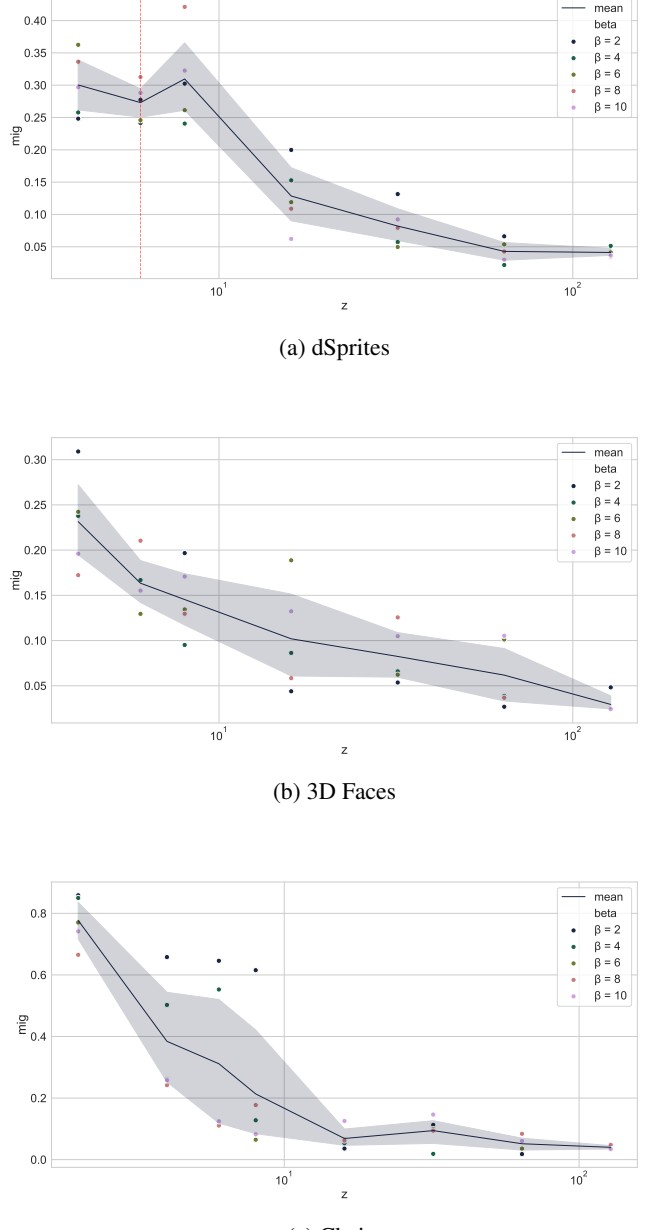

(a) dSprites

(b) 3D Faces

(c) Chairs

Figure J.35: MIG for $\beta$-TC VAEs as a function of $|z|$ for different values of $\beta$. Note that MIG decreases as we increase $|z|$, indicating that we get degenerate latent representations - that is different units in $z$ end up with similar mutual information to the same ground truth factors. The red line in a) is at $|z| = 6$, the number of ground-truth factors of variation for dSprites.

