# OpenReview forum: "Disentangling Improves VAEs' Robustness to Adversarial Attacks"
_ICLR.cc/2020/Conference — Reject_

### Official Review · AnonReviewer3 · 2019-10-14
**Official Blind Review #3**

**Rating:** 3

**Review:**

I am a bit confused about the model. While i understand the generative model in Fig 2a, i am a bit puzzled about the choice of proposal distribution eq(7)/Fig 2b for the Seatbelt-VAE. The key claim of the paper is that an attacker has to attack all layers at the same time to attack the reconstruction in eq (12). However, Figure 2b and eq(7) claim that all deeper layers only depend on the previous layer in the approximate posterior. Since in (12) we rely on the posterior for the attack, a successful attack on layer m < L should immediately generate the correct values from q_phi(z_{i+1}|z_i) i=m,...L-1.  So from that point of view it is not a seatbelt, as since in Fig 2b if the attacker manages to control z^1, he has immediate control of z_2 and therefore the now  attacker controlled z_2 will directly feed into the generated target. What might be is that the optimization problem (12) becomes harder to solve because of the increased model-complexity.

I am not too impressed by the attacks presented in Fig1 as well as the appendix. One of the key points of the old adversarial attacks was that the attack-image was indistinguishable from the true image by a human.  However, the adversarial inputs, even for VAEs are clearly not part of the distribution and the errors reported for eq (12) are very large to the point where attacking via the target image would probably be harder to spot. If we for example look at page 24, second row: there is no way, that the adversarial image has a likelihood similar to the target. This looks more like the algorithm did not manage to find a suitable direction.

I am therefore not sure whether the evaluation is correct: if we did not manage to find an attack image, does it proof there is none? And is it meaningful to report their error values if we did not manage an attack?
Btw: did the optimization of (12) begin with d=x_t-x or d=0? maybe starting with d=x_t-x would be more meaningful because it would make it easiest for an attacker to ensure the correct reconstruction.

Given the quality of the attack images, the error of eq(12) should be reported when choosing d=x_t-x as a baseline in Fig 5. It would also be good to see the actual VAE values.

Unfortunately, the reconstructions on dsprites are bad. But an important experiment could be to check whether you can attack the orientation of an object. Orientation is difficult to regularize via TC, since the parameterisation is inherently circular. Thus TC might make it difficult to encode orientation in higher layers and it should be easier to attack.


**Experience Assessment:**

I do not know much about this area.

**Review Assessment: Checking Correctness Of Derivations And Theory:**

I assessed the sensibility of the derivations and theory.

**Review Assessment: Checking Correctness Of Experiments:**

I assessed the sensibility of the experiments.

**Review Assessment: Thoroughness In Paper Reading:**

I read the paper at least twice and used my best judgement in assessing the paper.

---

> ### Author Response · Authors · 2019-11-10
> **(1/2) Authors' Response to Reviewer 3**
>
> (1/2) Thank you for your helpful comments. We quote each portion of your review and then respond. This means our response to you spans two openreview comments.
>
> "I am a bit confused about the model. While i understand the generative model in Fig 2a, i am a bit puzzled about the choice of proposal distribution eq(7)/Fig 2b for the Seatbelt-VAE. The key claim of the paper is that an attacker has to attack all layers at the same time to attack the reconstruction in eq (12). However, Figure 2b and eq(7) claim that all deeper layers only depend on the previous layer in the approximate posterior. Since in (12) we rely on the posterior for the attack, a successful attack on layer m < L should immediately generate the correct values from q_phi(z_{i+1}|z_i) i=m,...L-1. So from that point of view it is not a seatbelt, as since in Fig 2b if the attacker manages to control z^1, he has immediate control of z_2 and therefore the now attacker controlled z_2 will directly feed into the generated target. What might be is that the optimization problem (12) becomes harder to solve because of the increased model-complexity."
>
> Yes and thank you for a very insightful comment. If I match q(z_1|x_attack) for a seatbelt-VAE to the target perfectly (ie with a KL divergence of zero) then using an MC draw z_1*~q(z_1|x_attack) to define q(z_2|z_1)=q(z_2|z_1*) means that you should then match well in KL terms in z_2 as well, and so on up the chain.
>
> In the Appendix, G4 we show the effect of attacking z_1 only, z_L only and all z for Seatbelt-VAE. While the effect is small, we do observe a slight decrease in NLL of the target image under the attack for Chairs as we move from z_1 or z_L only (left and center plots) to all z (right-most plot). For 3D faces the effect is less clear.
>
> Let us give our intuition for having these extra terms in the loss when attacking Seatbelt:
> The addition of the ‘seatbelt’ autoregressive connections in the generative model means that there is a consequence to going off-manifold in higher layers of the posterior. The generator now directly uses the representations from these layers in defining the distribution over data. That means that not closely matching the posterior representation of the target image in higher layers does degrade performance.
>
> If one does not perfectly match in the lower layers, then as one propagates up the chain, defining each new conditionally-Gaussian posterior distribution on samples from the layers below, it is plausible that the approximate posterior distribution may move away from that of the target image.
>
> Concretely, consider two different distortions d^1 and d^2. Each gives rise to a different distribution in z_1: q(z_1|x+d^1) q(z_1|x+d^2). Consider the case that they have identical non-zero KL divergence to q(z_1|x_target). We cannot expect the draws from these two attacked distributions to give rise to similar latent representations higher up the chain - eg d^1 could happen to give draws defining a q(z_L|z_{L-1}) distribution that is far in KL from that produced from the target image, while d^2 might happen to do better.
>
> If one is lucky and the matching in high-layers comes directly from matching adequately in z_1, then the additional terms in the attack objective will not significantly alter the result. If however there are distortions that give similar KL divergences in z_1 but greatly diverge from the manifold in higher layers, the terms in the attack objective for the higher layers would help to steer the attacker back to good regions in the higher representations.
> We have made this nuanced discussion a key part of Section 4.

---

> > ### Author Response · Authors · 2019-11-10
> > **(2/2) Authors' Response to Reviewer 3**
> >
> > (2/2)
> > "I am not too impressed by the attacks presented in Fig1 as well as the appendix. One of the key points of the old adversarial attacks was that the attack-image was indistinguishable from the true image by a human. However, the adversarial inputs, even for VAEs are clearly not part of the distribution and the errors reported for eq (12) are very large to the point where attacking via the target image would probably be harder to spot. If we for example look at page 24, second row: there is no way, that the adversarial image has a likelihood similar to the target. This looks more like the algorithm did not manage to find a suitable direction.
> > Yes, we agree that adversarial attacks for VAEs are currently less effective than those for discriminative models. We are using the currently existing attack methods for VAEs."
> >
> > That the adversarial inputs look quite distorted is reasonable, as we are not trying to manipulate one value of output to be large (to fool a classifier), but instead trying to get hundreds of pixels to change so the output is close to a chosen target. It is, intuitively, a harder task. So even for a completely vanilla VAE the attack input does tend to be more distorted than for a classifier under attack. This reproduces the results in the papers proposing these attacks on vanilla VAEs, that adversarial attacks for these models have a noticeable amount of distortion. To improve our central argument, we have clarified this at the end of the first paragraph of Section 4.
> >
> > In Figure 1 we followed Gondim-Ribeiro et al. (2018) in displaying adversarial images for which the distortion gives us an adversarial loss that is just better than the average across different values of lambda. We have made this clearer in Section 5.2. In the case of Seatbelt-VAE, the adversarial reconstructions hardly change from the initial image, showing that, broadly, adversarial attacks are unsuccessful, and for 𝛽-TC VAEs the attacks are not ‘on target’.
> >
> > "I am therefore not sure whether the evaluation is correct: if we did not manage to find an attack image, does it proof there is none? And is it meaningful to report their error values if we did not manage an attack?"
> >
> > You raises a very relevant point. In Section 5 for each setting of each model we show results arising from 50 different adversarial attacks to each of 10 pairs of images. Overall we find them ineffective on disentangled models. Although one could assume that given sufficient iterations an attacker could eventually arrive at an effective attack, the fact that this does not occur within the 500 attacks (unlike attacks on vanilla VAEs) highlights the fact that these disentangled models are generally more robust, even if theoretically they could eventually be successfully attacked: that a successful attack is not obtainable in this setup is itself a mark of robustness.
> >
> > "Btw: did the optimization of (12) begin with d=x_t-x or d=0? maybe starting with d=x_t-x would be more meaningful because it would make it easiest for an attacker to ensure the correct reconstruction. Given the quality of the attack images, the error of eq(12) should be reported when choosing d=x_t-x as a baseline in Fig 5. It would also be good to see the actual VAE values."
> >
> > When performing gradient descent to obtain an adversarial input, if you start with d=x_t-x such that the adversarial input x_^*=x_t, then already you are basically already done: you are feeding in x_t so as to reconstruct x_t. The attackers do aim for their input to be relatively undistorted from the initial input. Thus, in line with Gondim-Ribeiro et al. (2018), one of the papers where these attacks are proposed, our initial d is made from sampling uniformly on the range [-10^(-8),10^(-8)].
> >
> > We do plot the distance between the adversarial input and the target image. See Appendix G, Figures G.8 - G.10, subfigures (a), (c), (e) of each.
> > Could you clarify what you mean by the actual VAE values?
> >
> > "Unfortunately, the reconstructions on dsprites are bad. But an important experiment could be to check whether you can attack the orientation of an object. Orientation is difficult to regularize via TC, since the parameterisation is inherently circular. Thus TC might make it difficult to encode orientation in higher layers and it should be easier to attack."
> >
> > Please can you clarify which reconstructions you are referring to? Note of course that for 𝛽-TC-VAEs (unlike Seatbelt-VAE) reconstructions quality degrades briskly as 𝛽 increases.
> > You make a very good point about rotation, and we have added experiments on this in Appendix H.1, where we attempt to rotate a dSprites heart by 180 degrees. In line with our other results, 𝛽-TC VAE is more robust than a Vanilla VAE, and Seatbelt-VAE is more robust still.

---

### Official Review · AnonReviewer2 · 2019-10-15
**Official Blind Review #2**

**Rating:** 6

**Review:**

The authors of this paper propose a new VAE model called seatbelt-VAE and investigate its robustness to output and latent adversarial attacks. Inspired by beta-TCVAE, DLGM, and BIVA, seatbelt-VAE allows multiple latent layers, enforce disentanglement via weighted total correlation on the top latent layer, and conditions the likelihood on all latent layers. Robustness to adversarial attacks is the focus in experiments. Visual and quantitative comparisons show that seatbelt-VAE is more robust for latent attack than benchmarks. Specifically I have the following three concerns:

1. Defining ELBO using samples from the entire dataset may bring in some benefits, but it complicates the calculation of ELBO and related distributions when minibatch or single sample are used in learning and inference. Please explicitly discuss this issue.

2. I would like to see how seatbelt-VAE performs in sampling and generating new samples, instead of just reconstruction.

3. Similarly, it would be beneficial to investigate disentanglement, that is the interpretation of the top latent factors in seatbelt-VAE.

Minors:
Factor analysis -> factor analysis
section X -> Section X
figure X -> Figure X


**Experience Assessment:**

I have published in this field for several years.

**Review Assessment: Checking Correctness Of Derivations And Theory:**

I assessed the sensibility of the derivations and theory.

**Review Assessment: Checking Correctness Of Experiments:**

I carefully checked the experiments.

**Review Assessment: Thoroughness In Paper Reading:**

I read the paper at least twice and used my best judgement in assessing the paper.

---

> ### Author Response · Authors · 2019-11-10
> **Authors' Response to Reviewer 2**
>
> Thank you for your helpful comments.
>
> 1. "Defining ELBO using samples from the entire dataset may bring in some benefits, but it complicates the calculation of ELBO and related distributions when minibatch or single sample are used in learning and inference. Please explicitly discuss this issue."
>
> Yes, good point. While we do use the notation of Esmaeili et al. (2019), writing our ELBO as a function of the dataset, rather than per-datapoint, we have followed the completely standard approach for training VAE-like models in that we trained each model to maximise its ELBO using stochastic gradient descent on minibatches of data. But yes, the terms in the ELBO that include average encoding distributions (which is after all a mixture of as many components as there are datapoints) require careful handling. For these terms we use a generalisation of the MWS sampler from Chen et al. (2018) to hierarchical models, enabling us to estimate them using minibatches. We discuss this in Appendix B, but we have added further discussion of these sampling methods as Section 3.1.
>
> 2. "I would like to see how seatbelt-VAE performs in sampling and generating new samples, instead of just reconstruction."
>
> We can show randomly generated samples from VAE and have added such samples as Appendix I.
>
> 3. "Similarly, it would be beneficial to investigate disentanglement, that is the interpretation of the top latent factors in seatbelt-VAE."
>
> Currently we are not satisfied that there are good metrics that suit disentangled hierarchical VAEs, which we believe is an area for future research.
>
> We can naively calculate the MIG (Chen et al., 2018) for the top layer. In Appendix J we now have a table showing the MIG for their top layer of 𝛽-TC DLGMs and Seatbelt-VAEs with L=2, for a range of values of 𝛽, for dSprites, 3D Faces and Chairs. We find that we do get a good MIG score, but that does not necessarily mean that sampling down from chosen values of z_L, spanning each latent variable, gives well-controlled generation. To avoid the implications that high values of MIG might lead to — that it means we have axis-wise control over generation — we chose to omit these results from the paper, though we now realise it is a key point to address. We now show in Appendix I latent traversals for L=2, 𝛽=2 𝛽-TC DLGMs and Seatbelt-VAEs trained on dSprites.
>
> We have also included in Appendix J MIG plots for 𝛽-TC VAEs as a function of |z| for different values of 𝛽, which shows that MIG decreases as we increase |z|. This would be explained by there being degeneracy in z: different units in z end up having similar MI to the same ground truth factor of variation, which explains the gap between the top-most and second top MI shrinking.

---

### Official Review · AnonReviewer1 · 2019-10-24
**Official Blind Review #1**

**Rating:** 3

**Review:**

This paper examines adversarial attacks to a VAE. It is known that by small norm perturbations on the conditioning input x of a VAE can dramatically change the generated output. This paper
empirically illustrates that alternative objectives can improve robustness, in the sense of previously proposed adversarial attacks such as (Tabacof et al. (2016); Gondim-Ribeiro et al. (2018); Kos et al.(2018)). This paper is concerned with the stability of reconstructions and their quality, and proposes Seatbelt-VAE to remedy some of the shortcomings in the original VAE objective.

The key idea of the Seatbelt-VAE is introducing a conditionally Gaussian chains (of length L) in the encoder and decoder distributions of a VAE. This is a plausible and sensible idea. Then the authors evaluate the robustness of reconstructions under various output attacks.

The methodological part of the paper is quite well written and easy to follow, despite the fact that it is somewhat overloaded with too many abbreviations. The experimental section is harder to read as the motivations and its organization is not clearly stated. Overall, this section feels as if it is too hastily written, many results put into appendix without much discussion. The organization can be much more improved.

The disentanglement achieved by this novel representation is characterized only anecdotally and by contrasting the resulting objectives to a beta-VAE. It would have been much more informative to illustrate and discuss further the representations learned by such a conditionally Gaussian architecture. Figure 6 and 7 partially try to achieve this by showing the interplay of depth L and the inverse-dispersion parameter beta but I found it hard to interpret this results, for which an entire page is devoted.

In the experiments, the ELBO is reported for various methods. I would argue that the ELBO is not a very representative proxy for robustness. For example VAE ELBO and beta-VAE ELBO are both lower bounds of the true marginal likelihood and it is possible that beta-VAE is much lower while attaining a higher robustness in the sense of being resilient to suitably defined attacks.

The authors claim that there are no clear classification tasks for the datasets -- but this is not accurate as both celeb-a has clear classification tasks in the form of predicting attributes. It would have been really quite informative if adversarial accuracy on downstream tasks would have been reported. Relying on qualitative results in Figure 1 is only providing partial evidence about the approach.

Robustness to independent noise, as the authors have, is a good experiment to have -- however typical adversarial examples may be quite structured and such a randomized strategy may not give an accurate indication about the nature of the representation.

Overall, the paper is quite promising but I feel that one more iteration maybe needed.

**Experience Assessment:**

I have published one or two papers in this area.

**Review Assessment: Checking Correctness Of Derivations And Theory:**

I assessed the sensibility of the derivations and theory.

**Review Assessment: Checking Correctness Of Experiments:**

I carefully checked the experiments.

**Review Assessment: Thoroughness In Paper Reading:**

I read the paper thoroughly.

---

> ### Author Response · Authors · 2019-11-10
> **(1/2) Authors' Response to Reviewer 1**
>
> (1/2) Thank you for your helpful comments. We quote the queries from your review and then respond. This means our response to you spans two openreview comments.
>
> "The key idea of the Seatbelt-VAE is introducing a conditionally Gaussian chains (of length L) in the encoder and decoder distributions of a VAE. This is a plausible and sensible idea. Then the authors evaluate the robustness of reconstructions under various output attacks.
>
> The methodological part of the paper is quite well written and easy to follow, despite the fact that it is somewhat overloaded with too many abbreviations. The experimental section is harder to read as the motivations and its organization is not clearly stated. Overall, this section feels as if it is too hastily written, many results put into appendix without much discussion. The organization can be much more improved."
>
> This is a good observation and we have clarified Section 5 with a short introduction to ensure our motivations, claims, and aims are better defined, as well as re-writing various other sections for increased clarity.
>
> "The disentanglement achieved by this novel representation is characterized only anecdotally and by contrasting the resulting objectives to a 𝛽-VAE. It would have been much more informative to illustrate and discuss further the representations learned by such a conditionally Gaussian architecture. Figure 6 and 7 partially try to achieve this by showing the interplay of depth L and the inverse-dispersion parameter 𝛽 but I found it hard to interpret this results, for which an entire page is devoted."
>
> Yes, we agree: exploring the disentanglement of these models is a very interesting avenue of research. We currently are not satisfied that there are good metrics that suit disentangled hierarchical VAEs, which we believe is an area for future research. However, we do have both MIG values and latent traversals for the top layer of Seatbelt-VAE and have added them as appendices - J and I respectively.
>
> The two main points of this paper are 1) currently-existing disentangled VAEs are more robust to adversarial attacks proposed for VAEs, and 2) we propose a hierarchical VAE that is effective at resisting adversarial attacks that also does not suffer from the same decrease in reconstruction quality as the degree of disentangling regularisation is increased.. Thus we have less interest in the use of a disentangled latent space for controlled generation. We have made the aims of the paper clearer at the end of the introduction.
>
> Also to clarify, Figure 6 shows the average NLL of the target image when a trained model is under attack and Figure 7 shows the average attack objectives reached by the adversary. In both figures we vary 𝛽 (degree of regularisation) and L (number of layers).
>
> "In the experiments, the ELBO is reported for various methods. I would argue that the ELBO is not a very representative proxy for robustness. For example VAE ELBO and 𝛽-VAE ELBO are both lower bounds of the true marginal likelihood and it is possible that 𝛽-VAE is much lower while attaining a higher robustness in the sense of being resilient to suitably defined attacks."
>
> In Figure 3 we plot the final ELBO, ie without the 𝛽 penalty on the KL divergence term, as a function of the 𝛽 used in training, for 𝛽-TC VAE, 𝛽-TC DLGM Seatbelt-VAE. We show that with Seatbelt-VAE we can achieve lesser degradation in reconstruction quality as we increase 𝛽. Later in the paper we show that the model is more robust to adversarial attacks, and as such Figure 3 should not be taken as evidence that Seatbelt-VAE is more or less robust than other models, solely that it can achieve better reconstructions than 𝛽-TC VAE at the same 𝛽 value. We agree that this point was not made clear in Section 5.1, and we have altered it to make our argument clearer.
>
> It may well be possible to design attacks that utilise the fact that a model has disentangled representations. We are not aware of research in that direction. Disentangling seems to be associated with robustness. One aim in this paper is to show that the currently-existing attacks defined for VAEs are less effective against disentangled models. Developing new attacks is beyond the scope of this paper.

---

> > ### Author Response · Authors · 2019-11-10
> > **(2/2) Authors' Response to Reviewer 1**
> >
> > (2/2)
> >
> > "The authors claim that there are no clear classification tasks for the datasets -- but this is not accurate as both celeb-a has clear classification tasks in the form of predicting attributes.It would have been really quite informative if adversarial accuracy on downstream tasks would have been reported. Relying on qualitative results in Figure 1 is only providing partial evidence about the approach."
> >
> > This is correct, we have clarified our language around classification tasks and metrics.
> > It is of course true that celeb-a does have attributes one can aim to predict, but given that there are 40 such binary attributes we thought that studying them would be hard to interpret, without substantially improving the paper in our view.
> >
> > More generally, we prefer to take a task-agnostic point of view in this work given the broad range of potential downstream tasks (and models for them) one could choose to study. As such, performance on the downstream tasks may be more reflective of the robustness of the model used for that task than the robustness of the generative model. Thus we prefer to side-step classifier based metrics. We make this point of view clear in section 5.2.
> >
> > Yes, we agree that Figure 1 and the further examples of attacks given in the appendix are only qualitative evidence. We show quantitative evidence: Figure 5 shows the values of the attack objective reached attacking 𝛽-TC VAEs as a function of 𝛽; Figures 6 and 7 respectively show the NLL of the target image under the attacked model, and the values reached of the attack objectives for Seatbelt-VAEs, for each as we vary 𝛽, L.
> >
> > "Robustness to independent noise, as the authors have, is a good experiment to have -- however typical adversarial examples may be quite structured and such a randomized strategy may not give an accurate indication about the nature of the representation."
> >
> > Yes we completely agree that structured attacks are not well represented by this experiment, but that is not the aim. We show the independent noise experiments to gain some intuition about the robustness we observe. With these experiments we demonstrate that disentangled VAEs are inherently denoising, which suggests they are more robust than their non-disentangled counterparts in more ways than just in terms of adversarial attack.

---

### Decision · Program_Chairs · 2019-12-19

**Decision:**

Reject

**Comment:**

This work a "Seatbelt-VAE" algorithm to improve the robustness of VAE against adversarial attacks. The proposed method is promising but the paper appears to be hastily written and leave many places to improve and clarify. This paper can be turned into an excellent paper with another round of throughout modification.

---

> ### Author Response · Authors · 2020-02-19
> **Response from Authors - Paper Re-written in light of feedback received**
>
> In light of the Program Chairs' helpful and frank comments, we have substantially re-worked the paper for clarity of exposition.
> Our updated version can be found on arxiv, at https://arxiv.org/abs/1906.00230.